_Article_

# NFATc1 drives Orai3 transcription and proteolysis by harnessing epigenome differences in the MARCH8 promoter

Sharon Raju (ID), Akshay Sharma, Sakshi Dahiya (ID), Gyan Ranjan (ID) & Rajender K Motiani (ID) ✉

## Abstract

**Several autonomous mechanisms regulate protein expression, such as transcription, translation, post-translational modifications, and epigenetic changes. Rarely, these processes are controlled by the same molecular player with overlapping roles. Here, we reveal that transcription factor NFATc1 regulates both transcription and degradation of the Ca²⁺ channel Orai3 in a context-dependent manner. We demonstrate that NFATc1 drives Orai3 transcription in non-metastatic pancreatic cancer cells. In invasive and metastatic pancreatic cancer cells, NFATc1 induces Orai3 lysosomal degradation by transcriptionally enhancing MARCH8 E3-ubiquitin ligase. We show that MARCH8 physically interacts with Orai3 intracellular loop eventually resulting in its ubiquitination at the N-terminal. Mechanistically, the dichotomy in the regulation of Orai3 expression emerges from the differences in MARCH8 epigenetic landscape. We uncover that MARCH8 promoter is hypermethylated in non-metastatic cells. Importantly, we demonstrate that MARCH8 restricts pancreatic cancer metastasis by targeting Orai3 degradation, thereby highlighting the pathophysiological importance of this signaling module. Taken together, we report a unique and clinically relevant scenario wherein the same transcription factor both enhances and curtails the expression of a target protein in cancer.**

**Keywords** Orai3; NFATc1; MARCH8; Transcription; Ubiquitination
**Subject Categories** Cancer; Metabolism; Signal Transduction

## Introduction

Proteins are involved in all aspects of cellular life, from cell division to cell death. The regulation of proteins in the cell is a complex process involving various mechanisms that control when and how proteins are synthesized, modified, localized, and degraded (Ross et al, 2021). Transcription regulates the amount of mRNA produced from a gene, while alternative splicing leads to diverse variants. Post-transcriptionally, mRNA stability and translation are controlled by microRNAs and RNA-binding proteins. Once synthesized, proteins can undergo numerous post-translational modifications (PTMs), such as phosphorylation, ubiquitination, and acetylation. These PTMs alter protein activity, stability, and interactions. To prevent accumulation of damaged or excessive proteins, cells employ mechanisms like proteasome/lysosomal-mediated degradation and autophagy (Ross et al, 2021). Rarely, these regulatory processes for a single target protein are orchestrated by the same molecular player with intermingling effects on several independent mechanisms. Typically, the components involved in a specific protein regulatory process are limited to that process alone. These regulatory mechanisms work together to ensure precise spatio-temporal control over cellular functions and responses. But in pathophysiological conditions like cancer, proteome imbalance is a prominent feature due to alterations in these processes (Chen et al, 2023b).

Pancreatic cancer (PC) is among the deadliest forms of cancer as it is highly metastatic in nature (Fitzmaurice et al, 2019). The 5-year survival rate of pancreatic cancer is around 13%, which is among the lowest. Ca²⁺ signaling plays a key role in regulating oncogenesis and metastasis in cancer (Vashisht et al, 2015; Tanwar et al, 2020; Bhatnagar et al, 2025). However, the functional relevance of Ca²⁺ dynamics in pancreatic cancer tumorigenesis is poorly understood. The store-operated calcium entry (SOCE) pathway is a ubiquitous process for Ca²⁺ influx into the cells. It is a process that starts with depletion of Ca²⁺ stores in the endoplasmic reticulum and results in Ca²⁺ influx across the plasma membrane. ER Ca²⁺ sensors STIM1/STIM2 sense the depletion of ER Ca²⁺ stores, oligomerize, and move towards ER-plasma membrane junctions. STIM proteins physically interact with calcium release-activated calcium (CRAC) channels, i.e., Orai channels in the plasma membrane. This association activates Orai channels and Ca²⁺ influx across the plasma membrane (Lopez et al, 2020; Kim et al, 2013). Dysregulation in Orai channels in particular Orai3 expression and function is associated with various types of cancer (Chalmers and Monteith, 2018; Vashisht et al, 2015; Tanwar et al, 2020). However, the molecular mechanisms regulating Orai3 expression, stability, and degradation remain largely unappreciated.

An earlier study from our group demonstrated that Orai3 forms a functional SOCE channel in pancreatic cancer cells and regulates key hallmarks of oncogenesis. Further, we demonstrated that Orai3 is transcriptionally upregulated in human pancreatic tumor

Laboratory of Calciomics and Systemic Pathophysiology (LCSP), Regional Centre for Biotechnology (RCB), Faridabad, Delhi-NCR, India. ✉E-mail: rajender.motiani@rcb.res.in

samples as compared to normal patient samples (Arora et al, 2021). Pancreatic cancerous cells have higher Orai3 mRNA expression than normal cells, but what drives Orai3 transcriptional upregulation remains unappreciated. In addition to transcriptional regulation, protein degradation plays an essential role in regulating protein levels. Nevertheless, Orai3 degradation process remains completely unexplored. Gaining insight into the mechanisms that drive Orai3 protein degradation can aid in developing treatments for the diseases associated with aberrant Orai3 expression, particularly pancreatic, breast, lung, and prostate cancers.

Here, using unbiased and robust bioinformatic analysis, we found that the $Ca^{2+}$-sensitive NFATc1 transcription factor has putative binding sites on the Orai3 promoter. Interestingly, NFATc1 gain-of-function and loss-of-function studies revealed a dichotomy in the regulation of Orai3 by NFATc1 in non-metastatic versus invasive and metastatic cells. Our data demonstrate that NFATc1 positively regulates Orai3 transcription in non-metastatic pancreatic cancer cells. Hence, it generates a positive feedforward loop wherein a $Ca^{2+}$-sensitive transcription factor drives transcription of a $Ca^{2+}$ channel. Surprisingly, NFATc1 acts as a bimodal regulator in metastatic pancreatic cancer cells by inducing both Orai3 transcription and Orai3 protein degradation. We show that downstream of NFATc1, MARCH8 E3 ubiquitin ligase ubiquitinates and degrades Orai3 via the lysosomal pathway. Further, we performed site-directed mutagenesis and deletions studies to uncover that MARCH8 physically interacts with Orai3 intracellular loop eventually resulting in selective ubiquitination of K2 residue with Orai3 N-terminal. Mechanistically, NFATc1 transcriptionally regulates MARCH8 in pancreatic cancer cells depending on the epigenetic profile of the MARCH8 promoter. We reveal that the MARCH8 promoter is hypomethylated in invasive and metastatic pancreatic cancer cells compared to non-metastatic cells. Therefore, NFATc1 stimulates MARCH8 transcription in non-metastatic cells and thereby mediates Orai3 protein degradation. Finally, we establish that MARCH8 acts as a tumor suppressor in pancreatic cancer. Our in vitro migration assays and in vivo zebrafish xenograft experiments show that MARCH8 negatively regulates metastasis. To summarize, we reveal that same transcription factor controls both mRNA transcription and protein degradation of the identical target. Further, we have identified and characterized MARCH8 as an important regulator of Orai3-driven pancreatic cancer metastasis. Importantly, the dichotomy in Orai3 regulation highlights an intricate pathway in pancreatic cancer cells that can control disease outcomes.

## Results

### NFATc1 binds to the Orai3 promoter and regulates Orai3 expression

To delineate the transcriptional regulation of Orai3, we carried out extensive bioinformatics analysis of the human Orai3 promoter through three independent tools: Eukaryotic promoter database (Perier, 2000), PSCAN (Zambelli et al, 2009), and Contra_V3 (Kreft et al, 2017). PSCAN was utilized to identify all potential transcription factor binding sites within the Orai3 promoter. PSCAN leverages the JASPAR Core 2020 transcription factor position weight matrix database. The analysis revealed that the NFAT family of transcription factors has potential binding sites on the Orai3 promoter (Fig. 1A). To validate this observation, the Orai3 promoter was analyzed for NFATc-binding sites using the EPD-Search Motif Tool, applying a stringent $P$ value cut-off of $P = 0.01$. Moreover, we assessed the sequence of the Orai3 promoter for NFATc-binding sites using Contra_V3 with a similarity matrix value = 0.75 and core value = 0.90. All three tools, each using a different algorithm, consistently identified three NFATc binding sites on the Orai3 promoter. The three NFATc binding sites were present at −920bp, −990bp, and −1017bp before the transcription start site. Furthermore, the multi-species alignment of the Orai3 promoter showed that these binding sites were conserved across primate species (Fig. 1B,C). Thus, our robust bioinformatics analysis suggests that NFAT transcription factors may transcriptionally regulate Orai3.

The NFAT family of transcription factors has four $Ca^{2+}$-sensitive members: NFATc1, 2, 3, and 4 (Müller and Rao, 2010). To identify which isoforms of NFAT regulate Orai3, we performed overexpression studies of the four isoforms of NFAT in the HEK-293T cell line. We found that only NFATc1 significantly upregulates Orai3 expression (Fig. 1D,E). Further, only NFATc1 overexpression significantly increased Orai3 mRNA levels compared to empty vector control and other NFATc isoforms (Fig. 1F). To investigate the association of NFATc1 and Orai3 levels in pancreatic tissue, we carried out expression analysis of NFATc1 and Orai3 using the "GEPIA" (Gene Expression Profiling Interactive Analysis) database (Tang et al, 2017) and observed a positive correlation between NFATc1 and Orai3 mRNA expression in pancreatic tissue (Fig. 1G).

To corroborate the bioinformatic analysis and to examine the role of predicted NFATc1 binding sites on Orai3 promoter, we performed dual-luciferase assays. Wild-type Orai3 promoter, Orai3 promoter with individual NFATc1 binding site deletions, and truncated Orai3 promoter with no NFATc1 binding sites were cloned into a luciferase reporter vector (Fig. 1H). Next, we carried out luciferase assays with NFATc1 overexpression in PANC-1 pancreatic cancer cells. The results showed an increase in relative luciferase activity with NFATc1 overexpression compared to the empty vector control in the wild-type Orai3 promoter and a substantial decrease in luciferase activity upon NFATc1 overexpression in Orai3 promoter deletions as well as truncated Orai3 promoter (Fig. 1I), suggesting that NFATc1 positively regulates Orai3 transcription. The results suggest that three predicted NFATc1-binding sites; −1017, −990, and −920 are essential for NFATc1-driven Orai3 transcription. To determine if only NFATc1 and no other NFATc isoforms physically bind to the Orai3 promoter at the predicted binding sites, we performed Chromatin Immunoprecipitation (ChIP) assays in PANC-1 cells either overexpressing pEGFPC1-NFATc1, pMIG-hNFATc2, pREP-NFATc3, pEGFP-C1-NFATc4, or an empty vector control (pEGFP-C1). For the immunoprecipitated cross-linked sonicated DNA amplification, primers were designed to amplify the individual NFATc1 binding sites. Primer Set 1 amplifies −1017 region and Primer Set 2 amplifies −990 and −927 regions. The ChIP-qPCR analysis showed that the Orai3 promoter with putative NFATc1 binding sites was highly enriched in NFATc1 pulldown compared to mock IP samples and other NFATc isoforms pulldown, thereby confirming physical binding of NFATc1 on Orai3 promoter (Fig. 1J). Also, these results suggest

that NFATc1 binds to all the predicted binding sites on the Orai3 promoter and does not show any noteworthy preference. Collectively, the ChIP and dual-luciferase assays provide substantial evidence for the binding of NFATc1 at the predicted binding sites on the Orai3 promoter.

## NFATc1 overexpression dichotomously regulates Orai3 in non-metastatic v/s metastatic pancreatic cancer cells

To delineate the role of NFATc1 in regulating Orai3 expression and function, we chose six human pancreatic cancer cell lines: MiaPaCa-2

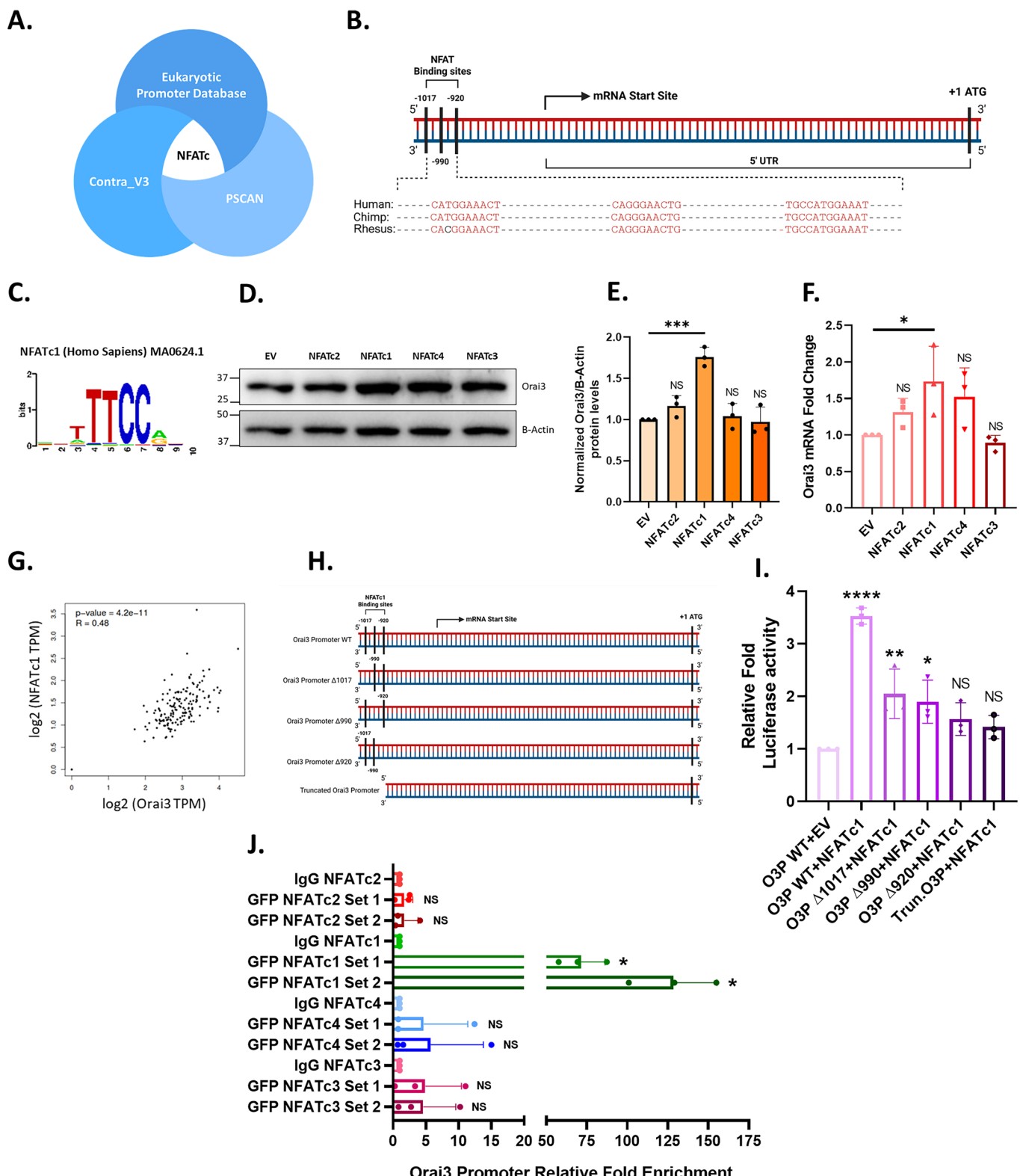

**Figure 1. NFATc1 binds on Orai3 Promoter and regulates Orai3 expression.**

(A) In-silico analysis via Eukaryotic Promoter Database, P-Scan and Contra_V3 recognize NFATc1 as a potential transcriptional regulator of Orai3. (B) Identification of putative NFATc binding sites on the Human Orai3 promoter using the EPD-Search Motif Tool at P value cut-off of 0.01 and bioinformatic characterization of conserved NFATc binding sites on the cross-species alignment of the Orai3 promoter using the ContraV3 transcription factor binding analysis tool. (C) Position weight matrix for human NFATc1 consensus binding sequence. (D) Representative western blot for Orai3 levels with transient overexpression of NFATc1, 2, 3, and 4 in HEK-293T cells. (E) Densitometric analysis of Orai3 levels in NFATc1, 2, 3, and 4 overexpressed HEK-293T cells ($N = 3$). ****$P = 0.0001$. (F) qRT-PCR analysis showing transcriptional upregulation of Orai3 upon overexpression of NFATc1, 2, 3, and 4 in HEK-293T cells compared to empty vector control ($N = 3$). *$P = 0.0388$. (G) NFATc1 and Orai3 mRNA expression analysis in GEPIA showing a positive correlation of NFATc1 and Orai3 in pancreatic tissues. (H) Schematic representation of Orai3 promoter with NFATc1 binding site deletions and the truncated Orai3 promoter with all potential sites deleted. (I) Normalized luciferase activity of wild-type Orai3 promoter, Orai3 promoter with NFATc1 binding site deletions, and truncated Orai3 promoter in PANC-1 cells upon NFATc1 overexpression for 48 h. ($N = 3$). ****$P < 0.0001$; **$P = 0.0052$; *$P = 0.0149$. (J) ChIP-qPCR analysis in PANC-1 cells for relative fold enrichment in NFATc1 immunoprecipitated DNA samples showing higher enrichment of Orai3 promoter region compared to IgG mock IP and other NFATc isoforms ($N = 3$). *$P = 0.0147$; *$P = 0.0147$. Data presented are mean ± SEM. For statistical analysis, ordinary one-way ANOVA was performed, followed by Tukey's multiple comparison test for (E, F, I), while one-sample $t$ test was performed for (J) using GraphPad Prism software. Here, NS means non-significant; *$P < 0.05$, **$P < 0.01$, ***$P < 0.001$ and ****$P < 0.0001$. Source data are available online for this figure.

and BxPC-3 (non-metastatic), PANC-1 and AsPC-1 (invasive), and CFPAC-1 and SW990 (metastatic). Initially, we performed NFATc1 overexpression in MiaPaCa-2, PANC-1, and CFPAC-1 cells, which led to a significant upregulation of Orai3 at the mRNA level compared to empty vector control in non-metastatic cells (Fig. 2A), invasive (Fig. 2F), and metastatic cells (Fig. 2K). However, at the protein level, we observed a dichotomy in Orai3 regulation by NFATc1. In non-metastatic cells, NFATc1 overexpression led to an increase in Orai3 at the protein levels (Fig. 2B,C). Surprisingly, NFATc1 overexpression in invasive (Fig. 2G,H) and metastatic cells (Fig. 2L,M) led to a decrease in Orai3 protein levels compared to the control. This suggests that some sort of negative feedback loop is functional in invasive and metastatic cells. It appears that in these cells, NFATc1, apart from transcriptionally regulating Orai3, initiates a protein degradation cascade.

To further corroborate the dichotomous role of NFATc1 on Orai3, we examined Orai3 function upon NFATc1 overexpression in the three cell types. For this, we performed live-cell $Ca^{2+}$ imaging with FURA-2AM dye, a ratio-metric $Ca^{2+}$ indicator. We utilized 2-aminoethoxydiphenyl borate (2APB), a pharmacological agent that can differentiate between functional Orai1 and Orai3 channels. It is widely recognized that 2APB, at concentrations ranging from 30 to 50 µM, inhibits Orai1 channels but activates Orai3 channels (Vashisht et al, 2018; Motiani et al, 2013, 2010; Arora et al, 2021). We utilized the standard thapsigargin-activated SOCE protocol (Tanwar et al, 2022, 2024). The $Ca^{2+}$ imaging studies showed that in non-metastatic cells, NFATc1 overexpression led to an increase in ER $Ca^{2+}$release (Fig. EV1A,B), augmentation in SOCE activity (Fig. EV1A,C), and higher 2APB potentiation (Fig. EV1A,D) compared to empty vector control. This data correlates with the protein data that NFATc1 overexpression upregulates Orai3 at the protein level in non-metastatic cells. As reported earlier (Motiani et al, 2010), 2APB is an efficient tool to observe functional activity of Orai3, we performed calcium imaging in HBSS-Ca buffer with 2APB addition alone. We observed that in non-metastatic cells, NFATc1 overexpression leads to higher 2APB-induced Orai3 potentiation compared to empty vector control (Fig. 2D,E). This data correlates with both the standard thapsigargin-activated SOCE protocol calcium imaging and the Orai3 protein data that NFATc1 overexpression results in augmented Orai3 protein levels in these cells (Fig. 2B,C). However, in invasive and metastatic cells, NFATc1 overexpression results in a decrease in 2APB-induced Orai3 potentiation compared to empty vector control (Fig. 2I,J,N,O). Although the extent of 2APB-induced Orai3 potentiation differs

among the cell lines, the trend of increase and decrease in potentiation corresponds to the changes in Orai3 protein levels in each cell line. Taken together, this functional analysis of Orai3 is in line with the protein data that NFATc1 overexpression down-regulates Orai3 protein expression in invasive and metastatic cells. $Ca^{2+}$ imaging with both standard thapsigargin-activated SOCE protocol and 2APB potentiation replicated the same result. Therefore, for further imaging experiments, we continued with the 2APB potentiation protocol.

## Competitive inhibition of NFAT validates dichotomous regulation of Orai3 in non-metastatic v/s metastatic PC cells

Overexpression of NFATc1 in PC cells indicated a bimodal regulation of Orai3 by NFATc1. To corroborate these findings, we inhibited NFAT activity with VIVIT. VIVIT is a small peptide that inhibits NFAT translocation to the nucleus by competing with NFATc for the calcineurin binding site (Aramburu et al, 1999). VIVIT peptide shows 25 times higher affinity for calcineurin binding and prevents calcineurin-facilitated NFAT dephosphorylation (Aramburu et al, 1999). Surprisingly, transfection of non-metastatic (Fig. 3A), invasive (Fig. 3F), and metastatic cells (Fig. 3K) with VIVIT showed an increase in Orai3 at mRNA compared to empty vector control. One possible explanation for the increase in Orai3 mRNA by inhibiting NFAT in non-metastatic, invasive, and metastatic cells could be that other endogenous transcription factors are activated, and they drive Orai3 transcription in this condition. To test this possibility, we performed luciferase assays with the wild-type Orai3 promoter and the truncated Orai3 promoter with no NFATc1 binding sites. NFAT inhibition via VIVIT transfection led to an increase in luciferase activity in both wild-type and truncated Orai3 promoter (Fig. EV2A). Hence, removal of NFATc1 binding sites had no significant effect on luciferase activity suggesting that apart from NFATc1, other endogenous transcription factors are involved in regulating Orai3 transcription, and inhibition of NFAT dephosphorylation activates other endogenous transcription factors to bind to the Orai3 promoter and upregulate Orai3 transcription.

Excitingly, NFATc1 inhibition by VIVIT resulted in a decrease in Orai3 protein expression in non-metastatic cells (Fig. 3B,C) and an increase in Orai3 protein levels in invasive (Fig. 3G,H) and metastatic cells (Fig. 3L,M). This Orai3 protein data upon NFATc1

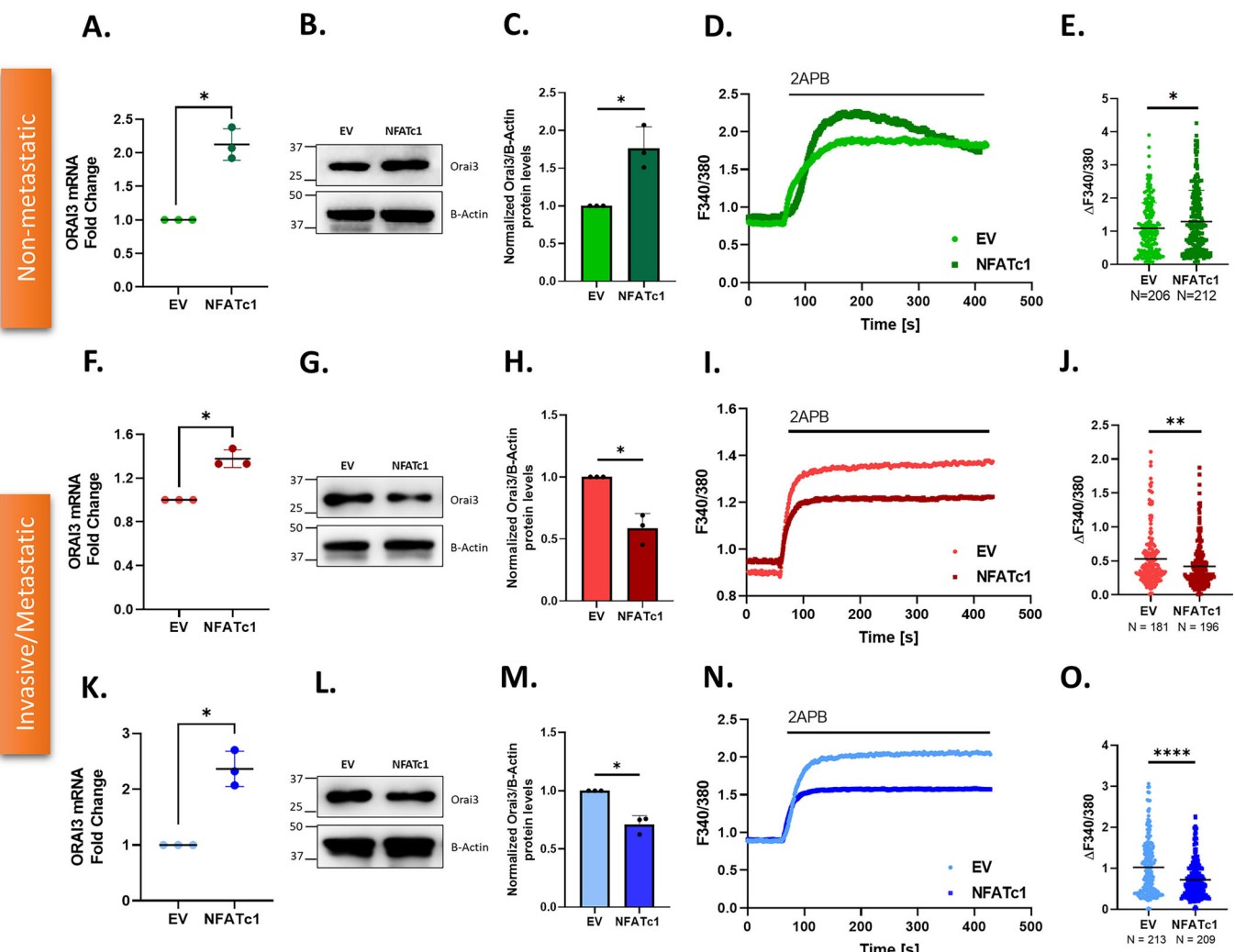

**Figure 2. NFATc1 overexpression dichotomously regulates Orai3 in non-metastatic v/s metastatic PC cells.**

(A) qRT-PCR analysis showing increase in Orai3 mRNA levels upon NFATc1 overexpression in MiaPaCa-2 compared to control ($N = 3$). *$P = 0.0142$. (B) Representative western blots showing an increase in Orai3 protein levels due to NFATc1 overexpression in MiaPaCa-2 cells compared to control. (C) Densitometric quantitation of Orai3 protein levels in NFATc1 overexpressed MiaPaCa-2 compared to control ($N = 3$). *$P = 0.0437$. (D) Representative $Ca^{2+}$ imaging trace of cells transfected with either the control vector pEGFP-C1 plasmid or NFATc1 overexpression plasmid in MiaPaCa-2. (E) Potentiation of Orai3 by 2-APB in NFATc1 overexpressed and empty vector control MiaPaCa-2, where "N" denotes the number of ROIs. *$P = 0.0190$. (F) qRT-PCR analysis showing an increase in Orai3 mRNA expression upon NFATc1 overexpression in PANC-1 compared to control ($N = 3$). *$P = 0.0150$. (G) Representative western blots showing a decrease in Orai3 protein levels due to NFATc1 overexpression in PANC-1 compared to control. (H) Western blot densitometry of Orai3 protein levels in NFATc1 overexpressed PANC-1 cells compared to control ($N = 3$). *$P = 0.0268$. (I) Representative $Ca^{2+}$ imaging trace of cells transfected with either the control pEGFP-C1 plasmid or NFATc1 overexpression plasmid in PANC-1. (J) Potentiation of Orai3 by 2-APB in NFATc1 overexpressed and empty vector control PANC-1, where "N" denotes the number of ROIs. **$P = 0.0037$. (K) qRT-PCR analysis showing an increase in Orai3 mRNA expression upon NFATc1 overexpression in CFPAC-1 compared to control ($N = 3$). *$P = 0.0176$. (L) Representative western blots showing a decrease in Orai3 protein levels due to NFATc1 overexpression in CFPAC-1 compared to control. (M) Western blot densitometry of Orai3 protein in NFATc1 overexpressed CFPAC-1 cells compared to control ($N = 3$). *$P = 0.0219$. (N) Representative $Ca^{2+}$ imaging trace of cells transfected with either the control vector pEGFP-C1 plasmid or NFATc1 overexpression plasmid in CFPAC-1 cells. (O) Potentiation of Orai3 by 2-APB in NFATc1 overexpressed and empty vector control CFPAC-1 where "N" denotes the number of ROIs. ****$P < 0.0001$. Data presented are mean ± SEM. For statistical analysis, one-sample $t$ test was performed for (A, C, F, H, K, M) while unpaired Student's $t$ test was performed for (E, J, O) using GraphPad Prism software. Here, *$P < 0.05$; **$P < 0.01$ and ****$P < 0.0001$. Source data are available online for this figure.

inhibition is in line with NFATc1 overexpression data. Collectively, the dichotomy in the regulation of Orai3 protein levels remains consistent in both NFATc1 gain-of-function and loss-of-function studies. Next, we performed standard thapsigargin-activated SOCE protocol $Ca^{2+}$ imaging in VIVIT-transfected non-metastatic and invasive cells. The $Ca^{2+}$ imaging data revealed that inhibition of NFAT activity caused a decrease in ER $Ca^{2+}$ Release (Fig. EV2B,C),

$Ca^{2+}$ entry (Fig. EV2B,D), and 2APB potentiation (Fig. EV2B,E) in non-metastatic cells. This aligned well with the protein data in non-metastatic cells. Thus, NFAT inhibition decreases Orai3 activity in non-metastatic cells. Using the 2APB-mediated Orai3 potentiation protocol, we observed that inhibition of NFAT activity caused a decrease in 2APB-induced Orai3 potentiation in non-metastatic cells (Fig. 3D,E). While in invasive (Fig. 3I,J) and metastatic cells

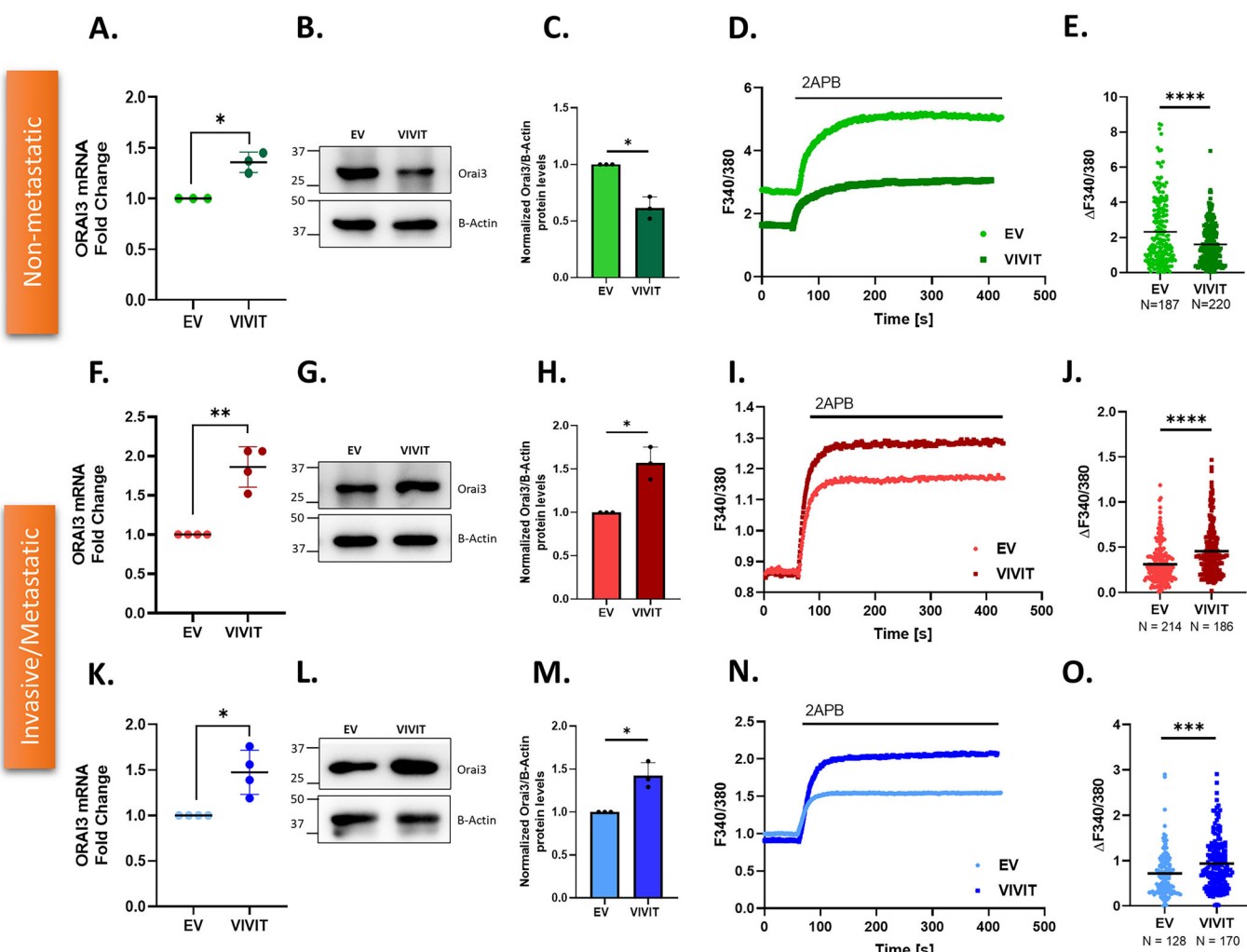

**Figure 3. Competitive inhibition of NFAT validates dichotomous regulation of Orai3 in non-metastatic v/s metastatic PC cells.**

(A) qRT-PCR analysis showing an increase in Orai3 mRNA levels upon VIVIT transfection in MiaPaCa-2 compared to control ($N = 3$). *$P = 0.0255$. (B) Representative western blots showing a decrease in Orai3 protein levels due to VIVIT transfection in MiaPaCa-2 cells compared to control. (C) Densitometric quantitation of Orai3 protein levels in VIVIT-transfected MiaPaCa-2 cells compared to control ($N = 3$). *$P = 0.0205$. (D) Representative Ca$^{2+}$ imaging trace of cells transfected with either the control vector pEGFP-C1 plasmid or VIVIT in MiaPaCa-2. (E) Potentiation of Orai3 by 2-APB in VIVIT-transfected and empty vector control MiaPaCa-2 cells, where "$N$" denotes the number of ROIs. ****$P < 0.0001$. (F) qRT-PCR analysis showing an increase in Orai3 mRNA expression upon VIVIT transfection in PANC-1 compared to control ($N = 4$). **$P = 0.0069$. (G) Representative western blots showing an increase in Orai3 protein levels due to VIVIT transfection in PANC-1 compared to control. (H) Western blot densitometry of Orai3 protein levels upon VIVIT transfection in PANC-1 cells compared to control ($N = 3$). *$P = 0.0341$. (I) Representative Ca$^{2+}$ imaging trace of cells transfected with either the control vector pEGFP-C1 plasmid or VIVIT in PANC-1 cells. (J) Potentiation of Orai3 by 2-APB in VIVIT-transfected and empty vector control PANC-1 cells, where "$N$" denotes the number of ROIs. ****$P < 0.0001$. (K) qRT-PCR analysis showing an increase in Orai3 mRNA expression upon VIVIT transfection in CFPAC-1 compared to control ($N = 4$). *$P = 0.0297$. (L) Representative western blots showing an increase in Orai3 protein levels due to NFATc1 overexpression in CFPAC-1 compared to control. (M) Western Blot densitometry of Orai3 protein levels in VIVIT-transfected CFPAC-1 cells compared to control ($N = 3$). *$P = 0.0417$. (N) Representative Ca$^{2+}$ imaging trace of cells transfected with either control vector pEGFP-C1 plasmid or VIVIT in CFPAC-1 cells. (O) Potentiation of Orai3 by 2-APB in VIVIT-transfected and empty vector control CFPAC-1 cells, where "$N$" denotes the number of ROIs. ***$P = 0.0005$. Data presented are mean ± SEM. For statistical analysis, one-sample $t$ test was performed for (A, C, F, H, K, M) while unpaired Student's $t$ test was performed for (E, J, O) using GraphPad Prism software. Here, * $p < 0.05$; **$P < 0.01$; ***$P < 0.001$ and ****$P < 0.0001$. Source data are available online for this figure.

(Fig. 3N,O), NFAT inhibition caused an increase in 2APB-induced Orai3 potentiation. Taken together, NFATc1 overexpression and inhibition data establish the divergent control of NFATc1 over Orai3 protein expression and activity in non-metastatic v/s invasive and metastatic cells.

Since VIVIT inhibits all NFAT isoforms, we next validated the specificity of NFATc1 in the dichotomous regulation of Orai3. We performed siRNA-mediated NFATc1 knockdown in pancreatic cancer cell lines (MiaPaCa-2, PANC-1, and CFPAC-1). We observed a robust decrease in the NFATc1 mRNA levels in all the cell lines (Fig. EV2F–H). Next, we checked Orai3 mRNA and protein levels upon NFATc1 knockdown in these cells. Similar to VIVIT transfection, we observed an increase in Orai3 mRNA levels upon NFATc1 knockdown in MiaPaCa-2 (Fig. EV2I), PANC-1 (Fig. EV2N), and CFPAC-1 (Fig. EV2S). On the other hand, NFATc1 knockdown decreases Orai3 protein levels and functional

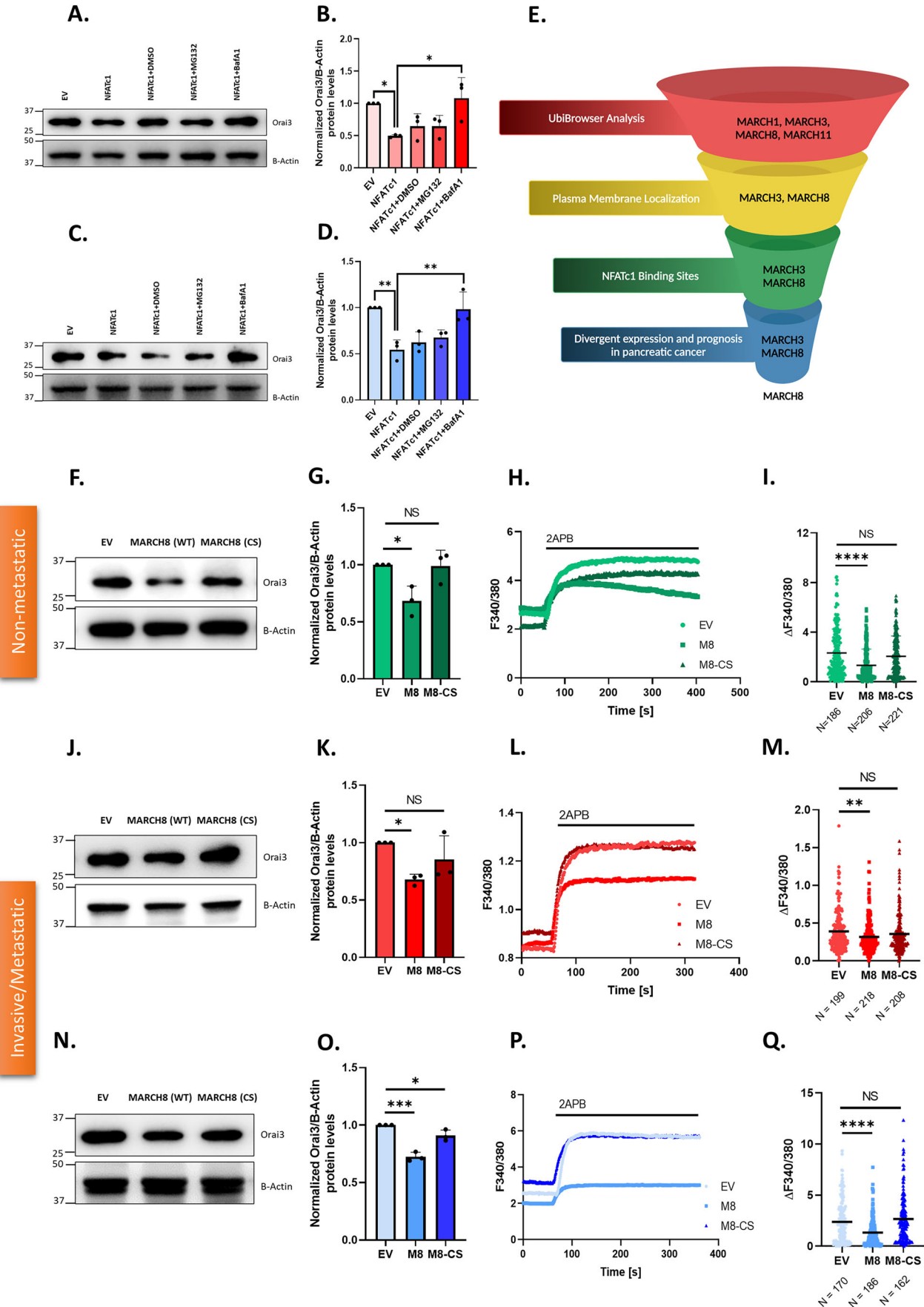

**Figure 4.   MARCH8 E3 ubiquitin ligase downregulates Orai3.**

(A) Representative western blots showing the effect of MG-132 (20 µM) and Bafilomycin A1 (250 nM) on NFATc1-induced Orai3 protein degradation in PANC-1 cells. (B) Densitometric analysis of Orai3 protein levels in NFATc1 overexpressed PANC-1 cells treated with MG-132 (20 µM) or Bafilomycin A1 (250 nM) ($N = 3$). *$P = 0.0437$; *$P = 0.0185$. (C) Representative western blots showing the effect of MG-132 (20 µM) and Bafilomycin A1 (250 nM) on NFATc1-induced Orai3 protein degradation in CFPAC-1 cells. (D) Densitometric analysis of Orai3 protein levels in NFATc1 overexpressed CFPAC-1 cells treated with MG-132 (20 µM) or Bafilomycin A1 (250 nM) ($N = 3$). **$P = 0.0044$; **$P = 0.0058$. (E) Schematic of the pipeline used for bioinformatics-based identification of potential E3 ubiquitin ligases targeting Orai3. (F) Representative western blot showing that overexpression of MARCH8 (WT) and not mutant-MARCH8 (CS) decreases Orai3 levels in MiaPaCa-2 cells compared to the control. (G) Western blot densitometry of Orai3 levels in MARCH8 (WT) and MARCH8 (CS) overexpressed MiaPaCa-2 cells ($N = 3$). *$P = 0.0219$. (H) Representative Ca$^{2+}$ imaging trace of cells transfected with either control vector pEGFP-N1 or MARCH8 (WT) or MARCH8 (CS) plasmids in MiaPaCa-2 cells. (I) Potentiation of Orai3 by 2-APB in cells transfected with either empty vector control or MARCH8 (WT) or MARCH8 (CS) overexpressed MiaPaCa-2 cells, where "N" denotes the number of ROIs. ****$P < 0.0001$. (J) Representative western blot showing that overexpression of MARCH8 (WT) and not mutant-MARCH8 (CS) decreases Orai3 levels in PANC-1 cells compared to the control. (K) Western Blot densitometry of Orai3 levels in MARCH8 (WT) and MARCH8 (CS) overexpressed PANC-1 cells ($N = 3$). *$P = 0.0321$. (L) Representative Ca$^{2+}$ imaging trace of cells transfected with either control vector pEGFP-N1 or MARCH8 (WT) or MARCH8 (CS) plasmids in PANC-1 cells. (M) Potentiation of Orai3 by 2-APB in cells transfected with either empty vector control or MARCH8 (WT) or MARCH8 (CS) overexpressed PANC-1 cells, where "N" denotes the number of ROIs. **$P = 0.0072$. (N) Representative western blot showing that overexpression of MARCH8 (WT) and not mutant-MARCH8 (CS) decreases Orai3 levels in CFPAC-1 cells compared to the control. (O) Western Blot densitometry of Orai3 levels in MARCH8 (WT) and MARCH8 (CS) overexpressed CFPAC-1 cells ($N = 3$). ***$P = 0.0002$; *$P = 0.0386$. (P) Representative Ca$^{2+}$ imaging trace of cells transfected with either control vector pEGFP-N1 or MARCH8 (WT) or MARCH8 (CS) plasmids in CFPAC-1 cells. (Q) Potentiation of Orai3 by 2-APB in cells transfected with either empty vector control or MARCH8 (WT) or MARCH8 (CS) overexpressed CFPAC-1 cells, where "N" denotes the number of ROIs. ****$P < 0.0001$. Data presented are mean ± SEM. For statistical analysis, ordinary one-way ANOVA was performed, followed by Tukey's multiple comparison test for (B, D, G, I, K, M, O, Q) using GraphPad Prism software. Here, NS means non-significant; *$P < 0.05$; **$P < 0.01$; and ****$P < 0.0001$. Source data are available online for this figure.

activity in MiaPaCa-2 (Fig. EV2J–M); while Orai3 protein levels and functional activity increase upon NFATc1 silencing in PANC-1 (Fig. EV2O–R) and CFPAC-1 (Fig. EV2T–W). Therefore, both siRNA-mediated NFATc1 knockdown and competitive inhibition of NFAT display the dichotomous regulation of Orai3 in non-metastatic v/s metastatic pancreatic cancer cells. Since NFATc1 siRNA and VIVIT gave the same results, we proceeded with VIVIT for further NFATc1 loss-of-function studies.

The dual nature of Orai3 regulation by NFATc1 depends on the non-metastatic v/s metastatic nature of pancreatic cancer cell lines. To further substantiate this hypothesis, we selected three more pancreatic cancer cell lines BxPC-3 (non-metastatic), AsPC-1 (invasive), and SW1990 (metastatic) for performing NFATc1 loss and gain-of-function experiments. Excitingly, we observed the same results in these additional cell lines. Both overexpression of NFATc1 and VIVIT mediated NFAT inhibition led to an increase in Orai3 mRNA levels compared to the empty vector control in BxPC-3 (non-metastatic) (Fig. EV3A,D), AsPC-1 (invasive) (Fig. EV3G,J) and SW1990 (metastatic) (Fig. EV3M,P). In BXPC-3, NFATc1 overexpression increased Orai3 protein levels (Fig. EV3B,C) while VIVIT transfection decreased Orai3 protein levels compared to empty vector control (Fig. EV3E,F), which is similar to results observed in MiaPaCa-2 (non-metastatic). In AsPC-1, NFATc1 overexpression decreased Orai3 protein levels (Fig. EV3H,I) while VIVIT transfection increased Orai3 protein levels compared to empty vector control (Fig. EV3K,L), which is similar to PANC-1 (invasive). In SW19090, NFATc1 overexpression decreased Orai3 protein levels (Fig. EV3N,O) while VIVIT transfection increased Orai3 protein levels compared to empty vector control (Fig. EV3Q,R), which is similar to CFPAC-1 (metastatic). Taken together, our NFATc1 gain and loss-of-function studies in six pancreatic cancer cell lines confirm the dichotomous regulation of Orai3 by NFATc1.

## MARCH8 E3 ubiquitin ligase degrades Orai3

NFATc1 overexpression and inhibition studies in invasive and metastatic PC cell lines suggest involvement of Orai3 protein

degradation in regulating Orai3 protein expression and activity. However, to the best of our knowledge, the Orai3 protein degradation machinery is unknown. Therefore, we started investigating Orai3 protein degradation mechanism. First of all, to determine whether NFATc1-induced Orai3 protein degradation occurs via the proteasomal or lysosomal pathway, we treated the NFATc1-overexpressing cells with either MG132 or bafilomycin A1. MG132 inhibits the proteasomal degradation pathway (Lee and Goldberg, 1998), while bafilomycin A1 blocks the lysosomal degradation cascade (Tapper and Sundler, 1995). We observed that treatment of bafilomycin A1 for 6 h in NFATc1 overexpressed invasive (Fig. 4A,B) and metastatic cells (Fig. 4C,D) inhibited Orai3 degradation by NFATc1 compared to the vehicle control. In contrast, treatment of MG132 for 6 h in NFATc1 overexpressed invasive (Fig. 4A,B) and metastatic cells (Fig. 4C,D) did not inhibit Orai3 degradation by NFATc1. This indicates that NFATc1 induces Orai3 degradation via the lysosomal ubiquitination pathway in invasive and metastatic PC cells.

As ubiquitination of proteins is mainly carried out by E3 ubiquitin ligases, we next started identifying the putative E3 ubiquitin ligases that target Orai3. We utilized a well-established bioinformatics tool (UbiBrowser) (Wang et al, 2022a) to predict potential Orai3 E3 ubiquitin ligases. We only got four putative hits through this analysis. Interestingly, all four predicted E3 ubiquitin ligases belong to the MARCH (Membrane-Associated RING-CH-type) family of membrane-bound E3 ubiquitin ligases, i.e., MARCH1, MARCH3, MARCH8, and MARCH11. Since Orai3 is a plasma membrane Ca$^{2+}$ channel, its ubiquitination by membrane-bound E3 ubiquitin ligases appears logical. MARCH family are a subset of RING domain-containing E3 ubiquitin ligases, which have specific transmembrane domains allowing them to be tethered to the plasma membrane or membranes of other organelles. We next evaluated the subcellular localization of the four predicted E3 ubiquitin ligases. Intriguingly, only MARCH3 and 8 have been reported to be localized at the plasma membrane (Lin et al, 2019) while MARCH1 and MARCH11 are localized to endo-lysosomes. In light of this analysis, we then focused our attention on MARCH3 and MARCH8. Since we observed Orai3 lysosomal degradation

downstream of NFATc1, we examined the presence of potential NFATc1 binding sites on the MARCH3 and MARCH8 promoters. Interestingly, we found that the promoters of both MARCH3 and MARCH8 have NFATc1 binding sites (Fig. EV4A). As Orai3 is overexpressed in pancreatic cancer tissue and is associated with poor prognosis (Arora et al, 2021), we next analyzed the expression of MARCH3 and MARCH8 in pancreatic cancer samples (Fig. EV4B,C). Further, we examined the association of MARCH3 and MARCH8 with pancreatic cancer patient survival using OncoLnc database (Anaya, 2016). Interestingly, we found that less than 15% of patients with high MARCH3 levels survived for >5.5 years, whereas 30% of patients with high MARCH8 levels survived for >7.5 years. Hence, high MARCH8 expression in pancreatic cancer patients provided a greater survival advantage compared to high MARCH3 levels. This suggests that in comparison to MARCH3, MARCH8 is associated with a favorable prognosis in pancreatic cancer wherein higher MARCH8 expression correlated with better patient survival (Fig. EV4D,E). In addition, MARCH8 is reported to have anti-oncogenic effects in breast and lung cancer (Chen et al, 2021; Qian et al, 2021). Moreover, MARCH8 promotes lysosomal degradation of target proteins (Chen et al, 2021). Taken together, this robust and extensive bioinformatics analysis suggested that MARCH8 could be the E3 ubiquitin ligase that bridges NFATc1 and Orai3 protein degradation (Fig. 4E). Therefore, we decided to examine the role of MARCH8 in Orai3 degradation downstream of NFATc1.

To study the effect of MARCH8 on Orai3 protein levels, we overexpressed either wild-type MARCH8 (WT) or inactive MARCH8 (CS) mutant in MiaPaCa-2 (non-metastatic), PANC-1 (invasive), and CFPAC-1 (metastatic) cells. The MARCH8 (CS) mutant is inactive as all the cysteine residues (Cys80, 83, 97, 99, 110, 123, and 126) are substituted with serine residues (Fujita et al, 2013). The overexpression of MARCH8 leads to a decrease in Orai3 protein levels in non-metastatic (Fig. 4F,G), invasive (Fig. 4J,K), and metastatic cells (Fig. 4N,O) while overexpression of MARCH8 (CS) mutant did not alter Orai3 protein levels in these cells. Next, Ca$^{2+}$ imaging upon MARCH8 (WT) and MARCH8 (CS) overexpression in MiaPaCa-2, PANC-1, and CFPAC-1 cells showed a decrease in 2APB-induced Orai3 potentiation upon MARCH8 (WT) overexpression but no significant change in 2APB-induced Orai3 potentiation upon MARCH8 (CS) overexpression in non-metastatic (Fig. 4H,I), invasive (Fig. 4L,M), and metastatic cells (Fig. 4P,Q). This Ca$^{2+}$ imaging analysis further substantiated the Orai3 protein data. Taken together, it demonstrates that overexpression of functional MARCH8 leads to a decrease in Orai3 expression and consequently its activity.

To further validate the role of MARCH8 in regulating Orai3 expression, we performed siRNA-mediated knockdown of MARCH8. We validated the knockdown of MARCH8 upon siMARCH8 transfections at mRNA and protein levels in MiaPaCa-2 (Fig. EV5A,B) PANC-1 (Fig. EV5C,D), and CFPAC-1 (Fig. EV5E,F). We next looked into Orai3 protein levels and observed that MARCH8 silencing leads to an increase in Orai3 protein levels in MiaPaCa-2 (Fig. 5A,B), PANC-1 (Fig. 5E,F), and CFPAC-1 (Fig. 5I,J) cell lines compared to empty vector control. Subsequently, Ca$^{2+}$ imaging in siRNA-mediated MARCH8 knockdown cells showed an increase in 2APB-induced Orai3 potentiation in MiaPaCa-2 (Fig. 5C,D), PANC-1 (Fig. 5G,H), and CFPAC-1 (Fig. 5K,L) cells compared to siNT control cells. Hence, the Orai3 protein levels and activity is exactly opposite to the MARCH8 expression. Collectively,

MARCH8 gain-of-function, overexpression of inactive MARCH8 and MARCH8 loss-of-function studies reveal that MARCH8 E3 ubiquitin ligase is a critical regulator of Orai3 protein levels.

## MARCH8 physically interacts with Orai3 and induces its ubiquitination

MARCH8 E3 ubiquitin ligase ubiquitinates its target protein at its lysine residues, which leads to target degradation. Hence, we performed ubiquitination assays to check the status of Orai3 ubiquitination by immunoprecipating Orai3 using an antibody against Orai3 followed by immunoblotting with ubiquitin antibody. As shown in Fig. 6A, the immunoprecipitated Orai3 shows ubiquitination. In order to identify the lysine residues ubiquitinated in Orai3, we used Phosphositeplus tool (Hornbeck et al, 2012) to predict the potential residues and narrowed to three lysine residues, i.e., K2 (at N-terminal), K274 and K279 (at C-terminal) (Fig. 6B). We independently mutated these lysine residues to arginine in pcDNA3.1 Orai3 vector and performed ubiquitination assays to decode, which lysine residues are important for Orai3 ubiquitination. The ubiquitination assays showed that only Orai3 K2R mutation severely affected ubiquitination status as compared to Orai3 WT, K274R, and K279R mutations (Fig. 6C). Hence, K2 residue at N-terminal of Orai3 is critical for its ubiquitination.

Next, we investigated the precise role of MARCH8 in Orai3 ubiquitination. We overexpressed MARCH8 in bafilomycin A1-treated PANC-1 cells and performed Orai3 immunoprecipitation. The western blot analysis shows that MARCH8 overexpression increases Orai3 ubiquitination compared to the empty vector control (Fig. 6D). This data suggests that MARCH8 E3 ubiquitin ligase ubiquitinates Orai3. To study the interaction of MARCH8 and Orai3, we carried out co-immunoprecipitation assays and observed that MARCH8 and Orai3 interact with each other (Fig. 6E). To investigate the interaction domains of MARCH8 in Orai3, we performed bioinformatic analysis which suggested that MARCH8 can interact with Orai3 in the intracellular loop present between the 1st and 2nd transmembrane domains at the LMVXXXL (AA113-120) motif. Further, MARCH family members are also reported to interact with proteins containing a GXXXG (AA235-239) motif, which is present in 3$^{rd}$ loop region of Orai3 (Wang et al, 2022a) (Fig. 6B). Hence, we created LMVXXXL and GXXXG deletion mutants of Orai3 in pcDNA3.1 Orai3 vector and performed co-immunoprecipitation to check the interaction of MARCH8 and Orai3 deletion mutants. Our Co-immunoprecipitation experiments revealed that Orai3 LMVXXXL deletion mutant does not interact with MARCH8 while Orai3 GXXXG deletion mutant interacts with MARCH8 similar to wild-type Orai3 (Fig. 6F). Hence, MARCH8 physically interacts with Orai3 at LMVXXXL domain. Further, we performed confocal microscopy-based immunofluorescence assays and found that Orai3 and MARCH8 colocalize in the plasma membrane (Fig. 6G–I). Collectively, biochemical and microscopy data demonstrate that MARCH8 physically interacts with Orai3 at the plasma membrane and induces Orai3 ubiquitination.

## NFATc1 oppositely regulates MARCH8 expression in non-metastatic v/s invasive and metastatic PC cells

Our investigation into the mechanisms behind Orai3 protein degradation began with the observation that NFATc1 promotes

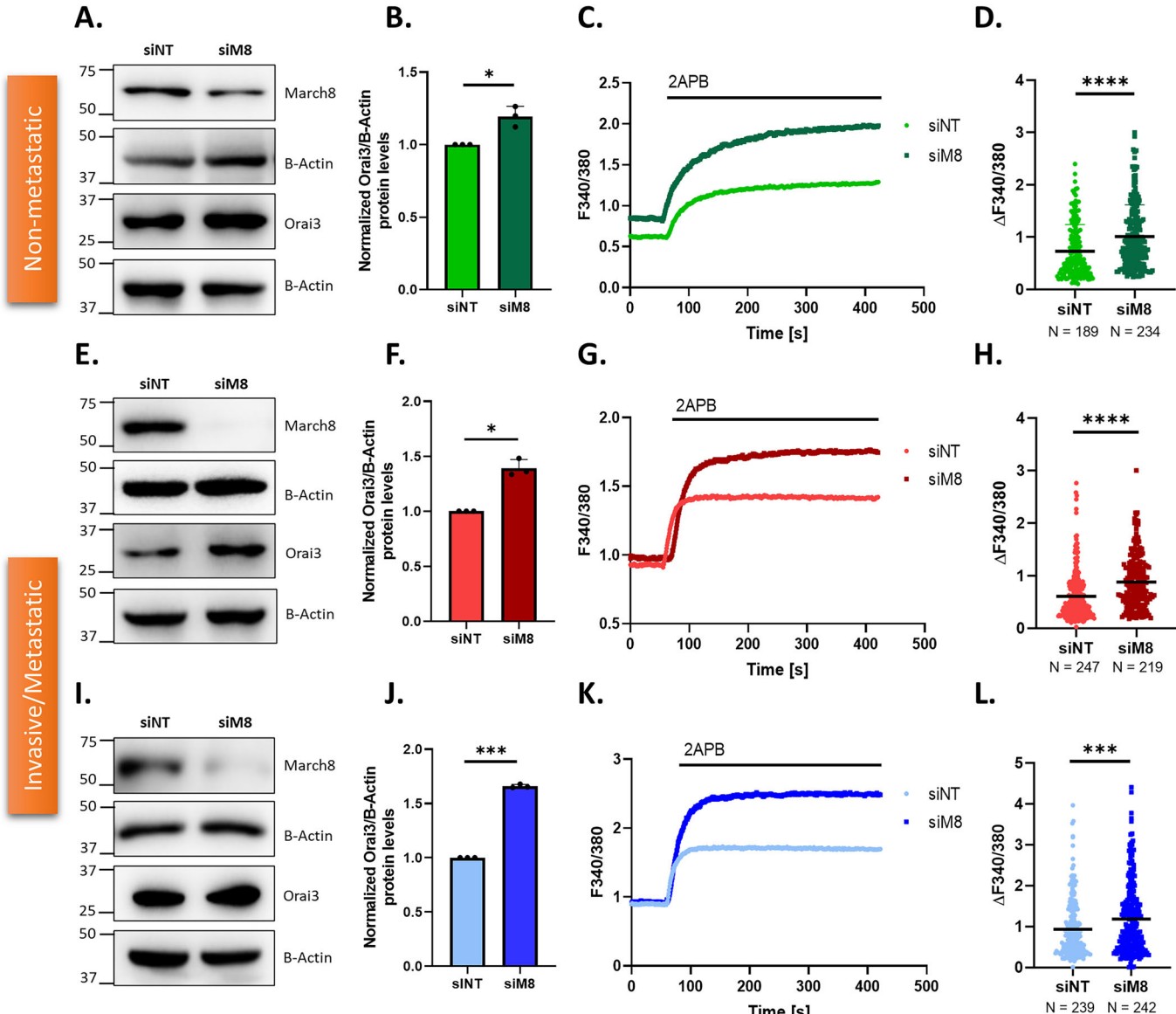

**Figure 5. MARCH8 knockdown upregulates Orai3.**

(A) Representative western blots showing an increase in Orai3 protein levels upon siRNA-mediated MARCH8 knockdown in MiaPaCa-2 cells compared to the control. (B) Western Blot densitometry of Orai3 levels in siNT and siMARCH8-transfected MiaPaCa-2 cells ($N = 3$). *$P = 0.0409$. (C) Representative $Ca^{2+}$ imaging trace of siNT and siMARCH8-transfected MiaPaCa-2 cells. (D). Potentiation of Orai3 by 2-APB in siNT and siMARCH8-transfected MiaPaCa-2 cells, where "$N$" denotes the number of ROIs. ****$P < 0.0001$. (E) Representative western blots showing an increase in Orai3 protein levels upon siRNA-mediated MARCH8 knockdown in PANC-1 cells compared to the control. (F) Western blot densitometry of Orai3 levels in siNT and siMARCH8-transfected PANC-1 cells ($N = 3$). *$P = 0.0128$. (G) Representative $Ca^{2+}$ imaging trace of siNT and siMARCH8-transfected PANC-1 cells. (H) Potentiation of Orai3 by 2-APB in siNT and siMARCH8-transfected PANC-1 cells, where "$N$" denotes the number of ROIs. ****$P < 0.0001$. (I) Representative western blots showing an increase in Orai3 protein levels upon siRNA-mediated MARCH8 knockdown in CFPAC-1 cells compared to the control. (J) Western blot densitometry of Orai3 levels in siNT and siMARCH8-transfected CFPAC-1 cells ($N = 3$). ***$P = 0.0002$. (K) Representative $Ca^{2+}$ imaging trace of siNT and siMARCH8-transfected CFPAC-1 cells. (L) Potentiation of Orai3 by 2-APB in siNT and siMARCH8-transfected CFPAC-1 cells, where "$N$" denotes the number of ROIs. ***$P = 0.0005$. Data presented are mean ± SEM. For statistical analysis, one-sample $t$ test was performed for (B, F, J) while unpaired Student's $t$ test was performed for (D, H, L) using GraphPad Prism software. Here, *$P < 0.05$; ***$P < 0.001$; and ****$P < 0.0001$. Source data are available online for this figure.

Orai3 degradation in invasive and metastatic pancreatic cancer cells. After identifying that the MARCH8 E3 ubiquitin ligase ubiquitinates Orai3, leading to its lysosomal degradation, one key question still remains: Does NFATc1 regulate MARCH8 levels, thereby driving Orai3 degradation in PC cells? To answer this question, we again utilized the bioinformatic tools to examine

human MARCH8 promoter. The bioinformatic analysis with three different algorithms and TF position weight matrix combinations showed one NFATc1 binding site on MARCH8 promoter, which is present at -256bp before the transcription start site (Fig. 7A). Further, the multi-species alignment of the MARCH8 promoter highlighted that this binding site is conserved across multiple

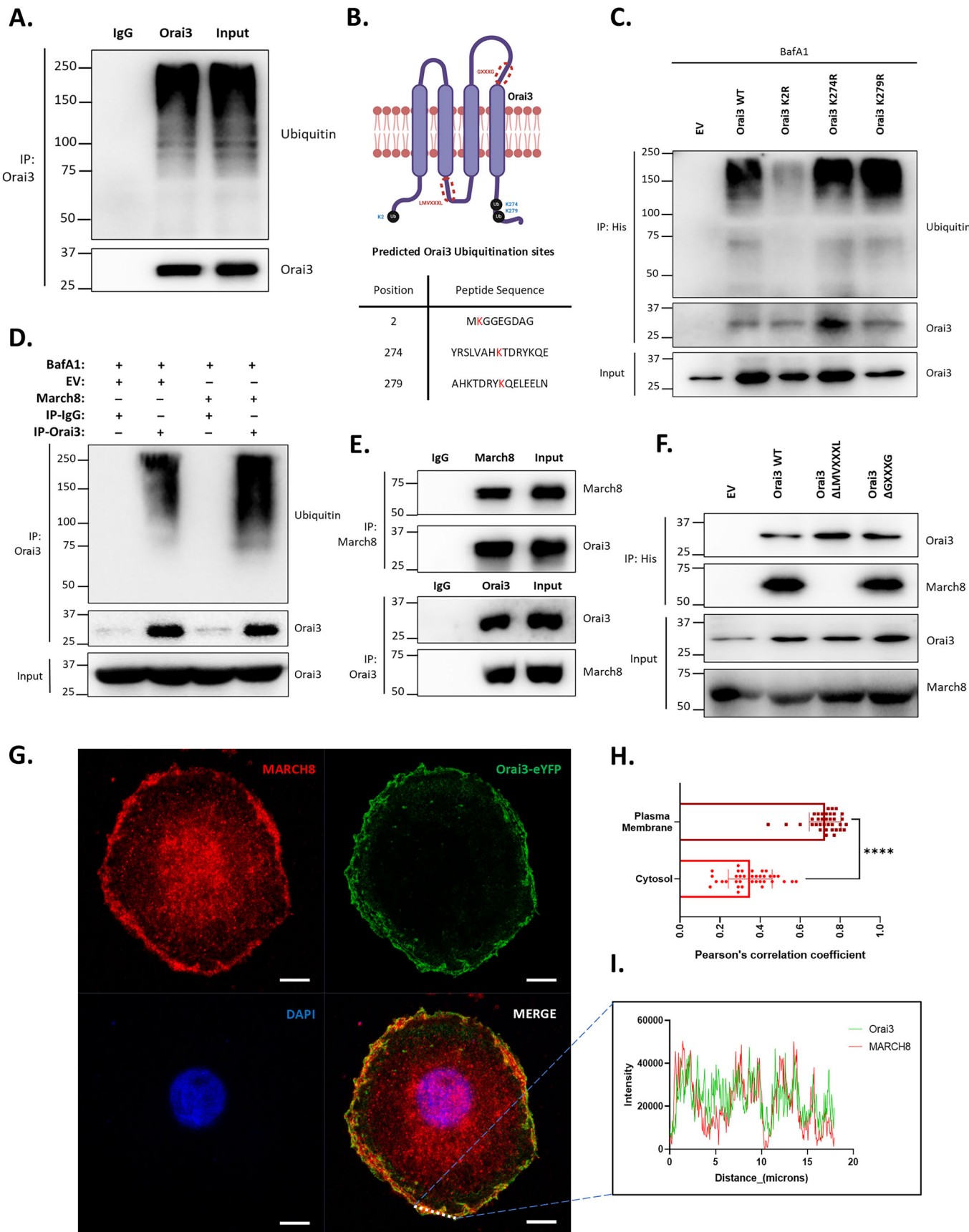

**Figure 6. MARCH8 physically interacts with Orai3 and induces ubiquitination.**

(A) Ubiquitination assay showing ubiquitination of immunoprecipitated Orai3 in bafilomycin A1 (250 nM) treated PANC-1 cells. (B) Schematic representation of Orai3 showing three potential lysine residues for ubiquitination (highlighted in blue) and two predicted MARCH8-interacting domains (highlighted in red circles). (C) Ubiquitination assay with Orai3 lysine residue mutants overexpressed in bafilomycin A1 (250 nM) treated PANC-1 cells. (D) Ubiquitination assay showing that overexpression of MARCH8 in bafilomycin (250 nM) treated PANC-1 cells increases ubiquitin levels in immunoprecipitated Orai3 compared to the empty vector control. The input blot shows endogenous Orai3 protein in the PANC-1 cells used for co-immunoprecipitation. (E) Co-immunoprecipitation and reverse co-immunoprecipitation assays in PANC-1 cells showing interaction of MARCH8 and Orai3. The co-immunoprecipitation assays were performed at least in three independent experiments. (F) Co-immunoprecipitation assay in PANC-1 cells overexpressed with Orai3 deletion mutants to precisely identify the MARCH8-interacting domain in Orai3. The co-immunoprecipitation assays were performed at least in three independent experiments. (G) Immunofluorescence assay for colocalization of MARCH8-Alexa Fluor 568 and Orai3-eYFP in PANC-1 cells. Scale: 10 μm. (H) Pearson's correlation coefficient of MARCH8-Alexa Fluor 568 and Orai3-eYFP in the cytosolic region and plasma membrane in PANC-1 cells ($n = 36$). ****$P < 0.0001$. (I) Profile intensity plot of MARCH8-Alexa Fluor 568 and Orai3-eYFP signals in immunofluorescence assay. Data presented are mean ± SEM. For statistical analysis, unpaired Student's $t$ test was performed for panel H using GraphPad Prism software. Here, ****$P < 0.0001$. Source data are available online for this figure.

primates species (Fig. 7A). In addition, expression analysis of NFATc1 and MARCH8 using the database "GEPIA" showed a positive correlation between NFATc1 and MARCH8 mRNA expression in pancreatic tissue (Fig. 7B). To validate the bioinformatic findings, we conducted dual-luciferase assays in which the human MARCH8 promoter was cloned into a luciferase reporter vector. These assays were performed in PANC-1 cells with NFATc1 overexpression. We observed an increase in relative luciferase activity upon NFATc1 overexpression in comparison to the empty vector control (Fig. 7C). We next carried out Chromatin Immunoprecipitation (ChIP) assays to examine NFATc1 binding on the MARCH8 promoter. We overexpressed either EGFP-C1-NFATc1 or the empty vector control (pEGFP-C1) for ChIP assays. The cross-linked and sonicated DNA was immunoprecipitated and amplified using human MARCH8 promoter-specific primers. ChIP-qPCR analysis revealed a significant enrichment of the MARCH8 promoter with a potential NFATc1 binding site compared to mock IP samples (Fig. 7D). Collectively, the findings from the ChIP and dual-luciferase assays clearly demonstrate that NFATc1 binds to the MARCH8 promoter.

To delineate the role of NFATc1 in regulating MARCH8 in PC cells, we performed NFATc1 overexpression and VIVIT transfection studies in MiaPaCa-2, PANC-1, and CFPAC-1 cell lines. qRT-PCR analysis showed that overexpression of NFATc1 in MiaPaCa-2 (non-metastatic) decreased MARCH8 mRNA levels compared to the control (Fig. 7E). While NFATc1 overexpression increased MARCH8 mRNA levels in invasive PANC-1 (Fig. 7K) and metastatic CFPAC-1 (Fig. 7Q). Likewise, western blot analysis showed the same trend; NFATc1 overexpression decreases MARCH8 protein levels in MiaPaCa-2 (Fig. 7F,G) while it increased MARCH8 protein expression in PANC-1 (Fig. 7L,M) and CFPAC-1 cells (Fig. 7R, S). Next, we performed qRT-PCR analysis upon VIVIT transfection in MiaPaCa-2 and observed an increase in MARCH8 mRNA levels compared to the control (Fig. 7H). However, VIVIT transfection decreased MARCH8 mRNA levels in PANC-1 (Fig. 7N) and CFPAC-1 (Fig. 7T). Similarly, western blot analysis demonstrated the same trend, i.e., VIVIT transfection increases MARCH8 protein levels in MiaPaCa-2 (Fig. 7I, J) while it decreases MARCH8 protein levels in PANC-1 (Fig. 7O,P) and CFPAC-1 cells (Fig. 7U,V). Taken together, these data establish NFATc1 as a repressor of MARCH8 transcription in non-metastatic cells and as an activator in invasive and metastatic PC cells.

## Epigenetic landscape of MARCH8 promoter dictates NFATc1-driven Orai3 regulation in non-metastatic v/s metastatic PC cells

If a transcription factor regulates its target gene differently across distinct cells, it could be because of differences in the epigenetic modifications in the promoter region of the target gene (Gibney and Nolan, 2010). One of the most common epigenetic modifications is DNA methylation, and changes in the methylation profile, especially in the promoter region, lead to differences in gene expression (Lakshminarasimhan and Liang, 2016). Therefore, we investigated the potential contribution of DNA methylation in NFATc1's dichotomous role in regulating MARCH8 transcription. To check whether the MARCH8 gene promoter is methylated, we used the DNMIVD database (Ding et al, 2020) and found that the MARCH8 promoter is hypermethylated in pancreatic tumor samples compared to normal pancreatic samples (Fig. 8A). Consequently, the expression of MARCH8 is less in pancreatic tumor samples compared to normal pancreatic samples (Fig. 8B). Hence, the expression profile of MARCH8 and the methylation status of the MARCH8 promoter are inversely correlated (Fig. 8C). Further, we looked for CpG islands in MARCH8 promoter using Methprimer (Li and Dahiya, 2002) and UCSC genome browser (Karolchik et al, 2009) and found one CpG island extending from −108 bp to +483 bp from the transcription start site (TSS) (Fig. 8D). Collectively, bioinformatic analysis suggests that the MARCH8 promoter is hypermethylated in pancreatic cancer and regulates its expression.

To corroborate the lead from the bioinformatic analysis, we treated MiaPaCa-2 (non-metastatic) cells with DNA methylation inhibitor, decitabine. Decitabine (5-Aza-2′-Deoxycytidine) is a deoxycytidine analog that integrates into DNA and inhibits DNA methyltransferase activity, thereby preventing DNA methylation (Jones and Taylor, 1980). qRT-PCR analysis shows that decitabine treatment in MiaPaCa-2 for 72 h significantly increases MARCH8 mRNA levels compared to the vehicle control (Fig. 8E). We next examined the methylation status of the MARCH8 promoter in non-metastatic, invasive, and metastatic cells. We performed MeDIP (Methyl DNA immunoprecipitation) using antibodies against 5-methylcytosine. We observed that the methylated MARCH8 promoter was highly enriched in MiaPaCa-2 compared to PANC-1 and CFPAC-1, suggesting that the MARCH8 promoter is hypermethylated in non-metastatic cells compared to invasive

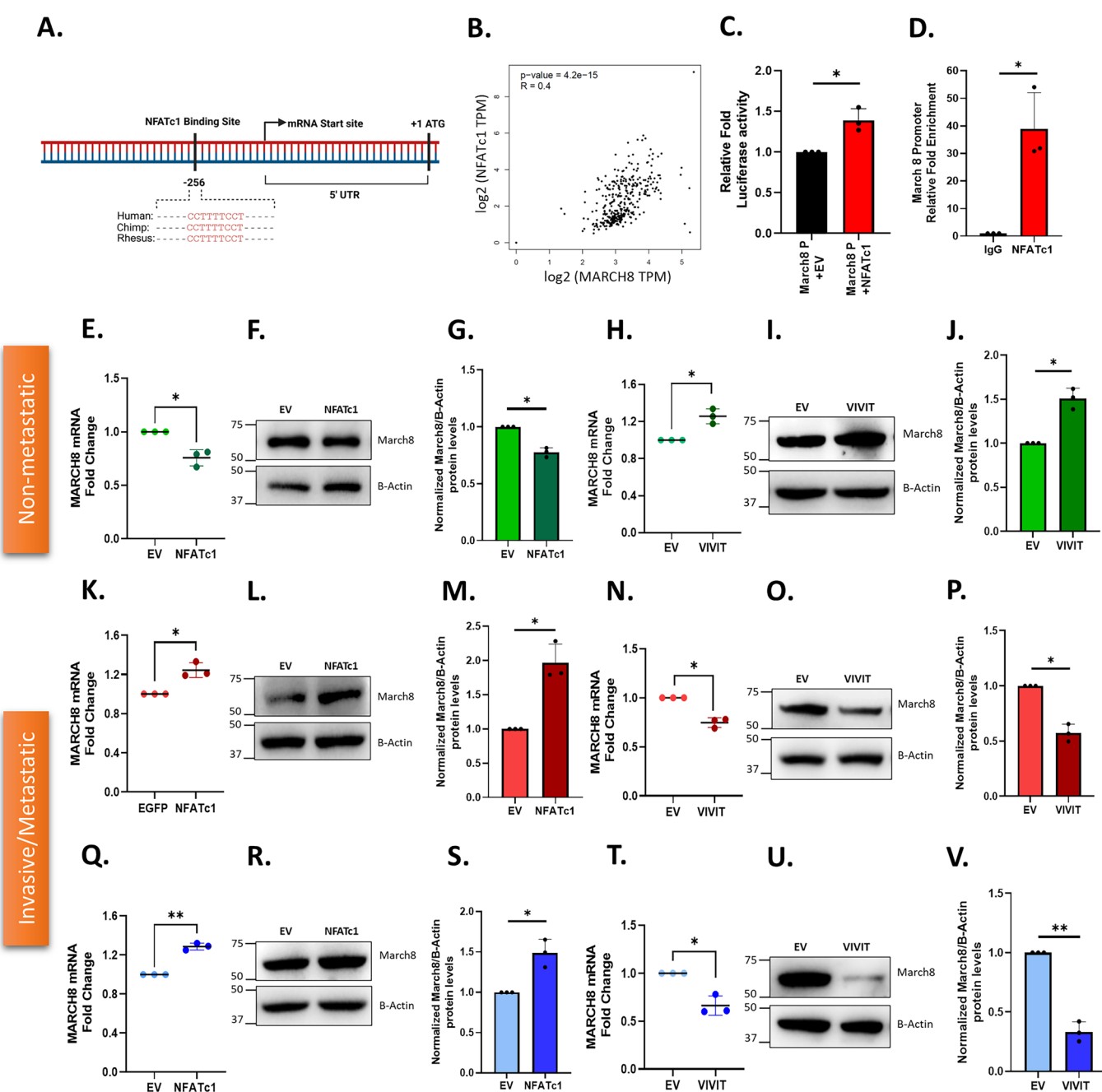

and metastatic cells (Fig. 8F). Finally, to investigate whether DNA methylation prevents NFATc1 from regulating MARCH8 in non-metastatic cells, we overexpressed NFATc1 in decitabine-treated MiaPaCa-2 cells. We found that MARCH8 protein levels increased compared to both vehicle control and empty vector control plus decitabine-treated cells (Fig. 8G,H). Consequently, due to an increase in MARCH8 levels, Orai3 protein levels decreased in NFATc1 overexpressed decitabine-treated cells compared to both vehicle control and empty vector control plus decitabine-treated cells (Fig. 8G,I). Taken together, our data demonstrates that the MARCH8 promoter is hypermethylated in non-metastatic cells and thereby NFATc1 does not upregulate MARCH8 in these cells. This

in turn prevents NFATc1-induced Orai3 degradation, which is observed in invasive and metastatic cells.

## MARCH8 E3 ubiquitin ligase abrogates pancreatic cancer metastasis

This study has identified MARCH8 as a critical regulator of Orai3, which we had earlier reported as a driver of PC metastasis (Arora et al, 2021) Although MARCH8 E3 ubiquitin ligase has been studied in some cancer types (Singh et al, 2017; Fan et al, 2017; Wang et al, 2022b; Chen et al, 2021), its role in PC remains largely unappreciated. Therefore, we examined the relationship between

**Figure 7. NFATc1 differently regulates MARCH8 expression in non-metastatic v/s metastatic PC cells.**

(A) Identification of putative NFATc1 binding sites on the human MARCH8 promoter using the EPD-Search Motif Tool at *P* vauel cut-off of 0.01 and bioinformatic characterization of conserved NFATc1 binding sites on the cross-species alignment of the MARCH8 promoter using the ContraV3 transcription factor binding analysis tool. (B) NFATc1 and MARCH8 mRNA expression analysis in GEPIA shows a positive correlation of NFATc1 and MARCH8 in pancreatic tissues. (C) Normalized luciferase activity of MARCH8 promoter in PANC-1 cells upon NFATc1 overexpression for 48 h ($N = 3$). *$P = 0.0421$. (D) ChIP-qPCR analysis for relative fold enrichment in NFATc1 immunoprecipitated DNA samples showing robust enrichment of MARCH8 promoter region compared to IgG mock IP ($N = 3$). *$P = 0.0377$. (E) qRT-PCR analysis showing a decrease in MARCH8 mRNA levels upon NFATc1 overexpression in MiaPaCa-2 compared to control ($N = 3$). *$P = 0.0308$. (F) Representative western blot showing a decrease in MARCH8 protein levels due to NFATc1 overexpression in MiaPaCa-2 cells compared to control. (G) Densitometric quantitation of MARCH8 protein levels in NFATc1 overexpressed MiaPaCa-2 compared to control ($N = 3$). *$P = 0.0106$. (H) qRT-PCR analysis showing an increase in MARCH8 mRNA levels upon VIVIT transfection in MiaPaCa-2 compared to control ($N = 3$). *$P = 0.0314$. (I) Representative western blots showing an increase in MARCH8 protein levels due to VIVIT transfection in MiaPaCa-2 cells compared to control. (J) Densitometric quantitation of MARCH8 protein levels in VIVIT-transfected MiaPaCa-2 compared to control ($N = 3$). *$P = 0.0169$. (K) qRT-PCR analysis showing an increase in MARCH8 mRNA levels upon NFATc1 overexpression in PANC-1 compared to control ($N = 3$). *$P = 0.0308$. (L) Representative western blots showing an increase in MARCH8 protein levels due to NFATc1 overexpression in PANC-1 cells compared to control. (M) Densitometric quantitation of MARCH8 protein levels in NFATc1 overexpressed PANC-1 compared to control ($N = 3$). *$P = 0.0265$. (N) qRT-PCR analysis showing a decrease in MARCH8 mRNA levels upon VIVIT transfection in PANC-1 compared to control ($N = 3$). *$P = 0.0124$. (O) Representative western blots showing a decrease in MARCH8 protein levels due to VIVIT transfection in PANC-1 cells compared to control. (P) Densitometric quantitation of MARCH8 protein levels in VIVIT-transfected PANC-1 compared to control ($N = 3$). *$P = 0.0111$. (Q) qRT-PCR analysis showing an increase in MARCH8 mRNA levels upon NFATc1 overexpression in CFPAC-1 compared to control ($N = 3$). **$P = 0.0050$. (R) Representative western blots showing an increase in MARCH8 protein levels due to NFATc1 overexpression in CFPAC-1 cells compared to control. (S) Densitometric quantitation of MARCH8 protein levels in NFATc1 overexpressed CFPAC-1 compared to control ($N = 3$). *$P = 0.0390$. (T) qRT-PCR analysis showing a decrease in MARCH8 mRNA levels upon VIVIT transfection in CFPAC-1 compared to control ($N = 3$). *$P = 0.0288$. (U) Representative western blots showing a decrease in MARCH8 protein levels due to VIVIT transfection in CFPAC-1 cells compared to control. (V) Densitometric quantitation of MARCH8 protein levels in VIVIT-transfected CFPAC-1 compared to control ($N = 3$). **$P = 0.0053$. Data presented are mean ± SEM. For statistical analysis, one-sample *t* test was performed for (C, D, E, G, H, J, K, M, N, P, Q, S, T, V) using GraphPad Prism software. Here, *$P < 0.05$; and **$P < 0.01$. Source data are available online for this figure.

elevated MARCH8 expression and the survival time of PC patients using data from the 'OncoLnc' database. This analysis revealed that the patients with low MARCH8 expression survived for a shorter duration as compared to patients with high MARCH8 expression (Fig. 9A). This highlights that higher MARCH8 levels are associated with better prognosis in PC patients. It further suggests that MARCH8 could play a role in PC progression and metastasis that eventually leads to mortality. To investigate MARCH8's role in PC progression and metastasis, we generated stable CFPAC-1 cell lines using lentiviral transduction, expressing either a non-targeting control shRNA (shNT) or an shRNA specifically targeting MARCH8 (shMARCH8). Western blot analysis confirmed a marked reduction in MARCH8 protein levels in CFPAC-1 shMARCH8 cells compared to the shNT control (Fig. 9B,C). After confirming the knockdown, we performed scratch wound assays to explore the role of MARCH8 in CFPAC-1 cell migration. Cell migration post-scratch was monitored by assessing wound closure temporally. The results showed that CFPAC-1 shMARCH8 cells closed the wound more effectively than shNT cells (Fig. 9D), shMARCH8 cells closed 50% of the wound within 24 h, while shNT cells closed less than 20% (Fig. 9E). Next, we performed transwell Boyden chamber-based migration assays with CFPAC-1 shNT, shMARCH8, MARCH8 overexpressed, and corresponding control cells. CFPAC-1 shMARCH8 showed increased cell migration compared to shNT cells, while CFPAC-1 MARCH8 over-expressed cells showed decreased cell migration compared to empty vector (EV) control cells (Fig. 9F,G). This demonstrates the involvement of MARCH8 in regulating pancreatic cancer cell migration. To examine the role of MARCH8 in PC metastasis in vivo, we carried out zebrafish xenograft experiments by injecting stable CFPAC-1 cells into the peri-vitelline space (PVS) of 2-day post-fertilization (dpf) zebrafish (Fig. 9H) and monitored metastatic events four hours post-injection (Martinez-Lopez et al, 2021; White and Patton, 2023). As shown in Fig. 9I,J, zebrafish injected with CFPAC-1 shMARCH8 cells exhibited increased metastatic events compared to those injected with CFPAC-1 shNT cells.

Altogether, the in vitro and in vivo data highlight that MARCH8 acts as a negative regulator of PC metastasis.

To summarize, in this study, we have identified that the same transcription factor regulates Orai3 transcription and protein degradation. Firstly, we report that NFATc1 binds to Orai3 promoter and controls its transcription in non-metastatic, invasive and metastatic PC cells. But additionally, NFATc1 activates Orai3 lysosomal degradation system facilitated by MARCH8 E3 ubiquitin ligase in invasive and metastatic PC cells. Through gain and loss-of-function experiments along with site-directed mutagenesis and deletion-based biochemical studies and high-resolution microscopy, we establish that MARCH8 physically interacts, ubiquitinates and subsequently degrades Orai3. Notably, we show that epigenetic differences in the MARCH8 promoter allow NFATc1 to drive MARCH8 transcription and promote Orai3 protein degradation in invasive and metastatic cells, while this mechanism is inefficient in non-metastatic cells. In non-metastatic PC cells, the MARCH8 promoter is hypermethylated and hence hinders NFATc1-guided transcription while in invasive and metastatic PC cells, MARCH8 promoter is relatively hypomethylated. Furthermore, we report a critical role of MARCH8 in PC metastasis, as our in vitro and in vivo findings suggest that MARCH8 restricts PC metastasis. Collectively, we have identified a dual regulator of Orai3 expression in a context-dependent manner (Fig. 10). Moreover, we have revealed the therapeutic potential of targeting MARCH8 E3 ubiquitin ligase to curtail PC metastasis.

## Discussion

$Ca^{2+}$ signaling plays a key role in regulating cancer progression. A previous study from our group revealed a critical role of Orai3 channel in regulating pancreatic cancer development and metastasis in vivo. We reported that Orai3 is transcriptionally upregulated in pancreatic tumor samples compared to normal pancreatic samples. Orai3 overexpression is well-documented in

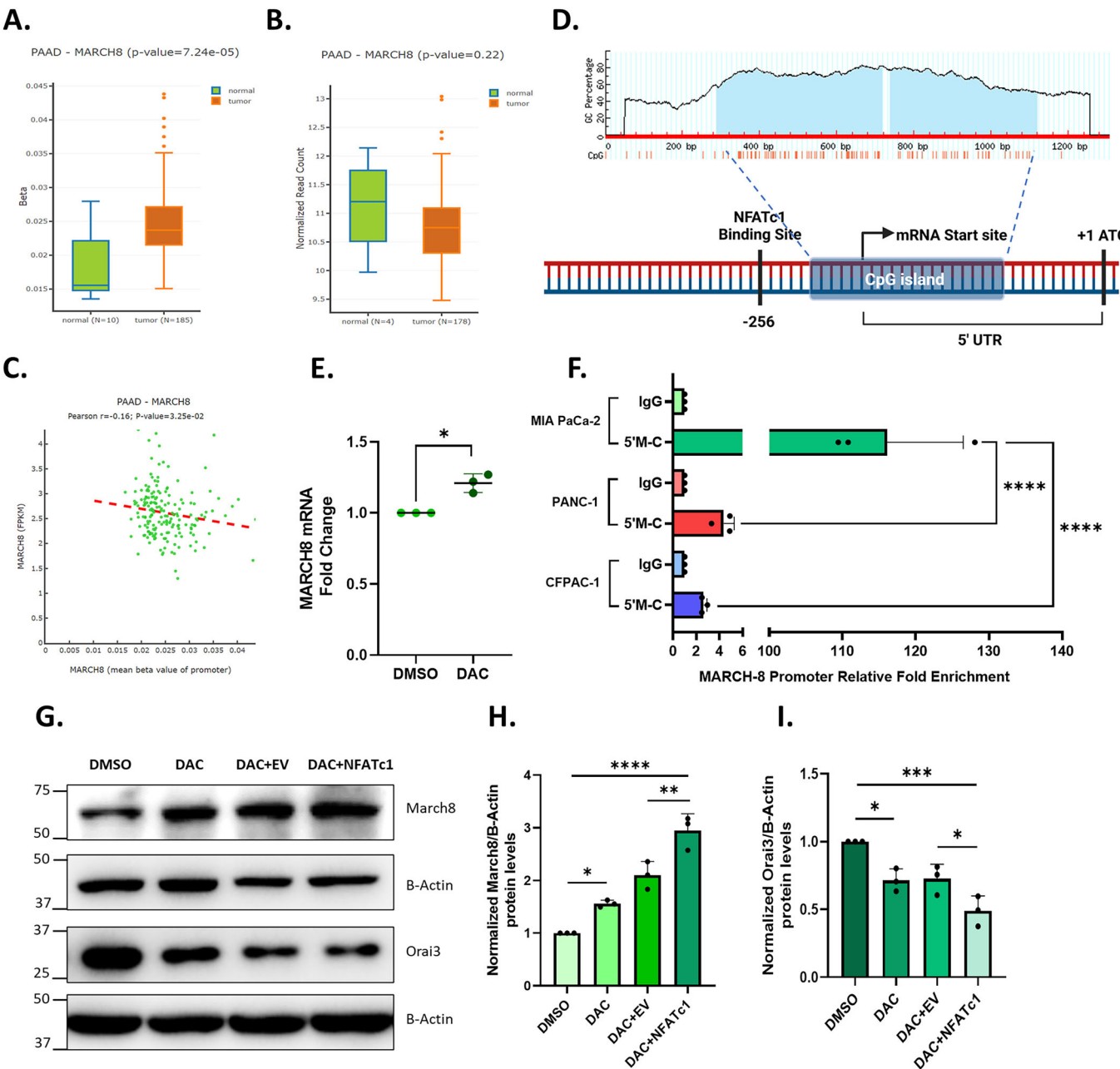

**Figure 8. Epigenetic landscape of MARCH8 promoter dictates NFATc1 driven regulation in non-metastatic v/s metastatic PC cells.**

(**A**) DNA methylation levels of the MARCH8 promoter region between normal tissues and pancreatic adenocarcinoma (PAAD) tissues analyzed by the DNMIVD database. (Normal: maxima-0.0279; minima-0.0135; median-0.0155; Q1-0.0147; Q3-0.0221. Tumor: maxima-0.0438; minima-0.0150; median-0.0236; Q1-0.0215; Q3-0.0271). (**B**) MARCH8 expression levels in normal pancreatic tissues and pancreatic adenocarcinoma (PAAD) tissues analyzed by the DNMIVD database. (Normal: maxima-12.14; minima-9.97; median-11.20; Q1-10.51; Q3-11.74. Tumor: maxima-13.04; minima-9.47; median-10.74; Q1-10.30; Q3-12.04). (**C**) Correlation between MARCH8 expression and DNA methylation level of MARCH8 promoter in PAAD tissues. (**D**) Schematic showing the location of the CpG island in the MARCH8 promoter. (**E**) qRT-PCR analysis showing increase in MARCH8 mRNA levels upon decitabine (5 µM) treatment in MiaPaCa-2 compared to control (N = 3). *P = 0.0310. (**F**) MeDIP qPCR analysis for relative fold enrichment in 5-methylcytosine immunoprecipitated DNA samples showing robust enrichment of MARCH8 promoter region in MiaPaCa-2 compared to PANC-1 and CFPAC-1 (N = 3). ****P < 0.0001; ****P < 0.0001. (**G**) Representative western blots showing an increase in MARCH8 levels and a decrease in Orai3 levels upon NFATc1 overexpression in decitabine-treated MiaPaCa-2 cells. (**H**) Densitometric analysis of MARCH8 levels upon NFATc1 overexpression in decitabine-treated MiaPaCa-2 cells (N = 3). *P = 0.0466; **P = 0.0051; ****P < 0.0001. (**I**) Densitometric analysis of Orai3 levels upon NFATc1 overexpression in decitabine-treated MiaPaCa-2 cells. *P = 0.0175; ***P = 0.0005; *P = 0.0447. Data presented are mean ± SEM. For statistical analysis, Mann–Whitney *U* test was performed for (**A, B**), one-sample *t* test was performed for (**E, F**), while ordinary one-way ANOVA was performed followed by Tukey's multiple comparison test for (**H, I**) using GraphPad Prism software. Here, *P < 0.05; **P < 0.01; ***P < 0.001; and ****P < 0.0001. Source data are available online for this figure.

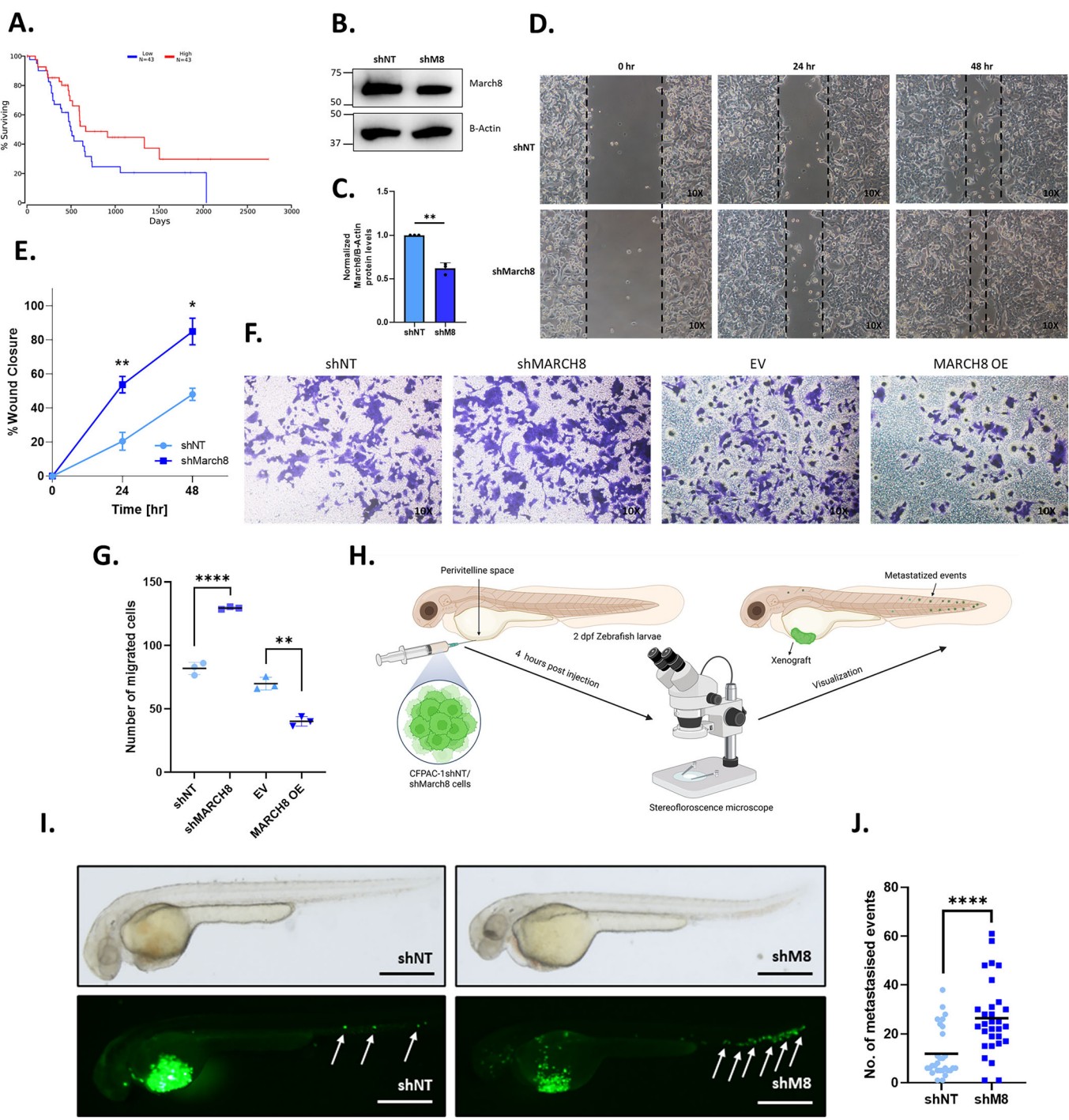

other types of cancers as well but the molecular players and signaling modules driving Orai3 regulation are still unknown. In this study, we discovered that NFATc1 regulates Orai3 expression. NFATc1 belongs to NFATc group of transcription factors. Interestingly, four isoforms of NFATc i.e., NFAT1, 2, 3, and 4 are $Ca^{2+}$ sensitive and are activated upon an increase in cytosolic $Ca^{2+}$ levels (Prakriya et al, 2006; Tanwar et al, 2024; Müller and Rao, 2010). Using in-silico analysis, we found potential binding sites of NFATc on the Orai3 promoter (Fig. 1B). Next, we

shortlisted NFATc1 as it significantly upregulates Orai3 at both mRNA and protein levels in HEK-293T cells (Fig. 1D–F) and GEPIA database showed a positive correlation of NFATc1 and Orai3 mRNAs in pancreatic tissues (Fig. 1G).

Interestingly, the NFATc1 overexpression and NFAT inhibition studies in pancreatic cancer cell lines showed a surprising dichotomy in the regulation of Orai3 by NFATc1 in non-metastatic v/s invasive and metastatic cells. We observed that in four distinct invasive and metastatic cells, NFATc1 acts as a

**Figure 9. MARCH8 E3 ubiquitin ligase abrogates metastatic potential in pancreatic cancer.**

(A) Survival analysis of pancreatic cancer patients wherein the blue trace corresponds to low MARCH8 expression ($n = 43$), and red trace corresponds to high MARCH8 expression ($n = 43$) highlighting that higher MARCH8 expression increases patient's survival time. (B) Representative western blots showing MARCH8 levels in CFPAC1 shNT and shMARCH8 cells. (C) Densitometric analysis of MARCH8 protein levels in CFPAC-1 shNT control and shMARCH8 ($N = 3$). **$P = 0.0095$. (D) Representative wound-healing images of CFPAC-1 shNT and shMARCH8 cells at 0, 24, and 48 h. (E) Quantitation of % wound closure by CFPAC-1 shNT and shMARCH8 cells over the interval of 48 h ($N = 3$). **$P = 0.0096$; *$P = 0.0127$. (F) Representative images of crystal violet-stained transwell-migrated shNT, shMARCH8, empty vector (EV) control and MARCH8 overexpressed CFPAC-1 cells. (G) Quantitative analysis of the number of migrated cells in the transwell migration assay from three independent experiments ($N = 3$). ****$P < 0.0001$; **$P = 0.0012$. (H) Schematic diagram showing the workflow for zebrafish xenograft injections. (I) Representative bright-field and fluorescent images of 2dpf zebrafish injected with GFP-labeled CFPAC-1 shNT and shMARCH8 cells. CFPAC-1 shMARCH8 cells show increased metastasized events compared to control. Metastasized events are indicated with white arrow marks. Scale: 1 mm. (J) Quantitation of metastasized events in CFPAC-1 shNT and shMARCH8 injected zebrafish (each dot represents data from individual embryos from three independent breeding experiments; $N = ~30$). ****$P < 0.0001$. Data presented are mean ± SEM. For statistical analysis, one-sample $t$ test was performed for (C) while unpaired Student $t$ test was performed for (E, G, J) using GraphPad Prism software. Here, *$P < 0.05$; **$P < 0.01$; and ****$P < 0.0001$. Source data are available online for this figure.

bimodal regulator to maintain Orai3 levels as NFATc1 over-expression leads to a decrease in Orai3 at the protein level (Figs. 2F–H,K–M and EV3G–I,M–O). This negative feedback loop in invasive and metastatic cells could be because invasive and metastatic cells have higher endogenous Orai3 levels compared to non-metastatic cells (Arora et al, 2021). A further increase in Orai3 levels by NFATc1 would allow an excess amount of $Ca^{2+}$ entering into the cell, which could lead to apoptosis. While in two independent non-metastatic cells i.e., MiaPaCa-2 and BxPC-3, NFATc1 transcriptionally upregulates Orai3, leading to increase in Orai3 at protein levels (Figs. 2A–C and EV3A–C). Therefore, our data demonstrates that a $Ca^{2+}$ sensitive transcription factor induces transcription of a $Ca^{2+}$ channel, thereby indicating the presence of a positive feedback loop in these cells. However, further studies are required to decipher the source of rise in cytosolic $Ca^{2+}$ and $Ca^{2+}$ handling toolkit involved in increasing cytosolic $Ca^{2+}$ levels for activating NFATc1.

Another interesting observation from our study is that inhibition of NFAT activity by VIVIT leads to transcriptional upregulation of Orai3 in non-metastatic, invasive, and metastatic cells (Fig. 3A,F,K). To understand this better, we performed luciferase assays with wild-type Orai3 promoter and the truncated Orai3 promoter with no NFATc1 binding sites. The NFAT inhibition via VIVIT transfection led to an increase in luciferase activity in both wild-type and the truncated Orai3 promoter (Fig. EV2A). This indicates that inhibition of NFAT's activity by VIVIT either activates or allows other endogenous transcription factors to upregulate Orai3 transcription independent of NFAT. Nonetheless, our data elegantly demonstrates that NFATc1 is a critical regulator of both Orai3 transcription and degradation in a context-dependent manner.

It is important to highlight that just like transcriptional regulation, Orai3's protein degradation process is not appreciated yet. The bimodal regulation of Orai3 by NFATc1 in invasive and metastatic pancreatic cancer cells led to the characterization of Orai3 protein degradation machinery. Using lysosomal and proteasomal degradation inhibitors, we observed that NFATc1 induces lysosomal degradation of Orai3 in both invasive and metastatic pancreatic cancer cells (Fig. 4A–D). Orai3 is a plasma membrane (PM) protein, and most PM proteins are degraded in lysosomes, although the mechanism and the players involved in the degradation process are different. Typically, the PM proteins are ubiquitinated by E3 ubiquitin ligases and are then delivered to the late endosomes/multivesicular body (MVB) via endocytosis. The

endosomes containing ubiquitinated PM proteins eventually fuse with lysosomes and are finally degraded by lysosomal proteases (Zhao et al, 2022). Our unbiased and robust bioinformatics narrowed down to MARCH8 E3 ubiquitin ligase as a potential regulator of Orai3 degradation (Fig. 4E). Indeed, MARCH8 overexpression and knockdown experiments show that MARCH8 induces Orai3 degradation in non-metastatic, invasive, and metastatic cells (Figs. 4 and 5). Mechanistically, MARCH8 ubiquitinates Orai3 and induces lysosomal degradation (Fig. 6). Importantly, by performing Orai3 site-directed mutagenesis and deletion studies, we reveal that MARCH8 physically interacts with Orai3 intracellular loop (between transmembrane 2 and 3) and facilitates Orai3 ubiquitination at K2 residue at N-terminal (Fig. 6B–F). Collectively, this data established that MARCH8 E3 ubiquitin ligase physically interacts with Orai3 at plasma membrane and induces lysosomal degradation.

Next, we investigated whether NFATc1 transcriptionally regulates MARCH8 and thereby induces Orai3 protein degradation. To examine this, we did in-silico analysis and found that NFATc1 has one potential binding site on human MARCH8 promoter, which is conserved across primates (Fig. 7A). Further, NFATc1 overexpression and VIVIT transfection studies in non-metastatic, invasive and metastatic PC cells establish that NFATc1 acts as a repressor in non-metastatic cells while acting as an activator in invasive and metastatic cells (Fig. 7). This clearly explains that the NFATc1 induced Orai3 protein degradation in invasive and metastatic cells is due to increase in MARCH8 expression. The transcriptional regulation of MARCH8 by NFATc1 presents an intriguing area for exploration beyond cancer biology. Both NFATc1 and MARCH8 are prominent players in human immunology. NFATc1 regulates the transcription of critical immune receptors such as ICOS, PD-1, CXCR5, CD40L, and chemokines like IL-4 and IL-21 (Vaeth and Feske, 2018). While MARCH8, an E3 ubiquitin ligase, targets MHC-II and CD80 receptors and plays a crucial role in antiviral defense (Lin et al, 2019). Therefore, investigating the relationship between NFATc1 and MARCH8 in immune cells would be beneficial for advancing our understanding of cancer immunotherapy, immune dysfunction, autoimmune disorders, and infectious disease biology.

The dual role of NFATc1 as a repressor and activator of MARCH8 in pancreatic cancer cells depending on the cell type points out that there could be epigenetic differences in the MARCH8 promoter region. Indeed, using in-silico analysis we found that MARCH8 promoter is epigenetically modified, more specifically shows DNA methylation.

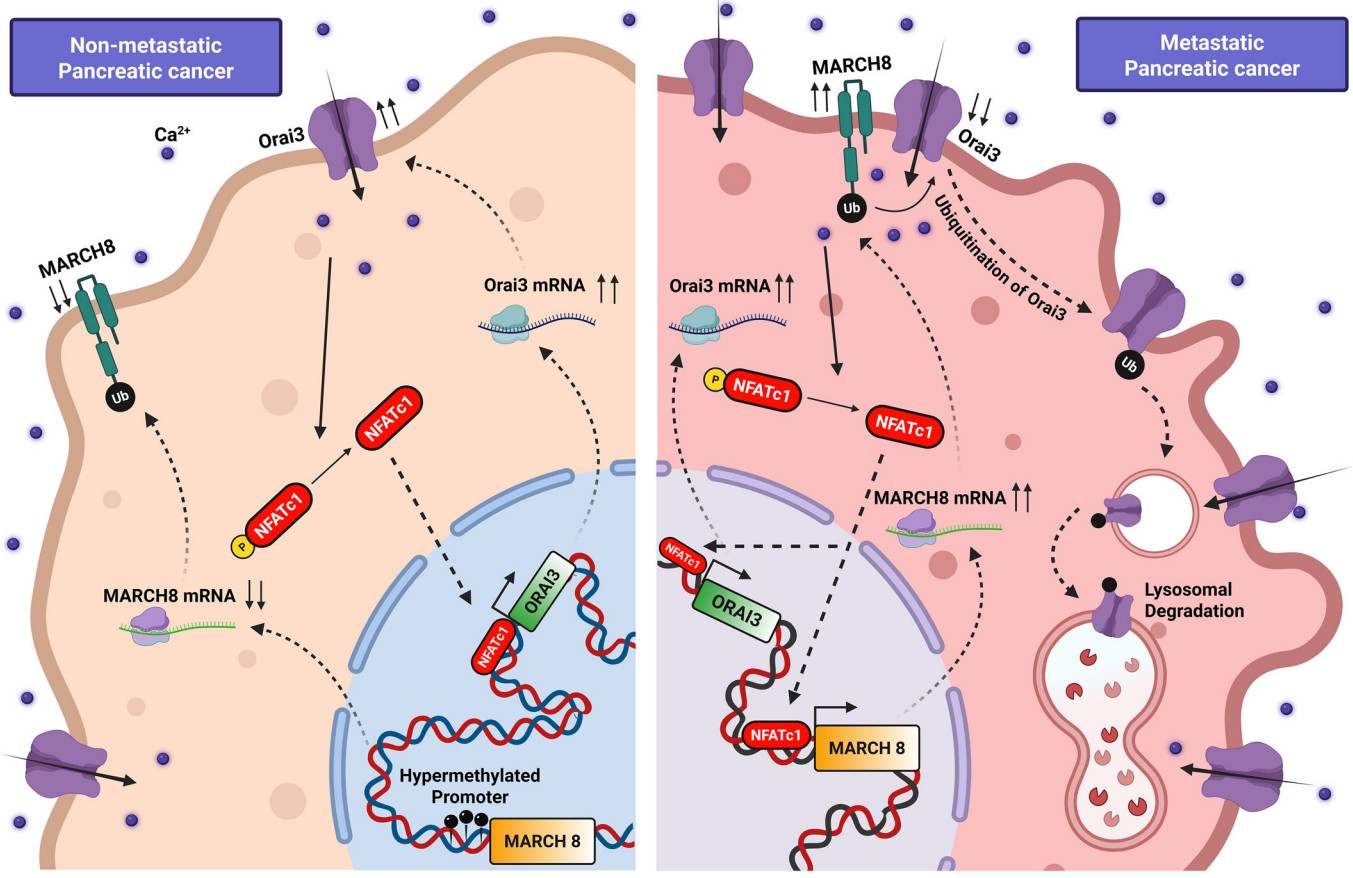

**Figure 10. Dual role of NFATc1 in regulating Orai3 transcription and protein degradation.**

In non-metastatic cells, NFATc1 drives Orai3 transcriptional upregulation, whereas in metastatic cells, NFATc1 induces Orai3 lysosomal degradation via MARCH8 E3 ubiquitin ligase. The dichotomy in NFATc1's function in non-metastatic versus metastatic cells is an outcome of differences in the methylation status of MARCH8 promoter in these cells.

DNA methylation is one of the most common epigenetic modifications in the genome and generally DNA methylation suppresses gene activation (Nishiyama and Nakanishi, 2021). Interestingly, we found that MARCH8 promoter is hypermethylated in pancreatic tumor samples compared to normal pancreatic samples (Fig. 8A) and consequently, the expression of MARCH8 is lower in tumor samples compared to normal pancreatic samples (Fig. 8B). Using MeDIP assays, we show that there are differences in DNA methylation levels of MARCH8 promoter in non-metastatic, invasive and metastatic cell lines. Our results demonstrate that the MARCH8 promoter is hypermethylated in non-metastatic PC cells (Fig. 8F). To examine if this methylation hinders NFATc1 from regulating MARCH8 transcription in non-metastatic cells, we overexpressed NFATc1 in MiaPaCa-2 cells treated with DNA methylase inhibitor (decitabine). Western blot analysis revealed that NFATc1 overexpression in decitabine-treated cells led to an increase in MARCH8 levels compared to other control conditions (Fig. 8G–I). In summary, the MARCH8 promoter is hypermethylated in non-metastatic PC cells, preventing NFATc1 from promoting MARCH8-mediated Orai3 protein degradation in these cells. Interestingly, our pilot studies suggest that there are epigenetic changes in the Orai3 promoter during cancer progression. However, it is beyond the scope of this manuscript to include that data.

Therefore, our future studies will present a comprehensive landscape of Orai3 epigenome during cancer progression.

Intriguingly, MARCH8 regulates cancer progression both positively and negatively depending upon the type of cancer (Chen et al, 2021; Xu et al, 2023; Chen et al, 2023a; Ying et al, 2024). MARCH8's role in regulating tumorigenesis depends on its degradation targets. For instance, in breast cancer cells, MARCH8 ubiquitinates STAT3 and CD44, suppressing tumor metastasis (Chen et al, 2021), while in hepatocellular carcinoma, MARCH8 induces degradation of PTEN thereby promoting malignancy (Xu et al, 2023). However, the role of MARCH8 in pancreatic cancer biology remains poorly appreciated. Here, our functional studies with stable knockdown and overexpression of MARCH8 signify its role in pancreatic cancer migration (Fig. 9D–G). More importantly, our in vivo zebrafish xenograft experiments show that stable knockdown of MARCH8 increases the metastatic potential of the pancreatic cancer cells (Fig. 9I,J). It is important to note that we have earlier reported that Orai3 positively drives PC progression and metastasis (Arora et al, 2021). Therefore, MARCH8 controls PC metastasis at least partially by modulating Orai3 expression and activity.

To summarize, we reveal that NFATc1 regulates both the transcription and lysosomal degradation of the Orai3 oncochannel

in a context-dependent manner. In non-metastatic cancer cells, NFATc1 drives Orai3 transcription, increasing its levels. In contrast, in invasive and metastatic cells, NFATc1 promotes Orai3 degradation by upregulating the MARCH8 E3 ubiquitin ligase. Biochemical assays, molecular biology-based Orai3 deletions, and super-resolution microscopy confirm that MARCH8 physically interacts with Orai3, leading to its degradation. This regulatory shift is linked to the epigenetic differences, as the MARCH8 promoter is hypermethylated in non-metastatic cells, preventing NFATc1 from inducing Orai3 degradation. The dual regulation of Orai3 by the same transcription factor NFATc1 underscores a complex pathway in pancreatic cancer cells that controls disease progression and clinical outcomes.

# Methods

### Reagents and tools table

| Reagent/resource | Reference or source | Identifier or catalog number |
|---|---|---|
| **Experimental models** | | |
| HEK-293T (*H.sapiens*) | ATCC | CRL-3216 |
| MiaPaCa-2 (*H.sapiens*) | ATCC | CRL-1420 |
| PANC-1 (*H.sapiens*) | ATCC | CRL-1469 |
| CFPAC-1 (*H.sapiens*) | ATCC | CRL-1918 |
| BXPC-3 (*H.sapiens*) | ATCC | CRL-1687 |
| ASPC-1 (*H.sapiens*) | ATCC | CRL-1682 |
| SW1990 (*H.sapiens*) | ATCC | CRL-2172 |
| ASWT Zebrafish (*D.rerio*) | RCB Zebrafish facility | N/A |
| **Recombinant DNA** | | |
| pEGFP-C1 | Addgene | #6084-1 |
| EGFPC1-huNFATc1EE-WT | Addgene | #24219 |
| pMIG-hNFATc2 | Addgene | #74050 |
| pEGFP-C1 NFAT3 | Addgene | #10961 |
| pREP-NFAT4 | Addgene | #11790 |
| pGL4.23[luc2/minP] | Promega | # 9PIE841 |
| pEGFP-N1 | Addgene | 6085-1 |
| GFP-VIVIT | Addgene | #11106 |
| MARCH8-eGFP-N1 WT | Fujita et al (2013) | N/A |
| MARCH8-eGFP-N1 CS | Fujita et al (2013) | N/A |
| pcDNA3.1(+)/myc-His A | Addgene | V80020 |
| pcDNA3.1 Orai3 | Addgene | # 16370 |
| Orai3-eYFP | Trebak Lab | N/A |
| **Antibodies** | | |
| Orai3 | Abcam | ab254260 |
| MARCH8 | Sigma-Aldrich | HPA014597-100UL |
| Ubiquitin | Santa Cruz | sc-271289 |

| Reagent/resource | Reference or source | Identifier or catalog number |
|---|---|---|
| β-tubulin | Abcam | ab21058 |
| β-Actin | Abcam | ab8226 |
| ECL Anti-Rat IgG HRP | Sigma-Aldrich | NA935V |
| ECL Anti-Mouse IgG HRP | Sigma-Aldrich | NA931V |
| Normal Rabbit IgG | CST | 2729S |
| Anti-GFP | Abcam | ab290 |
| 6X His tag | Abcam | ab18184 |
| Goat anti-Rabbit IgG (H + L) Cross-Adsorbed Secondary Antibody, Alexa Fluor 546 | ThermoFisher | A-11010 |
| Anti-5-methylcytosine | CST | 28692 |
| **Oligonucleotides and sequence-based reagents** | | |
| qRT-PCR primers | This study | Appendix Table S1 |
| Cloning primers | This study | Appendix Table S2 |
| Orai3 promoter site-directed mutagenesis primers | This study | Appendix Table S3 |
| ChIP primers | This study | Appendix Table S4 |
| siRNA | This study | Appendix Table S5 |
| Orai3 overexpression site-directed mutagenesis primers | This study | Appendix Table S6 |
| MeDIP Primers | This study | Appendix Table S7 |
| **Chemicals, enzymes, and other reagents** | | |
| DMEM - high glucose | Sigma-Aldrich | D5648 |
| RPMI-1640 | Sigma-Aldrich | R6504 |
| Leibovitz's L-15 Medium w/ 2mM L-Glutamine | Himedia | AL011S |
| Fetal Bovine Serum (FBS) | ThermoFisher | 10270-106 |
| Horse Serum | Himedia | RM1239 |
| NP40 Cell Lysis Buffer | ThermoFisher | FNN0021 |
| Protease Inhibitor Cocktail | Sigma-Aldrich | 04693116001 |
| Immobilon-P PVDF Membrane | Millipore | IPVH00010 |
| SM powder | HiMedia | GRM1254 |
| Tris free base | HiMedia | MB029 |
| TWEEN 20 | Merck | P2287 |
| Immobilon western chemiluminescence HRP substrate | Millipore | WBKLS0500 |
| TB Green Premix Ex Taq | Takara Bio | RR420A |
| KpnI-HF | NEB | R3142S |
| HindIII | NEB | R0104S |
| NheI-HF | NEB | R3131S |
| DpnI | NEB | R0176S |

| Reagent/resource | Reference or source | Identifier or catalog number |
|---|---|---|
| T4 Polynucleotide Kinase | NEB | M0201S |
| T4 DNA Ligase | NEB | M0202S |
| Protein A Agarose Beads | Merck | 16-125 |
| Phosphate buffer saline | HiMedia | M1452 |
| Formaldehyde, 37% | HiMedia | MB059 |
| Glycine | HiMedia | MB013 |
| HEPES | Sigma-Aldrich | H3375 |
| Sodium chloride | HiMedia | MB023 |
| EDTA | Sigma-Aldrich | 431788 |
| Glycerol | Honeywell | I5013 |
| Triton-X 100 | HiMedia | TC286 |
| EGTA | HiMedia | MB130 |
| UltraPure Herring Sperm DNA Solution | ThermoFisher | 15634017 |
| Bovine serum albumin | HiMedia | MB083 |
| Lithium chloride | HiMedia | GRM768 |
| Sodium deoxycholate | HiMedia | RM131 |
| Sodium bicarbonate | HiMedia | MB045 |
| FURA-2 AM | ThermoFisher | F1221 |
| Magnesium chloride | HiMedia | MB237 |
| Potassium chloride | Sigma-Aldrich | P9333 |
| Calcium chloride | Sigma-Aldrich | C1016 |
| D-Glucose | Sigma-Aldrich | G7021 |
| Thapsigargin | Merck | T9033 |
| 2-APB | Merck | 100065 |
| Bafilomycin A1 | Merck | B1793 |
| Paraformaldehyde | HiMedia | TC703 |
| SlowFade Gold Antifade Mountant with DAPI | ThermoFisher | S36938 |
| 5-Aza-2'-deoxycytidine | Merck | 189826 |
| Proteinase K | ThermoFisher | EO0491 |
| Lipofectamine 2000 | ThermoFisher | 11668019 |
| Methanol | SRLChem | 65524 |
| Isopropanol | SRLChem | 62986 |
| Crystal Violet solution | Sigma-Aldrich | V5265 |
| **Software** | | |
| Quant Studio RT-PCR software | https://www.thermofisher.com/in/en/home/global/forms/life-science/quantstudio-6-7-flex-software.html?erpType=Global_E1%2CSAP_PR1_2040 | |
| GraphPad | https://www.graphpad.com/ | |

| Reagent/resource | Reference or source | Identifier or catalog number |
|---|---|---|
| ImageJ | https://imagej.net/ij/ | |
| Snapgene | https://www.snapgene.com/ | |
| Zen (Black Edition) | https://www.micro-shop.zeiss.com/en/us/softwarefinder/software-categories/zen-black | |
| **Other** | | |
| JetPRIME Transfection Kit | Polyplus | 114-01 |
| Pierce BCA protein assay kit | ThermoFisher | 23227 |
| RNeasy mini kit | Qiagen | 74106 |
| Plasmid miniprep kit | Qiagen | 27106 |
| Plasmid midiprep kit | Qiagen | 12145 |
| High-capacity cDNA reverse transcription kit | ThermoFisher | 4368814 |
| Phusion High-Fidelity PCR Kit | ThermoFisher | F553L |
| Dual-Luciferase Reporter Assay System | Promega | E1910 |
| DNeasy Blood and Tissue Kit | Qiagen | 69504 |
| QIAquick PCR Purification Kit | Qiagen | 28106 |

## Cell culture

Human embryonic kidney cell line HEK-293T and human pancreatic cancer cell lines MiaPaCa-2, PANC-1, CFPAC-1, BXPC-3, ASPC-1, and SW1990 were acquired from ATCC. CFPAC-1, BXPC-3, and ASPC-1 were cultured in RPMI-1640 supplemented with L-Glutamine. MiaPaCa-2 were cultured in DMEM medium with 2.5% Horse serum and 10% fetal bovine serum (FBS). HEK 293T and PANC-1 were cultured in DMEM added with 10% FBS. SW1990 were cultured in Leibovitz's L-15 Medium and RPMI-1640 in 1:1 ratio with 10% FBS.

## qRT-PCR analysis

For the extraction of mRNA, the cell pellets were processed with the Qiagen RNeasy kit (Catalog #74106). cDNA was made from the extracted mRNA via a high-capacity cDNA reverse transcription kit from Thermo Fisher Scientific (Catalog #4368814). Real-time PCR were performed using SYBR green in Applied Biosystems Quant Studio 6 Flex. The analysis was done with Quant Studio RT-PCR software version 1.3. The mRNA expression of NFATc1, Orai3, and MARCH8 in experimental conditions were normalized to GAPDH (Housekeeping gene). Gene-specific primers used in this study are listed in Appendix Table S1.

## Western blotting

The total proteins from the cell pellets of experimental conditions were extracted using NP40 lysis buffer with protease inhibitor cocktail. The quantification of total protein concentration was done via BCA assay (Thermo Fisher Scientific, Catalog #23227). The extracted protein samples were separated by 10% SDS-PAGE and

electrotransferred to a PVDF membrane (Merck Millipore, Catalog #IPVH00010). Blocking of the membranes was done with 5% skimmed milk in Tris-buffered saline with Tween 20 (TBST) at room temperature for 2 h, and then incubated overnight at 4 °C with specific primary antibody in TBST, like rabbit anti-Orai3 (1:500, Abcam), rabbit anti-MARCH8 (1:1000, Sigma), and mouse anti-Ubiquitin (1:1000, Santa Cruz). After the overnight incubation, the membranes are incubated in horseradish peroxidase conjugated donkey anti-rabbit secondary antibody (1:10,000, Sigma) and horseradish peroxidase conjugated horse anti-mouse secondary antibody (1:1000, Cell Signalling Technologies) for 2 h at room temperature. Finally, the membranes were detected using the Immobilon Forte Western HRP Substrate (Merck Millipore, Catalog #WBLUF0500). β-tubulin (1:5000, Abcam) and β-Actin (1:10000, Abcam) were used as loading controls. Densitometric analysis of the blots was achieved using ImageJ software.

## Cloning of full-length Orai3 promoter, Orai3 promoter mutants, truncated Orai3 promoter and MARCH8 promoter

The human Orai3 promoter and MARCH8 promoter was obtained from Eukaryotic Promoter Database (EPD), and the sequence was analyzed using NCBI-BLAST to find mRNA start site and the first codon. Primers were designed to amplify 1055 bp region (−1024 to +31 w.r.t. start codon) of the Orai3 core promoter and 1042 bp region (−989 to +50 w.r.t. start codon) of the MARCH8 core promoter, respectively. PCR amplification of the full-length Orai3 promoter, truncated Orai3 promoter, and MARCH8 promoter was done using Phusion High-Fidelity Polymerase (Catalog #F503, Thermo Fisher Scientific), and the PCR amplicon was cloned into pGL4.23 luciferase reporter vector (Promega) at the KpnI/HindIII sites for Orai3 promoter and KpnI/NheI sites for MARCH8 promoter, respectively. The clones were verified through restriction digestion and sequencing. Further, deletion mutants of the Orai3 promoter (Δ1017, Δ990, Δ920) were generated at the NFATc1 consensus binding sites and were verified by sequencing. Primers used for cloning are listed in Appendix Tables S2 and S3.

## In vitro dual-luciferase assay

Before transfection, PANC-1 cells were seeded at a density of $0.5 \times 10^5$ cells/well in 24-well plates. Cells were transfected with Orai3P and eGFPC1-NFATc1, and MARCH8P and eGFPC1-NFATc1 as indicated, using jetPRIME transfection reagent (Catalog #114-01, Polyplus) as per the manufacturer's protocol. For transfection normalization, the Renilla luciferase plasmid was used. The cells were assayed for luciferase activity using the dual-luciferase assay kit (Catalog #E1910, Promega) as per the manufacturer's protocol 48 h post transfection. Data are representative of three biological replicates.

## Chromatin immunoprecipitation (ChIP)

PANC-1 cells transfected with EGFPC1-huNFATc1EE-WT, pMIG-hNFATc2, pEGFP-C1-NFATc4, pREP-NFATc3 overexpression plasmid or pEGFPC1-empty vector and were trypsinized after 48 h. The cell count was done, and 25 million cells aliquot was resuspended in 10 ml

1× PBS; The aliquots were fixed with 1% formaldehyde for 3 min at room temperature, then quenched with 1 M Glycine. After quenching, the cells were washed once with cold 1× PBS and pelleted using centrifugation at 1500 rpm for 5 min at 4 °C. To shear the chromatin, cell pellets were resuspended in 10 ml of Rinse Buffer 1 (50 mM Hepes pH 8, 140 mM NaCl, 1 mM EDTA, 10% glycerol, 0.5% NP-40, 0.25% Triton X-100) and incubated for 20 min on ice. Cells were next resuspended in Rinse Buffer 2 (10 mM Tris pH 8, 1 mM EDTA, 0.5 mM EGTA, 200 mM NaCl) and incubated on ice for 20 min. Finally, cells were resuspended in Shearing buffer (0.1% SDS, 1 mM EDTA, 10 mM Tris pH 8). For sonication, five million cells were resuspended in 200 μl of shearing buffer and sonicated using Bioruptor Pico with 30 s ON and 30 s OFF for 35 cycles at 4 °C. Sonicated chromatin was centrifuged at 16,000 rpm for 10 min to remove debris. DNA concentration of the sonicated chromatin was determined, and 25 μg of sonicated chromatin was used for immunoprecipitation using 2 μg GFP antibody or IgG control. The protein A agarose beads (Catalog #16-125, Merck Millipore) used for immunoprecipitation were first equilibrated with glycerol IP buffer, and blocking was done using 75 ng/μl Herring sperm DNA and 0.1 μg/μl bovine serum albumin. Post immunoprecipitation, the beads were washed twice each with low salt buffer (0.1% SDS, 1% Triton X100, 2 mM EDTA, 20 mM Tris–Cl pH 8, 150 mM NaCl), high salt buffer (0.1% SDS, 1% Triton X100, 2 mM EDTA, 20 mM Tris– Cl pH 8, 500 mM NaCl), LiCl buffer (0.25 M LiCl, 1% NP-40, 1% sodium deoxycholate, 1 mM EDTA, 10 mM Tris–Cl pH 8) and TE buffer. DNA was eluted from the beads using Elution Buffer (1% SDS, 100 mM NaHCO₃) for 30 min at 30 °C, and decrosslinking was performed at 80 °C for 1 h. Purification of DNA using phenol-chloroform extraction was performed and used for ChIP-qPCR using primers specific to human Orai3 core promoter and MARCH8 core promoter. Primers used for ChIP-qPCR are listed in Appendix Table S4.

## Calcium imaging

Calcium ($Ca^{2+}$) imaging was performed as reported earlier (Arora et al, 2021). Briefly, cells were cultured on confocal dishes for performing $Ca^{2+}$ imaging. Cells are incubated at 37 °C for 30 min in a culture medium containing 4 μM FURA-2 AM. After incubation, cells were washed 3 times and bathed in HEPES-buffered saline solution (140 mM NaCl, 1.13 mM MgCl₂, 4.7 mM KCl, 2 mM CaCl₂, 10 mM D-glucose, and 10 mM HEPES; pH 7.4) for 5 min before $Ca^{2+}$ measurements were made. A digital fluorescence imaging system (Nikon Eclipse Ti2 microscope coupled with CoolLED pE-340 Fura light source and a high-speed PCO camera) was used, and fluorescence images of several cells were recorded and analyzed. Fura-2AM was excited alternately at 340 and 380 nm, and the emission signal was captured at 510 nm. Figures showing $Ca^{2+}$ traces are an average from several cells (the number of cells is denoted as "N" on each trace) attached to a single imaging dish. Each experiment was performed at least 3–4 times, and the final data are plotted in the form of bar graphs. The data shown in a particular $Ca^{2+}$ imaging trace originates from multiple cells on a single imaging dish. The exact number of cells and traces for each condition is specified in the respective figure.

## siRNA-based transient transfections

siRNA transfections were done in MiaPaCa-2, PANC-1 and CFPAC-1 cells seeded in six-well plates. In all, 100 nM siNT

(Human) and siMARCH8 (Human) were transfected in cells using jetPRIME transfection reagent (Catalog #114-01, Polyplus) as per the manufacturer's protocol. Cells were harvested post 48 h of transfection for mRNA and protein expression changes. The siRNAs (smartpool of four individual siRNAs targeting gene of interest) were procured from Dharmacon and Eurogentec. The siRNA sequences are listed in the Appendix Table S5.

## Site-directed mutagenesis of Orai3 overexpression vector

Conventional PCR was performed using plasmid pcDNA3.1-Orai3 through Phusion High-Fidelity Polymerase (Catalog #F503, Thermo Fisher Scientific). Primers were designed to introduce the K2R, K274R, K279R mutations and LMVXXXL, GXXXG domain deletions in pcDNA3.1-Orai3 overexpression vector. The primer sequences are provided in Appendix Table S6. The PCR purified products were treated with DpnI, T4 polynucleotide kinase, and DNA ligase for 1 h at 37 °C. The products were transformed into DH5α competent cells. The mutations were confirmed by sequencing prior to experiments.

## Ubiquitination assay

For the ubiquitination assay, PANC-1 cells were cultured in T-75 flasks, and after reaching confluency, the cells were collected and lysed with glycerol IP buffer. The protein content in the lysate was calculated using the BCA assay, and from the cell lysate, 1 mg of protein was used to perform immunoprecipitation using an antibody against Orai3. The next day, the incubated lysates were precipitated by adding activated protein A agarose beads and incubating them overnight at 4 °C. Before immunoprecipitation, the protein A agarose beads were equilibrated with glycerol IP buffer and blocked with 0.1 μg/μl bovine serum albumin. Precipitation was also performed using Normal Rabbit IgG as an isotype control, raised in the same host. The resulting antigen–antibody complexes were processed in NP-40 lysis buffer for immunoblotting to check the levels of ubiquitin and Orai3. To study the effect of MARCH8 on Orai3 ubiquitination, PANC-1 cells were transfected with MARCH8-eGFP-N1 WT or pEGFPN1-empty vector using jetPRIME transfection reagent as per the manufacturer's protocol for 48 h. Before termination, the cells were treated with bafilomycin (250 nM) for 6 h. Further, the cells were processed as mentioned above. To explore the lysine residues ubiquitinated on Orai3, PANC-1 cells were transfected with pcDNA3.1 empty vector, pcDNA3.1-Orai3 WT, pcDNA3.1-Orai3 K2R, pcDNA3.1-Orai3 K274R, and pcDNA3.1-Orai3 K279R using jetPRIME transfection reagent as per the manufacturer's protocol for 48 h. Before termination, the cells were treated with bafilomycin (250 nM) for 6 h. Further, the cells were processed as mentioned above.

## Co-immunoprecipitation

For Co-immunoprecipitation, PANC-1 cells were lysed in glycerol IP buffer for 20 min on incubation on ice. Lysates were prepared by centrifugation at 16220 rpm for 10 min at 4 °C and collecting the supernatant. The supernatant after centrifugation was then incubated with anti-Orai3 antibody (Abcam) and anti-MARCH8 antibody (Sigma) for reverse Co-IP at 4 °C on an end-to-end rotor overnight. The next day, the incubated lysates were precipitated by the addition of activated protein A agarose beads for overnight at 4 °C. The protein A agarose beads used for immunoprecipitation were first equilibrated with glycerol IP buffer, and blocking was done using 0.1 μg/μl bovine serum albumin. Precipitations were also carried by IgG in the same host as the isotype control. The antigen–antibody complex obtained was processed in NP 40 lysis buffer for immunoblotting to check MARCH8 and Orai3 protein levels. To investigate the interaction domains of MARCH8 in Orai3, PANC-1 cells were transfected with pcDNA3.1 empty vector, pcDNA3.1-Orai3 WT, pcDNA3.1-Orai3ΔLMVXXXL, and pcDNA3.1-Orai3ΔGXXXG using jetPRIME transfection reagent as per the manufacturer's protocol for 48 h. Further, the cells were processed as mentioned above.

## Colocalization immunofluorescence assay

PANC-1 cells cultured on glass slides were transfected with eYFP-Orai3 using jetPRIME transfection reagent as per the manufacturer's protocol. After 48 h, the cells were washed with PBS and fixed with 4% paraformaldehyde in PBS for 15 min. After three washes with PBS, the cells were permeabilized with PBS containing 0.3% Triton-X100 for 15 min and blocked with PBS containing 2% bovine serum albumin and 0.1% Triton-X100 for 1 h. The cells were then incubated with anti-MARCH8 antibody (1:200) overnight. After incubation, the cells were washed thrice with PBS containing 0.1% Tween20 and incubated with Alexa Fluor 568 goat anti-rabbit-conjugated secondary antibody (1:500) at 25 °C for 2 h. Finally, the cells were washed three times with PBS containing 0.1% Tween20 mounted using SlowFade Gold antifade reagent with 4′,6-diamidono-2-phenylindole (DAPI) (Catalog #S36938, Thermo Fisher Scientific), and visualized using a Carl Zeiss LSM 880 laser scanning confocal microscope at 63X (with oil) magnification. The size of the pinhole for each laser channel is as follows: MARCH8-Alexa Fluor 564 – 4.88 AU, Orai3-eYFP – 1.77 AU, DAPI – 14.11 AU.

## Methylated DNA immunoprecipitation (MeDIP)

The genomic DNA of MiaPaCa-2, PANC-1, and CFPAC-1 cells was extracted using DNeasy Blood and Tissue Kit (Catalog #69504) according to the manufacturer's instructions. Genomic DNA was sonicated using Bioruptor Pico with 30 s ON and 30 s OFF for ten cycles at 4 °C to get DNA fragments between sizes 300 and 500 bp. Sonicated DNA was denatured by incubating at 95 °C for 10 min followed by incubation on ice for 5 min. In total, 3 μg of sonicated DNA was incubated with 1 μg anti-5-Methylcytosine (Cell Signalling Technologies) antibody along with normal rabbit IgG at 4 °C overnight. 30 μl Protein A agarose beads were added and incubated overnight at 4 °C. The beads were washed thrice with glycerol IP buffer at 4 °C for 5 min and eluted in 150 μl elution buffer (1 M Tris-HCl pH 8.0, 0.5 M EDTA, 10% SDS) with proteinase K (Catalog #EO0491, Thermo Fisher Scientific) treatment for 2 h at 55 °C on a rotating platform. Purification of DNA using phenol-chloroform extraction was performed and used for MeDIP-qPCR using primers specific to human MARCH8 core promoter. The sequences of primers used for MeDIP-qPCR are listed in the Appendix Table S7.

## Lentiviral stable cell line generation

For stable knockdown generation, human-specific shNT and shMARCH8 cloned in the lentiviral pGIPZ vector (Dharmacon, Lafayette, CO, USA) were used. As reported earlier, the lentiviral constructs pCMV-VSVG, pCMV- dR8.2 and pGIPZ-shNT/shMARCH8 were co-transfected in a flask containing 95% confluent HEK-293T cells. Lipofectamine 2000 (Catalog #11668019, Thermo Fisher) was used as a transfection reagent to transfect HEK-293T cells. Viral particles containing medium were collected at 48 and 72 h after transfection and were concentrated using Amicon filters through centrifugation. These concentrated viral particles were used to transduce CFPAC-1 cells seeded at 50% confluency, and knockdown was confirmed by performing western blot analysis.

## Scratch wound-healing assay

CFPAC-1 lentiviral stable cells were plated in a 24-well plate at a density of $3 \times 10^5$ cells per well. Once the cells reached full confluency, a scratch was made in the center of each well using a P20 pipette tip. The cells were then rinsed with PBS, and fresh media were added to each well. Scratch wounds were monitored at various time points under a bright-field microscope to observe cell migration. The migration rate of CFPAC-1 shNT and shMARCH8 cells was determined by measuring the distance traveled by the cells from the initial to the final time point. The wound closure was normalized to the 0-hour timepoints.

## Boyden chamber-based transwell-migration assay

Cell migration was assessed using 24-well Corning Costar inserts with 8-μm pores. Cells ($5 \times 10^4$) were added in the upper chamber in RPMI media (without FBS) and incubated at 37 °C, and migration was assessed at 48 h. Non-migrated cells were removed from the upper chamber with the help of cotton buds. Cells adhered at the bottom of the transwell were fixed with formaldehyde, permeabilized with 100% methanol, and stained with crystal violet, and bright-field images of three different fields were quantified using ImageJ. Experiments were performed in triplicate in three independent biological sample,s and data are reported as mean ± SEM.

## Zebrafish husbandry

The zebrafish used in this study were housed at the Regional Centre for Biotechnology (RCB). The zebrafish experiments were performed with the ethical approval of the Institutional Animal Ethics Committee (IAEC), RCB. The reference number of the approval is RCB/IAEC/2022/143.

## Zebrafish xenograft microinjections

CFPAC-1 shNT and shMARCH8 knockdown cells were micro-injected into the perivitelline space (PVS) of anesthetized 2 days post fertilization (dpf) zebrafish embryos and incubated at 34 °C. 3 h post injection, the embryos were screened and visualized for metastasized events, the images were acquired in Nikon SMZ800N Stereo Microscope and injections were performed using an Eppendorf FemtoJet 4i.

## Statistical analysis

All the experiments were performed at least three times. Data are presented as mean ± SEM and one-sample *t* test was performed for determining the statistical significance. For experiments with more than two conditions, one-way ANOVA test was performed. *P* value < 0.05 was considered significant and is presented as "*"; *P* value < 0.01 is presented as "**"; *P* value < 0.001 is presented as "***"; and *P* value < 0.0001 is presented as "****".

## Data availability

No data amenable to large-data repository deposition have been generated in this study.

The source data of this paper are collected in the following database record: biostudies:S-SCDT-10_1038-S44318-025-00572-4.

## Peer review information

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

## Acknowledgements

This work was supported by the Department of Biotechnology, India (project number BT/PR52477/MED/30/2540/2024). RKM also acknowledges funding support from RCB Institutional Core funding, Anusandhan National Research Foundation (ANRF) project number SERB-CRG/2023/004054, and DBT/Wellcome Trust India Alliance Fellowship (IA/I/19/2/504651). The authors thank members of the Motiani laboratory for discussions and critical reading of the manuscript. We thank Hideaki Fujita (Nagasaki International University, Japan) for sharing MARCH8-eGFP-N1 WT and MARCH8-eGFP-N1 CS plasmids. We also thank Mohamed Trebak (University of Pittsburgh, USA) for sharing the Orai3-eYFP plasmid. The technical assistance of Mr. Unni Narayanan and Ms. Tanisha Anand is highly appreciated.

## Author contributions

**Sharon Raju**: Formal analysis; Investigation; Visualization; Methodology; Writing—original draft. **Akshay Sharma**: Formal analysis; Investigation; Visualization; Methodology. **Sakshi Dahiya**: Formal analysis; Investigation. **Gyan Ranjan**: Investigation; Visualization. **Rajender K Motiani**: Conceptualization; Supervision; Funding acquisition; Writing—original draft; Project administration; Writing—review and editing.

Source data underlying figure panels in this paper may have individual authorship assigned. Where available, figure panel/source data authorship is listed in the following database record: biostudies:S-SCDT-10_1038-S44318-025-00572-4.

## Disclosure and competing interests statement

The authors declare no competing interests.

# Expanded View Figures

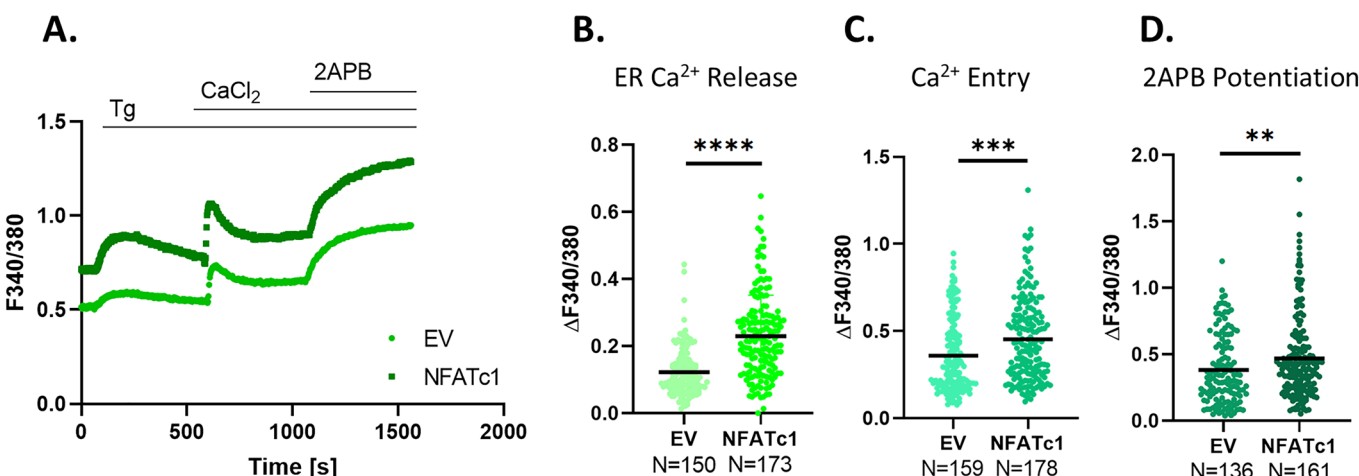

**Figure EV1. Calcium imaging in MiaPaCa-2 using standard thapsigargin activated SOCE protocol.**

(A) Representative $Ca^{2+}$ imaging traces of empty vector control and NFATc1 overexpression in MiaPaCa-2. (B) Quantitation of ER $Ca^{2+}$release upon NFATc1 overexpression in MiaPaCa-2 compared to empty vector control where "N" denotes the number of ROIs. ****$P< 0.0001$. (C) Change in $Ca^{2+}$entry upon overexpression of NFATc1 compared to empty vector control in MiaPaCa-2 where "N" denotes the number of ROIs. ***$P = 0.0002$. (D) 2-APB potentiation of Orai3 in NFATc1 overexpressed and empty vector control MiaPaCa-2 where "N" denotes the number of ROIs. **$P = 0.0097$. Data presented are mean ± SEM. For statistical analysis, unpaired Student's $t$ test was performed for(B–D) using GraphPad Prism software. Here, **$P < 0.01$; ***$P < 0.001$ and ****$P < 0.0001$. Source data are available online for this figure.

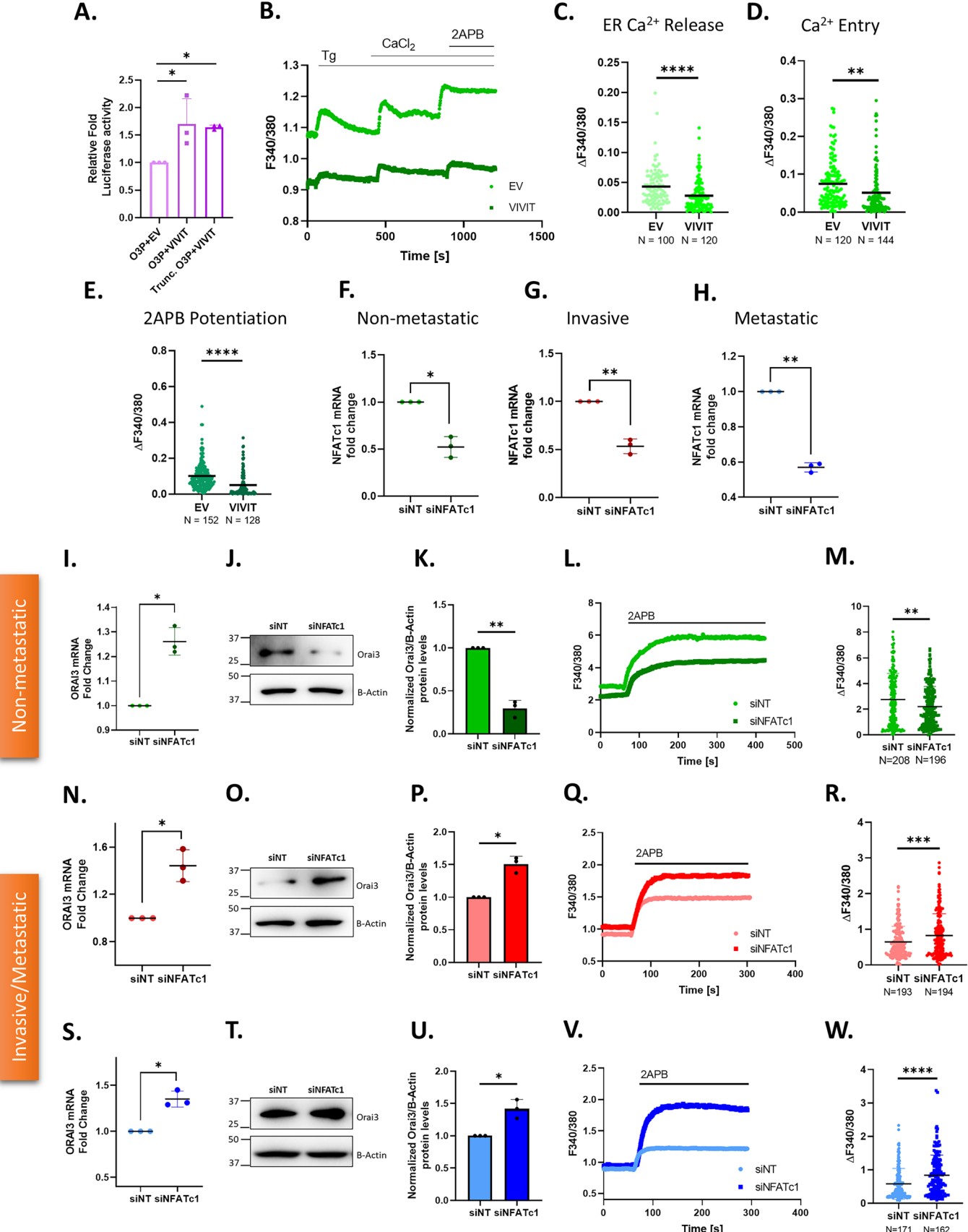

◀ **Figure EV2. siRNA mediated NFAT knockdown validates dichotomous regulation of Orai3 in non-metastatic v/s metastatic PC cells.**

(A) Normalized luciferase activity of wild-type Orai3 promoter and truncated Orai3 promoter in PANC-1 cells upon VIVIT transfection for 48 h ($N = 3$). *$P = 0.0307$; *$P = 0.0438$. (B) Representative $Ca^{2+}$ imaging trace of cells transfected with control vector pEGFP-N1 plasmid and VIVIT transfection in MiaPaCa-2. (C) Quantitation of ER $Ca^{2+}$release after VIVIT transfection in MiaPaCa-2 compared to empty vector control where "N" denotes the number of ROIs. ****$P < 0.0001$. (D) Change in $Ca^{2+}$entry upon VIVIT Transfection in MiaPaCa-2 where "N" denotes the number of ROIs. **$P = 0.0018$. (E) 2-APB potentiation of Orai3 in VIVIT-transfected and empty vector control MiaPaCa-2 where "N" denotes the number of ROIs. ****$P < 0.0001$. (F) qRT-PCR analysis showing NFATc1 knockdown validation in MiaPaCa-2 ($N = 3$). *$P = 0.0173$. (G) qRT-PCR analysis showing NFATc1 knockdown validation in PANC-1 ($N = 3$). **$P = 0.0088$. (H) qRT-PCR analysis showing NFATc1 knockdown validation in CFPAC-1 ($N = 3$). **$P = 0.0013$. (I) qRT-PCR analysis showing increase in Orai3 mRNA levels upon NFATc1 knockdown in MiaPaCa-2 compared to control ($N = 3$). *$P = 0.0148$. (J) Representative western blots showing decrease in Orai3 protein levels upon NFATc1 knockdown in MiaPaCa-2 cells compared to control.

(K) Densitometric quantitation of Orai3 protein levels in NFATc1 knockdown MiaPaCa-2 cells compared to control ($N = 3$). **$P = 0.0060$. (L) Representative $Ca^{2+}$ imaging trace of MiaPaCa-2 cells transfected with either control siNT or siNFATc1. (M). 2-APB potentiation of Orai3 in siNT and siNFATc1-transfected MiaPaCa-2 where "N" denotes the number of ROIs. **$P = 0.0024$. (N) qRT-PCR analysis showing increase in Orai3 mRNA expression upon NFATc1 knockdown in PANC-1 compared to control ($N = 3$). *$P = 0.0295$. (O) Representative western blots showing increase in Orai3 protein levels due to NFATc1 knockdown in PANC-1 compared to control. (P) Western blot densitometry of Orai3 protein levels after NFATc1 knockdown in PANC-1 cells compared to control ($N = 3$). *$P = 0.0185$. (Q) Representative $Ca^{2+}$ imaging trace of PANC-1cells transfected with either control siNT or siNFATc1. (R) Potentiation of Orai3 by 2-APB in control siNT and siNFATc1-transfected PANC-1 where "N" denotes the number of ROIs. ***$P = 0.0009$. (S) qRT-PCR analysis showing increase in Orai3 mRNA expression upon NFATc1 knockdown in CFPAC-1 compared to control ($N = 3$). *$P = 0.0201$. (T) Representative western blots showing increase in Orai3 protein levels due to NFATc1 knockdown in CFPAC-1 compared to control. (U) Western blot densitometry of Orai3 protein in NFATc1 knockdown CFPAC-1 cells compared to control (N = 3). *$P = 0.0381$. (V) Representative $Ca^{2+}$ imaging trace of CFPAC-1 cells transfected with either control siNT or siNFATc1. (W). Potentiation of Orai3 by 2-APB in siNT and siNFATc1-transfected CFPAC-1 where "N" denotes the number of ROIs. ****$P < 0.0001$. Data presented are mean ± SEM. For statistical analysis, unpaired Student's $t$ test was performed for (C–E, M, R, W) while one-sample $t$ test was performed for (F, G, H, I, K, N, P, S, U) using GraphPad Prism software. Here, *$P < 0.05$; **$P < 0.01$; ***$P < 0.001$ and ****$P < 0.0001$. Source data are available online for this figure.

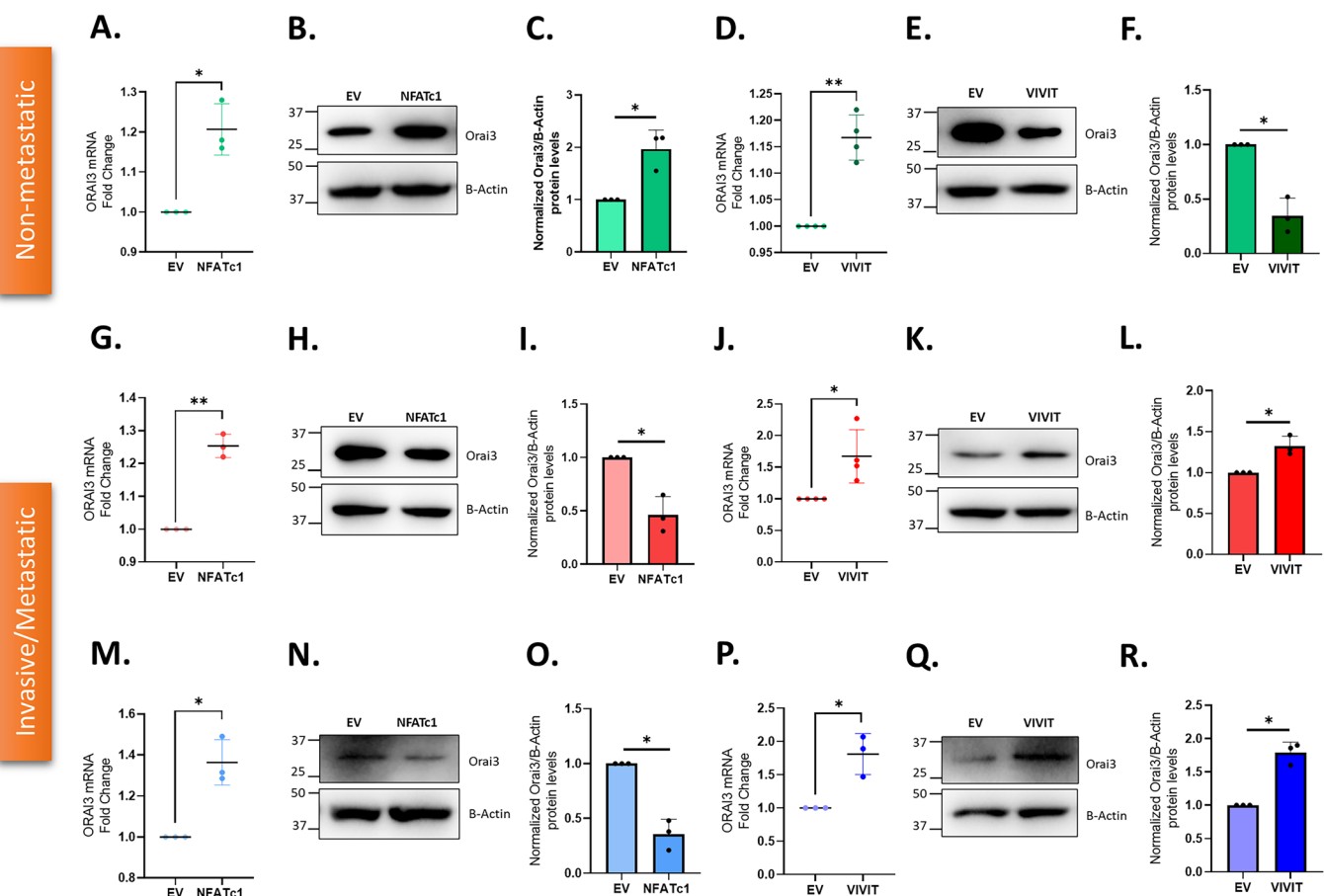

**Figure EV3. NFATc1 differently regulates Orai3 expression in BxPC-3 (non-metastatic), ASPC-1 (invasive) and SW1990 (metastatic) PC cells.**

(**A**) qRT-PCR analysis showing increase in Orai3 mRNA levels upon NFATc1 overexpression in BxPC-3 compared to control ($N = 3$). *$P = 0.0308$. (**B**) Representative western blots showing increase in Orai3 protein levels due to NFATc1 overexpression in BxPC-3 cells compared to control. (**C**) Densitometric quantitation of Orai3 protein levels in NFATc1 overexpressed BxPC-3 compared to control ($N = 3$). *$P = 0.0435$. (**D**) qRT-PCR analysis showing increase in Orai3 mRNA levels upon VIVIT transfection in BxPC-3 compared to control ($N = 3$). **$P = 0.0045$. (**E**) Representative western blots showing decrease in Orai3 protein levels due to VIVIT transfection in BxPC-3 cells compared to control. (**F**) Densitometric quantitation of Orai3 protein levels in VIVIT-transfected BxPC-3 compared to control ($N = 3$). *$P = 0.0198$. (**G**) qRT-PCR analysis showing increase in Orai3 mRNA levels upon NFATc1 overexpression in ASPC-1 compared to control ($N = 3$). **$P = 0.0063$. (**H**) Representative western blots showing decrease in Orai3 protein levels due to NFATc1 overexpression in ASPC-1 cells compared to control. (**I**) Densitometric quantitation of Orai3 protein levels in NFATc1 overexpressed ASPC-1 compared to control ($N = 3$). *$P = 0.0327$. (**J**) qRT-PCR analysis showing increase in Orai3 mRNA levels upon VIVIT transfection in ASPC-1 compared to control ($N = 3$). *$P = 0.0494$. (**K**) Representative western blots showing increase in Orai3 protein levels due to VIVIT transfection in ASPC-1 cells compared to control. (**L**) Densitometric quantitation of Orai3 protein levels in VIVIT-transfected ASPC-1 compared to control ($N = 3$). *$P = 0.0436$. (**M**) qRT-PCR analysis showing increase in Orai3 mRNA levels upon NFATc1 overexpression in SW1990 compared to control ($N = 3$). *$P = 0.0296$. (**N**) Representative western blots showing decrease in Orai3 protein levels due to NFATc1 overexpression in SW1990 cells compared to control. (**O**) Densitometric quantitation of Orai3 protein levels in NFATc1 overexpressed SW1990 compared to control ($N = 3$). *$P = 0.0147$. (**P**) qRT-PCR analysis showing increase in Orai3 mRNA levels upon VIVIT transfection in SW1990 compared to control ($N = 3$). *$P = 0.0449$. (**Q**) Representative western blots showing increase in Orai3 protein levels due to VIVIT transfection in SW1990 cells compared to control. (**R**) Densitometric quantitation of Orai3 protein levels in VIVIT-transfected SW1990 compared to control ($N = 3$). *$P = 0.0140$. Data presented are mean ± SEM. For statistical analysis, one-sample *t* test was performed for (**A, C, D, F, G, I, J, L, M, O, P, R**) using GraphPad Prism software. Here, *$P < 0.05$ and **$P < 0.01$. Source data are available online for this figure.

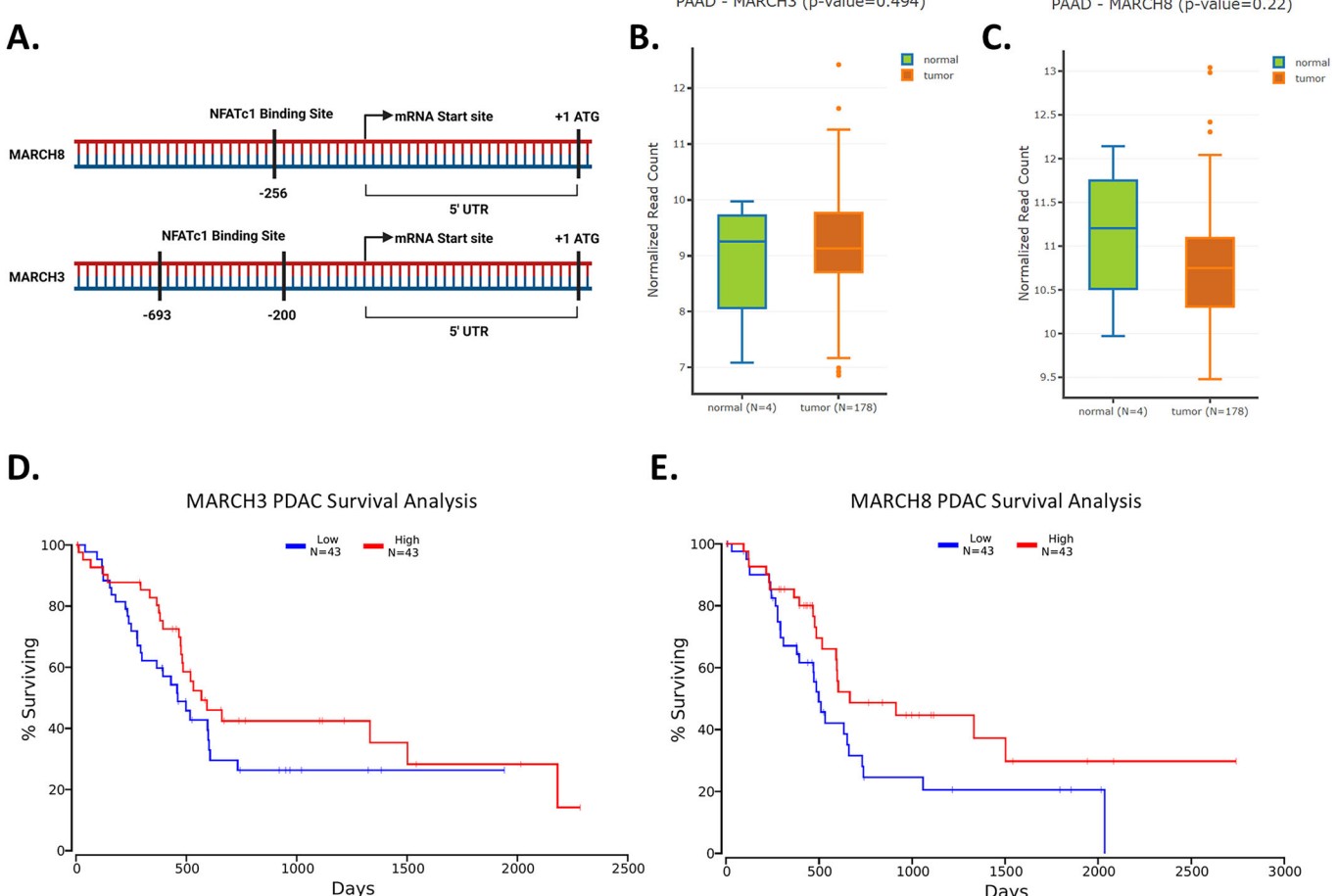

**Figure EV4. Bioinformatic analysis of MARCH3 and MARCH8 E3 ubiquitin ligase.**

(A) Identification of putative NFATc1 binding sites on the human MARCH3 and MARCH8 promoter using the EPD-Search Motif Tool at *P* value cut-off of 0.01. (B) MARCH3 expression levels in normal pancreatic tissues and pancreatic adenocarcinoma (PAAD) tissues analyzed by the DNMIVD database. (Normal: maxima-9.971; minima-7.082; median-9.252; Q1-8.061; Q3-9.718. Tumor: maxima-12.418; minima-6.856; median-9.129; Q1-8.704; Q3-9.762). (C) MARCH8 expression levels in normal pancreatic tissues and pancreatic adenocarcinoma (PAAD) tissues analyzed by the DNMIVD database. (Normal: maxima-12.14; minima-9.97; median-11.20; Q1-10.51; Q3-11.74. Tumor: maxima-13.04; minima-9.47; median-10.74; Q1-10.30; Q3-12.04). (D) Survival analysis of pancreatic cancer patients wherein blue trace corresponds to low MARCH3 expression ($n = 43$) and red trace corresponds to high MARCH3 expression ($n = 43$). (E) Survival analysis of pancreatic cancer patients wherein blue trace corresponds to low MARCH8 expression ($n = 43$) and red trace corresponds to high MARCH8 expression ($n = 43$). For statistical analysis, Mann–Whitney *U* test was performed for (B, C). Source data are available online for this figure.

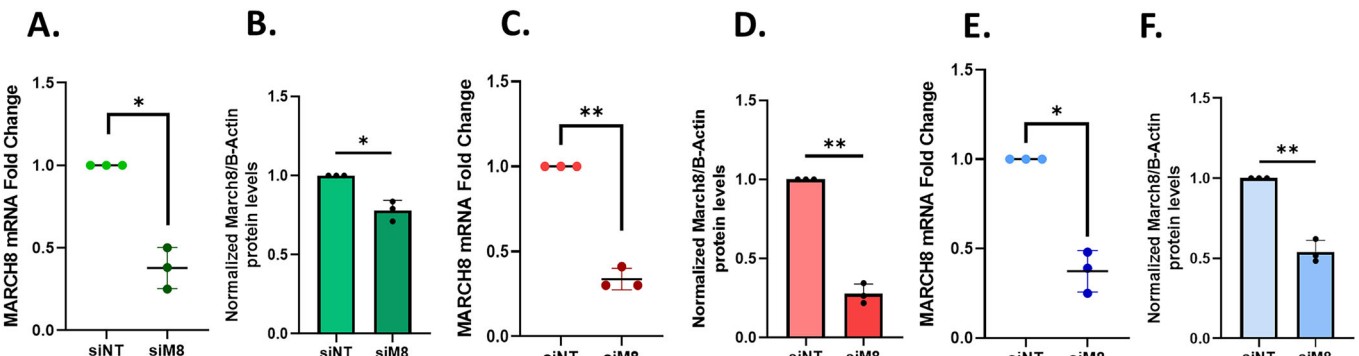

**Figure EV5.  siRNA mediated MARCH8 knockdown validation at RNA and protein levels in PC cells.**

(**A**) qRT-PCR analysis showing decrease in MARCH8 mRNA levels upon siMARCH8 transfection in MiaPaCa-2 compared to control siNT ($N = 3$). *$P = 0.0131$. (**B**) Densitometric quantitation of MARCH8 protein levels in siMARCH8 transfection MiaPaCa-2 cells compared to siNT control ($N = 3$). *$P = 0.0266$. (**C**) qRT-PCR analysis showing decrease in MARCH8 mRNA levels upon siMARCH8 transfection in PANC-1 compared to control siNT ($N = 3$). **$P = 0.0030$. (**D**) Densitometric quantitation of MARCH8 protein levels in siMARCH8 transfection in PANC-1 cells compared to siNT control ($N = 3$). **$P = 0.0025$. (**E**) qRT-PCR analysis showing decrease in MARCH8 mRNA levels upon siMARCH8 transfection in CFPAC-1 compared to control siNT ($N = 3$). *$P = 0.0112$. (**F**) Densitometric quantitation of MARCH8 protein levels in siMARCH8 transfection in CFPAC-1 cells compared to siNT control ($N = 3$). **$P = 0.0080$. Data presented are mean ± SEM. For statistical analysis, one-sample $t$ test was performed for (**A**–**F**) using GraphPad Prism software. Here, *$P < 0.05$ and **$P < 0.01$. Source data are available online for this figure.

