## [Peer Review File · The EMBO Journal]

NFATc1 drives Orai3 transcription and proteolysis by harnessing epigenome differences in the MARCH8 promoter

Sharon Raju, Akshay Sharma, Sakshi Dahiya, Gyan Ranjan, and Rajender Motiani
Corresponding author: Rajender Motiani (rajender.motiani@rcb.res.in)

Review Timeline:

Transferred from Review Commons:	16th Apr 25
Editorial Decision:	28th Apr 25
Revision Received:	7th Jul 25
Editorial Decision:	1st Aug 25
Revision Received:	19th Aug 25
Accepted:	11th Sep 25

Editor: Daniel Klimmeck

Transaction Report:

This manuscript was transferred to The EMBO Journal following peer review at Review Commons.

Review #1**1. Evidence, reproducibility and clarity:****Evidence, reproducibility and clarity (Required)**

The manuscript entitled, "NFAT2 drives both Orai3 transcription and protein degradation by harnessing the differences in epigenetic landscape of MARCH8 E3 ligase" offers an extensive study of how Orai3 levels are controlled during pancreatic cancer progression. The central hypothesis is that NFAT2 stimulates both Orai3 and MARCH8 transcription, resulting in both Orai3 transcription and degradation. They further establish that MARCH8 expression/Orai3 degradation is epigenetically regulated in PDAC, with a progressive loss of methylation during cancer progression leading to increased Orai3 transcription, stability and Ca²⁺ entry.

Overall, I'm certain that there is new information to be learned here. However, as detailed below, the manuscript makes a number of general claims about what happens during PDAC progression, but this is based on only one cell line per disease state. While they should not be expected to do a complete analysis in more cell lines, a demonstration that Orai3 and MARCH8 expression are correlated with disease progression in a panel of cell lines and/or on the TCGA database would increase enthusiasm considerably. In addition, although I found the work with MARCH8 to be highly convincing, the fact that NFAT2 knockdown increased rather than reduced Orai3 transcription does not support the central hypothesis. The explanation that this results from compensation is not very meaningful; that NFAT2 drives Orai3 transcription is in the title of the paper. These observations clearly demonstrate that this relationship is more complicated than suggested. Finally, there are a number of missing controls and unclear aspects to the authors' CHIP data that could help explain some of these discrepancies.

****Specific Comments:****

1. The observation that both transcriptional regulation and protein degradation of Orai3 is regulated downstream of one transcription factor is not, in and of itself, entirely surprising. All proteolytic components are transcriptionally regulated and this phenomenon is likely relatively common. However, what I do think is both impressive and important is that the authors have characterized both components of the pathway within a disease context. While I am not going to search the literature for how often transcription and proteolysis are co-regulated for other proteins, it is the case for many short-lived proteins and perhaps many others. As such, discussion throughout the abstract and introduction that co-

regulation of these processes is unprecedented should be removed.

2. In discussing figure 1, the authors switch from claiming to be studying NFATc binding to studying NFAT expression. This use of 2 different naming conventions is certain to confuse readers; the authors should use the approved current naming system in referring to NFAT isoforms. In which case NFAT2 is NFATc1.

3. The CHIP analyses in figures 1H and 7D are important findings, however, there is missing information. Typically, CHIP is used to validate putative binding sites; as such, one would expect 3 separate qPCR reactions for Orai3, not one. It is also important to note that qPCR products should be uniform in size and under 100 bp; here, the product size is not stated. Finally, demonstrating that an antibody targeting ANY other NFAT isoform fails to pull down whatever product this is would increase confidence considerably.

Also, the gold standard for validating CHIP is to mutate the sites and eliminate binding. The "silver" standard would be to mutate them in your luciferase vector and demonstrate that NFATc1 no longer stimulates luciferase expression. Since neither of these was done, the CHIP data provided should not be considered formally validated.

4. In figures 2 and 3, only one cell line is used to represent each of 3 conditions of pancreatic cancer. That is insufficient to make generalized conclusions; some aspects of this figure (expression and stability, not function) should be extended to 2 to 3 cell lines/condition. TCGA data validating this point would also be helpful.

5. Upon finding that NFAT inhibition stimulates Orai3 transcription (same as O/E), the authors essentially conclude that this confirms regulation of Orai3 by NFAT and that there must be compensation. This is not supported by any data; the use of siRNA validates that Orai3 has some dependence on NFATc1 for transcription, but the nature of this relationship is not adequately explained.

6. In Figure 6A,B, does the Orai3 western blot show any of the heavier bands seen in the ubiquitination IP if you show the whole blot? It should.

2. Significance:

Significance (Required)

My expertise is in calcium signaling, particularly within the context of disease states. I currently have a PDAC study in its late stages, but I have worked more with melanoma.

Issues about significance were raised in my comments above; generalization of these observations requires the appropriate use of a panel of cell lines and/or TCGA usage. In addition, some observations require additional investigation for confidence; necessary to achieve significance.

The extent of the advance is quite reasonable for a high profile paper in this field, should the issues I and the other reviewers raise be formally and thoroughly addressed.

Given that the study crosses lines between signaling, cancer, epigenetics, transcription and ubiquitination, I think that it is of potential interest to a general audience.

3. How much time do you estimate the authors will need to complete the suggested revisions:

Estimated time to Complete Revisions (Required)

(Decision Recommendation)

Between 3 and 6 months

4. Review Commons values the work of reviewers and encourages them to get credit for their work. Select 'Yes' below to register your reviewing activity at Web of Science Reviewer Recognition Service (formerly Publons); note that the content of your review will not be visible on Web of Science.

Yes

Review #2

1. Evidence, reproducibility and clarity:

Evidence, reproducibility and clarity (Required)

Raju et al. presents a nice comprehensive study of the differential regulation of Orai3 at the transcriptional and stability levels in metastatic versus non-metastatic pancreatic cancer (PC) cells. They convincingly show that NFAT2 regulates Orai3 transcription in all PC cells but interestingly, in the metastatic PC cells NFAT2 also upregulates the expression of MARCH8 an E3 ubiquitin ligase that targets Orai3 for lysosomal degradation. The MARCH8 locus is hypermethylated in the non-metastatic cell line, thus preventing MARCH8 upregulation in those cells. The data is convincing and complementary. I only a few suggestions below.

****Specific Comments:****

1. Figure 1 all overexpression no evidence of endogenous NFAT2 regulating Orai3. I realize there may be limitations on available NFAT isoform specific antibodies so it is not essential to directly show this but a comment to that effect in the paper would be useful.
2. Figure 1F. Show RNA levels of Orai3 following overexpression of the other NFAT isoforms.
3. Fig. S3D,E. For both MARCH3 and 8 higher expression levels correlate with better survival whereas in the text it is stated that this is the case only for MARCH8. Please correct.
4. For the 2APB stimulation experiments there is a large variation in the level of the response between experiments even for the same cell type. For example compare the level of the 2APB-stimulated Orai3 influx between Fig. 4H and 5C on the MiaPaCa-2 cells. Also there doesn't seem to be a correlation between the levels of Orai3 protein from WB and the 2APB stimulated entry among the different cells lines. This needs to be addressed and differences explained.
5. Fig. 6A and 6B. Show the full Orai3 and Ubiquitin WBs. As presented the figure current just shows that there are ubiquitin proteins in Orai3 pull down, not that Orai3 is ubiquitinated.
6. Fig. 6C and 6D. Show the line in 6C from which the intensity profile in 6D was generated. Also give the details of the imaging setup in methods: size of the pinhole, imaging mode, etc. The colocalization is not very convincing.
7. Also all the imaging and pull down do not prove conclusively direct interaction between MARCH8 and Orai3, they rather show that the proteins are in the same complex. Although it is unlikely best for the text to be moderated accordingly.
8. May be worth showing that overexpression of MARCH8 in the metastatic cell lines decreases their migration and metastasis as the argument is that these cells need high Orai3 but not too high. So it would be predicted that overexpression of MARCH8 should lower Orai3 levels enough to prevent their metastasis.
9. Fig. 10. Show higher levels of Orai3 protein in the metastatic side.
10. Please show all full WBs in the supplementary data.

2. Significance:

Significance (Required)

Significant and relevant study that will be of great interest to the cancer and calcium signaling fields.

3. How much time do you estimate the authors will need to complete the suggested revisions:

Estimated time to Complete Revisions (Required)

(Decision Recommendation)

Less than 1 month

4. Review Commons values the work of reviewers and encourages them to get credit for their work. Select 'Yes' below to register your reviewing activity at Web of Science Reviewer Recognition Service (formerly Publons); note that the content of your review will not be visible on Web of Science.

Yes

Review #3

1. Evidence, reproducibility and clarity:

Evidence, reproducibility and clarity (Required)

The study by Raju et al. demonstrated that NFAT2 drives both Orai3 transcription and protein degradation. They find a clearly distinct mechanism between non-metastatic cancerous and metastatic cells. While in non-metastatic cells NFAT2 drives Orai3 transcription and increases Orai3 expression, in invasive and metastatic cells degradation of Orai3 is driven. They find a physical interaction of MARCH8 with Orai3 resulting in degradation. This degradation is not happening in non-metastatic cells as MARCH8 promoter is highly methylated. This study is highly interesting for a broad readerships and provides a solid basis for the development of novel therapeutic strategies for cancer treatment. Before publication the authors should address a few minor comments.

1. The authors show that MARCH8 physically associates with Orai3 using Co-IP and Co-localization studies. For the co-localization studies the authors should still provide a quantitative analysis. Furthermore, can the authors detect FRET between March and Orai3? Can you please state the labels used in the co-localization experiments also in the figure legend.
2. In the abstract it is only getting clear at the end that pancreatic cancer cells are used. It would be great if the authors could introduce this fact already more at the beginning of the abstract
3. In the scheme in Fig. 10, the authors highlight that Orai3 is ubiquitinated. Do they have any idea where the site of action of ubiquitination in Orai3 is located?
4. In other cancer types recent reports suggest a co-expression of Orai1 and Orai3 and even the formation of heteromers. Does only Orai3 or also Orai1 play a role in pancreatic

cancer cells? Could there be a difference in degradation when Orai3 forms homomers or heteromers with Orai1.

2. Significance:

Significance (Required)

The authors highlight and decode a dual role of NFAT2 in controlling Orai3 expression, which is highly interesting to gain insight in different states of cancer cells (non-metastatic, metastatic). The findings form a great basis for a deeper understanding of potential therapeutic targets.

3. How much time do you estimate the authors will need to complete the suggested revisions:

Estimated time to Complete Revisions (Required)

(Decision Recommendation)

Less than 1 month

Yes

Revision Plan

Manuscript number: RC-2025-02850

Corresponding author(s): Dr. Rajender K Motiani

[The “revision plan” should delineate the revisions that authors intend to carry out in response to the points raised by the referees. It also provides the authors with the opportunity to explain their view of the paper and of the referee reports.]

The document is important for the editors of affiliate journals when they make a first decision on the transferred manuscript. It will also be useful to readers of the reprint and help them to obtain a balanced view of the paper.

*If you wish to submit a full revision, please use our "Full Revision" template. **It is important to use the appropriate template to clearly inform the editors of your intentions.**]*

1. General Statements [optional]

We thank all three Reviewers for appreciating our work and for sharing constructive feedback to further enhance the quality of our work. It is really gratifying to read that the Reviewers believe that this work will be of interest to broad audience and will be suitable for a high profile journal. Further, the experiments suggested by the reviewers will add value to the work and will substantiate our findings. It is important to highlight that we have already performed most of the suggested experiments except a couple of experiments that we have plan to carry out during full revision. Please find below the details of experiments performed and planned to address the reviewers comments.

2. Description of the planned revisions

Reviewer #1

Comment 6. In Figure 6A, B, does the Orai3 western blot show any of the heavier bands seen in the ubiquitination IP if you show the whole blot? It should.

Reviewer #2

Comment 5. Fig. 6A and 6B. Show the full Orai3 and Ubiquitin WBs. As presented the figure current just shows that there are ubiquitin proteins in Orai3 pull down, not that Orai3 is ubiquitinated.

Revision Plan

Reviewer #3

Comment 3. In the scheme in Fig. 10, the authors highlight that Orai3 is ubiquitinated. Do they have any idea where the site of action of ubiquitination in Orai3 is located?

Response: We thank the Reviewer 1, 2 and 3 regarding their query on the co-immunoprecipitation assays performed for studying Orai3 ubiquitination. The reviewers are asking for ubiquitination status of Orai3 and the potential sites for Orai3 ubiquitination. To address these comments, we are planning to perform co-immunoprecipitation assays with mutated Orai3 with mutations of potential ubiquitination sites. We have already performed bioinformatic analysis and it revealed presence of three potential ubiquitination sites on Orai3: K2 (present on N-terminal region), K274 and K279 (present on C-terminal region). We would mutate these lysine residues on Orai3 protein via site-directed mutagenesis and check the Orai3 ubiquitination status. These experiments will answer the question raised by Reviewers and strengthen the Orai3 ubiquitination data.

Please refer to below diagrammatic illustration showing potential ubiquitination sites on Orai3:

Reviewer #2

Comment 7. Also, all the imaging and pull down do not prove conclusively direct interaction between MARCH8 and Orai3, they rather show that the proteins are in the same complex. Although it is unlikely best for the text to be moderated accordingly.

Revision Plan

Response: We understand the concern raised by Reviewer 2 regarding direct or indirect interaction of MARCH8 and Orai3. Hence, we are planning to perform co-immunoprecipitation assays in which we delete the MARCH8 interacting domain in Orai3 protein and check the for direct interaction of these proteins. Bioinformatic analysis and literature survey have highlighted two possible MARCH8 interacting domains in Orai3. The first domain is present in 2nd loop region, present between the 2nd and 3rd transmembrane domains at the LMVXXXL (AA113-120) motif and the second domain is present at the GXXXG (AA235-239) motif, present in the 3rd loop region of Orai3. We will remove these domains from Orai3 protein individually and check its effect on MARCH8 interaction. These experiments will provide conclusive evidence of direct interaction between Orai3 and MARCH8.

Please refer to below diagrammatic illustration displaying potential MARCH8 binding sites on Orai3:

3. Description of the revisions that have already been incorporated in the transferred manuscript

Reviewer #1

Comment 1. The observation that both transcriptional regulation and protein degradation of Orai3 is regulated downstream of one transcription factor is not, in and of itself, entirely surprising. All proteolytic components are transcriptionally regulated and this phenomenon is likely relatively common. However, what I do think is both impressive and important is that the

Revision Plan

authors have characterized both components of the pathway within a disease context. While I am not going to search the literature for how often transcription and proteolysis are co-regulated for other proteins, it is the case for many short-lived proteins and perhaps many others. As such, discussion throughout the abstract and introduction that co-regulation of these processes is unprecedented should be removed.

Response: We thank the Reviewer for thinking that our work is both impressive and important. Further, we understand the Reviewer's point that transcription and proteolysis may be co-regulated for other proteins. However, our extensive literature search did not result in such scenarios. Therefore, to best of our knowledge, we are revealing for the first time that same transcription factor regulates both transcription and protein degradation of the same target in a context dependent manner in a single study. In case, Reviewer would still recommend to modify the text in abstract and introduction, we would do it.

Comment 2. In discussing figure 1, the authors switch from claiming to be studying NFATc binding to studying NFAT expression. This use of 2 different naming conventions is certain to confuse readers; the authors should use the approved current naming system in referring to NFAT isoforms. In which case NFAT2 is NFATc1.

Response: We would like to thank the Reviewer for highlighting this point. We have effectively addressed this comment by changing the nomenclature of NFAT2 to NFATc1 throughout the manuscript text and figures.

Comment 3. The ChIP analyses in figures 1H and 7D are important findings, however, there is missing information. Typically, ChIP is used to validate putative binding sites; as such, one would expect 3 separate qPCR reactions for Orai3, not one. It is also important to note that qPCR products should be uniform in size and under 100 bp; here, the product size is not stated. Finally, demonstrating that an antibody targeting ANY other NFAT isoform fails to pull down whatever product this is would increase confidence considerably.

Also, the gold standard for validating ChIP is to mutate the sites and eliminate binding. The "silver" standard would be to mutate them in your luciferase vector and demonstrate that NFATc1 no longer stimulates luciferase expression. Since neither of these was done, the ChIP data provided should not be considered formally validated.

Response: We thank the Reviewer for raising this highly relevant concern. In this revised manuscript, we have addressed this comment by performing several additional experiments.

Revision Plan

The new data provided in the revised manuscript corroborates our earlier results. Indeed, this data has notably strengthened our work.

In the revised manuscript, we performed ChIP assay where we increased the number of sonication cycles to 35 so as to make sheared chromatin of around 100 bp. Next, we designed primers to amplify individual NFATc1 binding sites on Orai3 promoter, but due to close proximity of the NFATc1 binding sites, we could design two primer sets. The primer first set to amplify the -1017 bp binding site and the second set to amplify the -990 and -920 bp. Further, as suggested by the Reviewer, we performed immunoprecipitation with the four isoforms of NFAT. Our results show that only NFATc1 pulldown shows significant enrichment of Orai3 promoter with both the primer sets as compared to the IP mock samples and other NFAT isoforms (**Figure 1J**). Hence, our data reveals that only NFATc1 binds to these predicted sites on the Orai3 promoter and it doesn't show a preference among these binding sites.

Further, as suggested by the Reviewer, we mutated the Orai3 promoter in luciferase vector with deletions of the individual NFATc1 binding sites and also cloned a truncated Orai3 promoter with no NFATc1 binding sites into the luciferase vector. The luciferase assays with these mutant and truncated promoters show that upon co-expression of NFATc1, the luciferase activity of the mutant Orai3 promoter with deletion of individual NFATc1 binding site is significantly reduced in comparison to wild type Orai3 promoter. Furthermore, the maximum decrease in luciferase activity was seen with the truncated Orai3 promoter with no NFATc1 binding sites (**Figure 1I**). These results show that NFATc1 binds to the predicted binding sites on Orai3 promoter. Taken together, the additional ChIP assays with the four isoforms of NFAT and luciferase assays with mutated & truncated Orai3 promoters validates the transcriptional regulation of Orai3 by NFATc1.

Comment 4. In figures 2 and 3, only one cell line is used to represent each of 3 conditions of pancreatic cancer. That is insufficient to make generalized conclusions; some aspects of this figure (expression and stability, not function) should be extended to 2 to 3 cell lines/condition. TCGA data validating this point would also be helpful.

Response: We really appreciate the feedback given by Reviewer 1. To strengthen our manuscript, we have addressed this comment by performing experiments in 2 cell

Revision Plan

lines/condition of pancreatic cancer. This new data in the revised manuscript provides substantial evidence for the dichotomous regulation of Orai3 by NFATc1.

In the revised manuscript, we carried out NFATc1 overexpression and NFAT inhibition via VIVIT studies in three additional cell lines: BXPC-3 (non-metastatic), ASPC-1 (invasive) and SW1990 (metastatic). The results in these cell-lines support our earlier findings as both overexpression of NFATc1 and VIVIT mediated NFAT inhibition leads to transcriptional upregulation of Orai3 in BXPC-3 (non-metastatic) (**Figure S3A, D**), ASPC-1 (invasive) (**Figure S3G, J**) and SW1990 (metastatic) (**Figure S3M, P**). These results are similar to our earlier data from MiaPaCa-2 (non-metastatic), PANC-1 (invasive) and CFPAC-1 (metastatic) cells. Further, NFATc1 overexpression leads to an increase in Orai3 protein levels in BXPC-3 (non-metastatic) (**Figure S3B, C**) and a decrease in Orai3 protein levels in ASPC-1 (invasive) (**Figure S3H, I**) and SW1990 (metastatic) (**Figure S3N, O**). Moreover, VIVIT transfection leads to a decrease in Orai3 protein levels in BXPC-3 (non-metastatic) (**Figure S3E, F**) and an increase in Orai3 protein levels in ASPC-1 (invasive) (**Figure S3K, L**) and SW1990 (metastatic) (**Figure S3Q, R**). The findings in these cell lines recapitulates the data obtained earlier from MiaPaCa-2 (non-metastatic), PANC-1 (invasive) and CFPAC-1 (metastatic) cell lines. Therefore, this new data supports our conclusion regarding the dichotomous regulation of Orai3 by NFATc1 across the three conditions of pancreatic cancer.

Comment 5. Upon finding that NFAT inhibition stimulates Orai3 transcription (same as O/E), the authors essentially conclude that this confirms regulation of Orai3 by NFAT and that there must be compensation. This is not supported by any data; the use of siRNA validates that Orai3 has some dependence on NFATc1 for transcription, but the nature of this relationship is not adequately explained.

Response: We thank the Reviewer for asking this question. In our manuscript, we performed NFATc1 inhibition studies using VIVIT and siRNA-mediated NFATc1 knockdown. Both of these assays show increase in Orai3 mRNA levels in all non-metastatic, invasive and metastatic pancreatic cancer cell lines. To understand if the increase in Orai3 mRNA levels is via transcriptional regulation, we performed luciferase assay which showed that VIVIT mediated NFAT inhibition leads to increase in luciferase activity suggesting the binding of other transcription factors on the Orai3 promoter. To corroborate this hypothesis, in our revised manuscript, we performed luciferase assay in wild type Orai3 promoter and truncated Orai3

Revision Plan

promoter with no NFATc1 binding sites. NFAT inhibition via VIVIT transfection led to an increase in luciferase activity in both wild type and truncated Orai3 promoter (**Figure S2A**). Hence, removal of NFATc1 binding sites had no significant effect on luciferase activity suggesting that apart from NFATc1, other endogenous transcription factors are involved in regulating Orai3 transcription. We have not identified all the transcription factors that can modulate Orai3 upon NFAT inhibition as it is beyond the scope of this study. We sincerely hope the Reviewer 1 would be satisfied with this additional data.

Reviewer #2

Comment 1. Figure 1 all overexpression no evidence of endogenous NFAT2 regulating Orai3. I realize there may be limitations on available NFAT isoform specific antibodies so it is not essential to directly show this but a comment to that effect in the paper would be useful.

Response: We apologize to the Reviewer for not highlighting the NFAT2 (NFATc1) loss of function data effectively. Actually, in the **Figure 3** and **Supplementary Figure 2** of the original manuscript, we showed VIVIT mediated NFAT inhibition and siRNA induced NFATc1 silencing data to provide the evidence that endogenous NFATc1 regulates Orai3.

Comment 2. Figure 1F. Show RNA levels of Orai3 following overexpression of the other NFAT isoforms.

Response: As suggested by the Reviewer, in the revised manuscript, we overexpressed the four NFAT isoforms: NFATc2, NFATc1, NFATc4 & NFATc3 and checked Orai3 mRNA levels. qRT-PCR analysis shows that overexpression of NFATc1 results in the highest and significant increase in Orai3 mRNA levels compared to the empty vector and other NFAT isoforms (**Figure 1F**). This data corroborates the western blot data of NFAT isoforms overexpression highlighting the transcriptional regulation of Orai3 by NFATc1.

Comment 3. Fig. S3D, E. For both MARCH3 and 8 higher expression levels correlate with better survival whereas in the text it is stated that this is the case only for MARCH8. Please correct.

Response: The survival analysis of pancreatic cancer patients with low MARCH3 and MARCH8 levels shows that around 30% of patients with low MARCH3 levels survived for <5.5 years, while 0% of patients with low MARCH8 levels survived for <5.5 years indicating that low

Revision Plan

MARCH8 levels in pancreatic cancer patients are associated with decreased survival rate as compared to patients with low MARCH3 levels. Additionally, the survival analysis of pancreatic cancer patients with high MARCH3 and MARCH8 levels show that less than 15% of patients with high MARCH3 levels survived for >5.5 years, whereas in case of MARCH8 30% of patients with high MARCH8 levels survived for >7.5 years. Hence high MARCH8 expression in pancreatic cancer patients provided significant survival advantage compared to high MARCH3 levels. Therefore, in the text, we meant that compared to MARCH3, higher MARCH8 levels correlate with better survival. As suggested by the Reviewer, we have modified the text to make this point clearer.

Comment 4. For the 2APB stimulation experiments there is a large variation in the level of the response between experiments even for the same cell type. For example, compare the level of the 2APB-stimulated Orai3 influx between Fig. 4H and 5C on the MiaPaCa-2 cells. Also there doesn't seem to be a correlation between the levels of Orai3 protein from WB and the 2APB stimulated entry among the different cell lines. This needs to be addressed and differences explained.

Response: We understand the concern raised by Reviewer 2 regarding calcium imaging experiments in MiaPaCa-2 cell line. Therefore, in the revised manuscript, we repeated calcium imaging experiments in MiaPaCa-2 and updated the representative traces as well as quantitative analysis (**Figure 2D, E, 3D, E, 4H, I, S2L, M**). Further, we have discussed this point in the text of the manuscript.

Comment 6. Fig. 6C and 6D. Show the line in 6C from which the intensity profile in 6D was generated. Also give the details of the imaging setup in methods: size of the pinhole, imaging mode, etc. The colocalization is not very convincing.

Response: As recommended by the Reviewer, in the revised manuscript, we have indicated the region used for intensity profile generation by drawing a line in the representative image (**Figure 6D**). Further, we have updated the methodology of colocalization microscopy with details of the size of the pinhole and imaging mode.

Comment 8. May be worth showing that overexpression of MARCH8 in the metastatic cell lines decreases their migration and metastasis as the argument is that these cells need high Orai3 but not too high. So, it would be predicted that overexpression of MARCH8 should lower Orai3 levels enough to prevent their metastasis.

Revision Plan

Response: We would like to thank the Reviewer for this highly relevant suggestion. In our revised manuscript, we carried out transwell migration assays with MARCH8 overexpression as well as MARCH8 knockdown in CFPAC-1 (metastatic) cells. Our data shows that stable lentiviral knockdown of MARCH8 increased the number of migrated CFPAC-1 cells compared to shNT CFPAC-1 cells while MARCH8 overexpression decreased the number of migrated CFPAC-1 cells compared to empty vector control cells (**Figure 9F, G**). Therefore, as pointed out by the Reviewer, MARCH8 overexpression lowers Orai3 levels in metastatic pancreatic cancer cells and hinders their metastatic potential.

Comment 9. Fig. 10. Show higher levels of Orai3 protein in the metastatic side.

Response: As suggested, we have updated the summary figure (**Figure 10**) showing higher Orai3 protein levels in the metastatic side.

Comment 10. Please show all full WBs in the supplementary data.

Response: As recommended by the Reviewer, we have provided all full western blots in a supplementary file (**Supplementary File 1**).

Reviewer #3

Comment 1. The authors show that MARCH8 physically associates with Orai3 using Co-IP and Co-localization studies. For the co-localization studies the authors should still provide a quantitative analysis. Furthermore, can the authors detect FRET between March and Orai3? Can you please state the labels used in the co-localization experiments also in the figure legend.

Response: As suggested by Reviewer 3, in the revised manuscript, we have provided quantitative analysis of Orai3 and MARCH8 co-localization. Further, we have stated the labels used in the co-localization experiment in the figure legend of the revised manuscript. Unfortunately, we could not perform FRET assay between Orai3 and MARCH8 due to limited resources. Instead, as discussed in the planned revisions section, we are planning to perform co-immunoprecipitation assay with mutated Orai3 protein in which the MARCH8 interacting

Revision Plan

domains are deleted to investigate direct interaction of Orai3 and MARCH8. We believe that Reviewer 3 will be satisfied with this experiment.

Comment 2. In the abstract it is only getting clear at the end that pancreatic cancer cells are used. It would be great if the authors could introduce this fact already more at the beginning of the abstract.

Response: As recommended by the Reviewer, in the revised manuscript, we have introduced the use of pancreatic cancer cells at the beginning of the abstract.

Comment 4. In other cancer types recent reports suggest a co-expression of Orai1 and Orai3 and even the formation of heteromers. Does only Orai3 or also Orai1 play a role in pancreatic cancer cells? Could there be difference in degradation when Orai3 forms homomers or heteromers with Orai1.

Response: We thank the reviewer for asking this interesting question. There is only one report on Orai1's role in pancreatic cancer. It was suggested that Orai1 can contribute to apoptotic resistance of pancreatic cancer cells (Kondratska et al. BBA-Molecular Cell Research, 2014). However, only one cell line i.e. PANC-1 was used in this study. While our earlier work and other studies have demonstrated that Orai3 drives pancreatic cancer metastasis (Arora et al. Cancers, 2021) and proliferation (Dubois et al. BBA-Molecular Cell Research, 2021) respectively. Therefore, emerging literature suggests that both Orai1 and Orai3 can contribute to different aspects of pancreatic cancer progression. But whether Orai1 and Orai3 form heteromers in pancreatic cancer cells remains unexplored. Further, we believe that the degradation machinery and the underlying molecular mechanisms would be analogous for both Orai3 homomers and heteromers. Nonetheless, the rate of degradation may differ for Orai3 homomers and heteromers as literature suggests that usually proteins are more stable in large heteromeric protein complexes.

Dear Dr Motiani,

Thank you for transferring your manuscript EMBOJ-2025-121095-T | [RC-2025-02850] with Review Commons referee reports and your referee response to The EMBO Journal, and also for your patience with our feedback at this time of the year. I have now carefully assessed your revision plan and rebuttal, and in addition discussed this in the editorial team.

We acknowledge that you have carried out and are in addition planning a substantial experimental revision to complement your study and address the critique raised by the referees and consider your response to be sensible. We can thus invite you to revise your study along the lines sketched in your outline for further consideration by the EMBO Journal.

Please note however that the generality and technical concerns by referee #1 (Ref#1, pts.3-6) are unambiguous and need to be completed to the reviewer's satisfaction.

Please feel free to contact me if you have any questions or need further input on the referee comments.

When submitting your revised manuscript, please carefully review the instructions below.

Please feel free to approach me any time should you have additional questions related to this.

Thank you for the opportunity to consider your work for publication.

I look forward to your revision.

Kind regards,

Daniel Klimmeck

Daniel Klimmeck, PhD
Senior Editor
The EMBO Journal

Instruction for the preparation of your revised manuscript:

- 1) a .docx formatted version of the manuscript text (including legends for main figures, EV figures and tables). Please make sure that the changes are highlighted to be clearly visible.
- 2) individual production quality figure files as .eps, .tif, .jpg (one file per figure).
- 3) a .docx formatted letter INCLUDING the reviewers' reports and your detailed point-by-point response to their comments. As part of the EMBO Press transparent editorial process, the point-by-point response is part of the Review Process File (RPF), which will be published alongside your paper.
- 4) a complete author checklist, which you can download from our author guidelines ([https://wol-prod-cdn.literatumonline.com/pb-assets/embo-site/Author Checklist%20-%20EMBO%20J-1561436015657.xlsx](https://wol-prod-cdn.literatumonline.com/pb-assets/embo-site/Author%20Checklist%20-%20EMBO%20J-1561436015657.xlsx)). Please insert information in the checklist that is also reflected in the manuscript. The completed author checklist will also be part of the RPF.
- 5) Please note that all corresponding authors are required to supply an ORCID ID for their name upon submission of a revised manuscript.
- 6) It is mandatory to include a 'Data Availability' section after the Materials and Methods. Before submitting your revision, primary datasets produced in this study need to be deposited in an appropriate public database, and the accession numbers and database listed under 'Data Availability'. Please remember to provide a reviewer password if the datasets are not yet public (see

<https://www.embopress.org/page/journal/14602075/authorguide#datadeposition>).

7) Our journal encourages inclusion of *data citations in the reference list* to directly cite datasets that were re-used and obtained from public databases. Data citations in the article text are distinct from normal bibliographical citations and should directly link to the database records from which the data can be accessed. In the main text, data citations are formatted as follows: "Data ref: Smith et al, 2001" or "Data ref: NCBI Sequence Read Archive PRJNA342805, 2017". In the Reference list, data citations must be labeled with "[DATASET]". A data reference must provide the database name, accession number/identifiers and a resolvable link to the landing page from which the data can be accessed at the end of the reference. Further instructions are available at .

8) At EMBO Press we ask authors to provide source data for the main and EV figures. Our source data coordinator will contact you to discuss which figure panels we would need source data for and will also provide you with helpful tips on how to upload and organize the files.

Numerical data can be provided as individual .xls or .csv files (including a tab describing the data). For 'blots' or microscopy, uncropped images should be submitted (using a zip archive or a single pdf per main figure if multiple images need to be supplied for one panel). Additional information on source data and instruction on how to label the files are available at .

9) We replaced Supplementary Information with Expanded View (EV) Figures and Tables that are collapsible/expandable online (see examples in <https://www.embopress.org/doi/10.15252/emj.201695874>). A maximum of 5 EV Figures can be typeset. EV Figures should be cited as 'Figure EV1, Figure EV2' etc. in the text and their respective legends should be included in the main text after the legends of regular figures.

11) For data quantification: please specify the name of the statistical test used to generate error bars and P values, the number (n) of independent experiments (specify technical or biological replicates) underlying each data point and the test used to calculate p-values in each figure legend. The figure legends should contain a basic description of n, P and the test applied. Graphs must include a description of the bars and the error bars (s.d., s.e.m.).

We realize that it is difficult to revise to a specific deadline. In the interest of protecting the conceptual advance provided by the work, we recommend a revision within 3 months (27th Jul 2025). Please discuss the revision progress ahead of this time with the editor if you require more time to complete the revisions.

Response to Reviewers' comments

We acknowledge that you have carried out and are in addition planning a substantial experimental revision to complement your study and address the critique raised by the referees and consider your response to be sensible. We can thus invite you to revise your study along the lines sketched in your outline for further consideration by the EMBO Journal.

We thank the Editor, the EMBO Editorial team and all three Reviewers for appreciating our work and for sharing constructive feedback to further enhance the quality of our work. It is really gratifying to read that the Reviewers believe that this work will be of interest to broad audience and will be suitable for a high profile journal. Further, the experiments suggested by the reviewers have added value to the work and have substantiated our findings. Moreover, we thank the EMBO Editorial team for considering our response/revision plan to be sensible.

It is important to highlight that we have performed all the suggested experiments. Additionally, we have performed extensive site directed mutagenesis and deletions studies coupled with co-immunoprecipitation assays to reveal that MARCH8 physically interacts with the Orai3 intracellular loop (between transmembrane domain two and three) to eventually ubiquitinate K2 residue within Orai3 N-terminal. Please find below the detailed point wise point response to the reviewers' comments.

Reviewer #1:

The manuscript entitled, "NFAT2 drives both Orai3 transcription and protein degradation by harnessing the differences in epigenetic landscape of MARCH8 E3 ligase" offers an extensive study of how Orai3 levels are controlled during pancreatic cancer progression. The central hypothesis is that NFAT2 stimulates both Orai3 and MARCH8 transcription, resulting in both Orai3 transcription and degradation. They further establish that MARCH8 expression/Orai3 degradation is epigenetically regulated in PDAC, with a progressive loss of methylation during cancer progression leading to increased Orai3 transcription, stability and Ca²⁺ entry.

Overall, I am certain that there is new information to be learned here. However, as detailed below, the manuscript makes a number of general claims about what happens during PDAC progression, but this is based on only one cell line per disease state. While they should not be expected to do a complete analysis in more cell lines, a demonstration that Orai3 and MARCH8 expression are correlated with disease progression in a panel of cell lines and/or on the TCGA database would increase enthusiasm considerably. In addition, although I found the work with

MARCH8 to be highly convincing, the fact that NFAT2 knockdown increased rather than reduced Orai3 transcription does not support the central hypothesis. The explanation that this results from compensation is not very meaningful; that NFAT2 drives Orai3 transcription is in the title of the paper. These observations clearly demonstrate that this relationship is more complicated than suggested. Finally, there are a number of missing controls and unclear aspects to the authors' CHIP data that could help explain some of these discrepancies.

Response: We sincerely appreciate the Reviewer's critical feedback and believing that there is new information and novelty in this study. We further thank the Reviewer for finding our work highly convincing. We have carried out a substantial number of new experiments in the revised manuscript to answer the few questions asked by the Reviewer.

As suggested by the Reviewer, in the revised manuscript we have demonstrated the correlation between Orai3 and MARCH8 expression in a panel of six cell lines. Further, we have performed additional experiments to explain how NFAT2 knockdown can potentially increase Orai3 transcription. Moreover, as recommended by the Reviewer, we have carried out additional ChIP and luciferase assays along with appropriate controls. Taken together, we have extensively updated our manuscript with additional data to address the concerns raised by the Reviewer. We sincerely hope that the Reviewer would be satisfied with the new data and our explanations.

Comment 1: The observation that both transcriptional regulation and protein degradation of Orai3 is regulated downstream of one transcription factor is not, in and of itself, entirely surprising. All proteolytic components are transcriptionally regulated and this phenomenon is likely relatively common. However, what I do think is both impressive and important is that the authors have characterized both components of the pathway within a disease context. While I am not going to search the literature for how often transcription and proteolysis are co-regulated for other proteins, it is the case for many short-lived proteins and perhaps many others. As such, discussion throughout the abstract and introduction that co-regulation of these processes is unprecedented should be removed.

Response: We thank the Reviewer for thinking that our work is both impressive and important. Further, we understand the Reviewer's point that transcription and proteolysis may be co-regulated for other proteins. However, our extensive literature search did not result in such scenarios. Therefore, to best of our knowledge, we are revealing for the first time that same transcription factor regulates both transcription and protein degradation of the same target in a context dependent manner in a single study. We believe it is important to highlight this point as it would be of interest to the EMBO Journal audience. In case, Reviewer would still recommend to modify the text in abstract and introduction, we would do it.

Comment 2: In discussing figure 1, the authors switch from claiming to be studying NFATc binding to studying NFAT expression. This use of 2 different naming conventions is certain to confuse readers; the authors should use the approved current naming system in referring to NFAT isoforms. In which case NFAT2 is NFATc1.

Response: We would like to thank the Reviewer for highlighting this point. We have effectively addressed this comment by changing the nomenclature of NFAT2 to NFATc1 throughout the manuscript text and figures.

Comment 3: The ChIP analyses in figures 1H and 7D are important findings, however, there is missing information. Typically, ChIP is used to validate putative binding sites; as such, one would expect 3 separate qPCR reactions for Orai3, not one. It is also important to note that qPCR products should be uniform in size and under 100 bp; here, the product size is not stated. Finally, demonstrating that an antibody targeting ANY other NFAT isoform fails to pull down whatever product this is would increase confidence considerably.

Also, the gold standard for validating ChIP is to mutate the sites and eliminate binding. The "silver" standard would be to mutate them in your luciferase vector and demonstrate that NFATc1 no longer stimulates luciferase expression. Since neither of these was done, the ChIP data provided should not be considered formally validated.

Response: We thank the Reviewer for raising this highly relevant concern. In this revised manuscript, we have addressed this comment by performing several additional experiments. The new data provided in the revised manuscript corroborates our earlier results. Indeed, this data has notably strengthened our work.

In the revised manuscript, we performed ChIP assay where we increased the number of sonication cycles to 35 to make sheared chromatin of around 100 bp. Next, we designed primers to amplify individual NFATc1 binding sites on Orai3 promoter, but due to close proximity of the NFATc1 binding sites, we could design two primer sets. The primer first set to amplify the -1017 bp binding site and the second set to amplify the -990 and -920 bp. Next, as suggested by Reviewer 1, we performed immunoprecipitation with the four isoforms of NFAT. Our results show that only NFATc1 pulldown shows significant enrichment of Orai3 promoter with both the primer sets as compared to the IP mock samples and other NFAT isoforms (**Figure 1J**). Hence, our data reveals that only NFATc1 binds to these predicted sites on the Orai3 promoter and doesn't show a preference among these binding sites.

Further, as recommended by the Reviewer, we mutated the Orai3 promoter in luciferase vector with deletions of the individual NFATc1 binding sites and also cloned a truncated Orai3 promoter with no NFATc1 binding sites into the luciferase vector. The luciferase assays with these mutant and truncated promoters show that upon co-expression of NFATc1, the luciferase activity of the mutant Orai3 promoter with deletion of individual NFATc1 binding site is significantly reduced in comparison to wild type Orai3 promoter. Furthermore, the maximum decrease in luciferase activity was seen with the truncated Orai3 promoter with no NFATc1 binding sites (**Figure 1I**). These results show that NFATc1 binds to the predicted binding sites on Orai3 promoter. Taken together, the additional ChIP assays with the four isoforms of NFAT and luciferase assays with mutated & truncated Orai3 promoters validates the transcriptional regulation of Orai3 by NFATc1.

Comment 4. In figures 2 and 3, only one cell line is used to represent each of 3 conditions of pancreatic cancer. That is insufficient to make generalized conclusions; some aspects of this figure (expression and stability, not function) should be extended to 2 to 3 cell lines/condition. TCGA data validating this point would also be helpful.

Response: We really appreciate the feedback given by the Reviewer. To strengthen our manuscript, we have addressed this comment by performing experiments in two cell lines/condition. This new data in the revised manuscript provides substantial evidence for the dichotomous regulation of Orai3 by NFATc1.

In the updated manuscript, we carried out NFATc1 overexpression and NFAT inhibition via VIVIT studies in three additional pancreatic cancer cell lines: BXPC-3 (non-metastatic), ASPC-1 (invasive) and SW1990 (metastatic). The results in these cell lines support our earlier findings as both overexpression of NFATc1 and VIVIT mediated NFAT inhibition leads to transcriptional upregulation of Orai3 in BXPC-3 (non-metastatic) (**Figure EV3A, D**), ASPC-1 (invasive) (**Figure EV3G, J**) and SW1990 (metastatic) (**Figure EV3M, P**). These results are similar to our earlier data from MiaPaCa-2 (non-metastatic), PANC-1 (invasive) and CFPAC-1 (metastatic) cells. Further, NFATc1 overexpression leads to an increase in Orai3 protein levels in BXPC-3 (non-metastatic) (**Figure EV3B, C**) and a decrease in Orai3 protein levels in ASPC-1 (invasive) (**Figure EV3H, I**) and SW1990 (metastatic) (**Figure EV3N, O**). Moreover, VIVIT transfection leads to a decrease in Orai3 protein levels in BXPC-3 (non-metastatic) (**Figure EV3E, F**) and an increase in Orai3 protein levels in ASPC-1 (invasive) (**Figure EV3K, L**) and SW1990

(metastatic) (**Figure EV3Q, R**). The findings in these cell lines recapitulates the data obtained earlier from MiaPaCa-2 (non-metastatic), PANC-1 (invasive) and CFPAC-1 (metastatic) cell lines. Therefore, our new data supports our conclusion regarding the dichotomous regulation of Orai3 by NFATc1 across the different stages of pancreatic cancer.

Comment 5: Upon finding that NFAT inhibition stimulates Orai3 transcription (same as O/E), the authors essentially conclude that this confirms regulation of Orai3 by NFAT and that there must be compensation. This is not supported by any data; the use of siRNA validates that Orai3 has some dependence on NFATc1 for transcription, but the nature of this relationship is not adequately explained.

Response: We thank the Reviewer for asking this question. In our manuscript, we performed NFATc1 inhibition studies using VIVIT and siRNA-mediated NFATc1 knockdown. Both of these assays show an increase in Orai3 mRNA levels in all non-metastatic, invasive and metastatic pancreatic cancer cell lines. To understand if the increase in Orai3 mRNA levels is via transcriptional regulation, we performed luciferase assays which showed that VIVIT mediated NFAT inhibition leads to increase in luciferase activity suggesting the binding of other transcription factors on the Orai3 promoter. To corroborate this hypothesis, in our revised manuscript, we performed luciferase assays with wild type Orai3 promoter and truncated Orai3 promoter with no NFATc1 binding sites. In these assays, NFAT inhibition via VIVIT transfection led to an increase in luciferase activity in both wild type and truncated Orai3 promoter (**Figure EV2A**). This implicates that in conditions of NFATc1 silencing or inhibition, other endogenous transcription factors are involved in regulating Orai3 transcription. We have not identified all the transcription factors that can modulate Orai3 upon NFAT inhibition as it is beyond the scope of this study. We sincerely hope the Reviewer would be satisfied with this additional data.

Comment 6: In Figure 6A, B, does the Orai3 western blot show any of the heavier bands seen in the ubiquitination IP if you show the whole blot? It should.

Response: As recommended by the Reviewer, the full western blot is included in the “Source Data” file of the manuscript. The whole Orai3 western blot does not clearly show the heavier polyubiquitinated species seen in the ubiquitination IP. This is likely due to their relatively lower abundance and the sensitivity limits of the Orai3 antibody under standard western blot conditions. In contrast, the ubiquitin IP enriches for polyubiquitinated proteins, making the modified Orai3 species more readily detectable. Indeed, our literature survey suggests that it is

not surprising if polyubiquitinated proteins are not observed along with target protein in the same blot due to the points discussed above.

In the revised manuscript, we have performed several additional experiments that clearly demonstrate Orai3 ubiquitination by MARCH8. Moreover, we have identified the precise site of Orai3 ubiquitination as well as MARCH8 interaction domain. We sincerely hope that the Reviewer would be satisfied with our new data that supports Orai3 ubiquitination by MARCH8.

Reviewer #2

Raju et al. presents a nice comprehensive study of the differential regulation of Orai3 at the transcriptional and stability levels in metastatic versus non-metastatic pancreatic cancer (PC) cells. They convincingly show that NFAT2 regulates Orai3 transcription in all PC cells but interestingly, in the metastatic PC cells NFAT2 also upregulates the expression of MARCH8 an E3 ubiquitin ligase that targets Orai3 for lysosomal degradation. The MARCH8 locus is hypermethylated in the non-metastatic cell line, thus preventing MARCH8 upregulation in those cells. The data is convincing and complementary. I only a few suggestions below.

Response: We thank the Reviewer for appreciating our study and believing that our data is convincing & complementary. Further, we thank him/her for insightful suggestions to further enhance the quality of our manuscript. In this revision, we have carried out all the suggested experiments. Please find below the detailed point wise response to the Reviewers' comments.

Comment 1: Figure 1 all overexpression no evidence of endogenous NFAT2 regulating Orai3. I realize there may be limitations on available NFAT isoform specific antibodies so it is not essential to directly show this but a comment to that effect in the paper would be useful.

Response: We apologize to the Reviewer for not highlighting the NFAT2 (NFATc1) loss of function data effectively. Actually, in the **Figure 3** and **Expanded View Figure 2** of the original manuscript, we showed VIVIT mediated NFAT inhibition and siRNA induced NFATc1 silencing data to provide the evidence that endogenous NFATc1 regulates Orai3. Further, in the revised manuscript, we have performed luciferase assays with the NFAT isoforms (NFATc1, NFATc2, NFATc3 and NFATc4) and show that only NFATc1 drives Orai3 transcription (**Figure 1J**). Taken together, our data clearly demonstrates that the endogenous NFATc1 regulates Orai3 transcription.

Comment 2: Figure 1F. Show RNA levels of Orai3 following overexpression of the other NFAT isoforms.

Response: As suggested by the Reviewer, in the revised manuscript, we overexpressed the four NFAT isoforms: NFATc1, NFATc2, NFATc3 & NFATc4 and checked Orai3 mRNA levels. qRT-PCR analysis shows that only overexpression of NFATc1 results in the significant increase in Orai3 mRNA levels compared to the empty vector control (**Figure 1F**). This data corroborates the western blot data of NFAT isoforms overexpression highlighting the transcriptional regulation of Orai3 by NFATc1.

Comment 3: Fig. S3D, E. For both MARCH3 and 8 higher expression levels correlate with better survival whereas in the text it is stated that this is the case only for MARCH8. Please correct.

Response: The survival analysis of pancreatic cancer patients with low MARCH3 and MARCH8 levels shows that around 30% of patients with low MARCH3 levels survived for <5.5 years, while 0% of patients with low MARCH8 levels survived for <5.5 years indicating that low MARCH8 levels in pancreatic cancer patients are associated with decreased survival rate as compared to patients with low MARCH3 levels. Additionally, the survival analysis of pancreatic cancer patients with high MARCH3 and MARCH8 levels show that less than 15% of patients with high MARCH3 levels survived for >5.5 years, whereas in case of MARCH8 30% of patients with high MARCH8 levels survived for >7.5 years. Hence high MARCH8 expression in pancreatic cancer patients provided significant survival advantage compared to high MARCH3 levels. Therefore, in the text, we meant that compared to MARCH3, higher MARCH8 levels correlate with better survival. As suggested by the Reviewer, we have modified the text to make this point clearer.

Comment 4: For the 2APB stimulation experiments there is a large variation in the level of the response between experiments even for the same cell type. For example, compare the level of the 2APB-stimulated Orai3 influx between Fig. 4H and 5C on the MiaPaCa-2 cells. Also there doesn't seem to be a correlation between the levels of Orai3 protein from WB and the 2APB stimulated entry among the different cell lines. This needs to be addressed and differences explained.

Response: We understand the concern raised by Reviewer 2 regarding inconsistencies in calcium imaging experiments in MiaPaCa-2 cell line. Therefore, in the revised manuscript, we repeated calcium imaging experiments in MiaPaCa-2 and updated the representative traces as

well as quantitative analysis (**Figure 2D, E, 3D, E, 4H, I, EV2L, M**). Further, we have discussed this point in the text of the manuscript.

Comment 5: Fig. 6A and 6B. Show the full Orai3 and Ubiquitin WBs. As presented the figure current just shows that there are ubiquitin proteins in Orai3 pull down, not that Orai3 is ubiquitinated.

Response: We acknowledge Reviewer's concern regarding the ubiquitination of Orai3. The data in the previous manuscript version only suggested that there are ubiquitinated proteins in Orai3 pulldown and did not fully address the ubiquitination status of Orai3. Our new data in the revised manuscript addresses the ubiquitination status in detail and hence strengthens our manuscript.

In the revised manuscript, we did bioinformatic analysis to identify the potential lysine residues that could be ubiquitinated in Orai3. We predicted 3 lysine residues, K2 (at N-terminal), K274 and K279 (at C-terminal) (**Figure 6B**). We mutated these lysine residues to arginine and performed ubiquitination assay to decode which lysine residues are essential for Orai3 ubiquitination. The ubiquitination assay revealed that the Orai3 K2A mutation significantly reduced ubiquitination compared to the wild-type (WT) Orai3, as well as the K274A and K279A mutants (**Figure 6C**). This demonstrates that K2 is essential for Orai3 ubiquitination. Together, these results provide strong evidence supporting the ubiquitination of Orai3. Further, as suggested by the Reviewer, the full blots are provided in the "Source Data" files of the manuscript.

Comment 6. Fig. 6C and 6D. Show the line in 6C from which the intensity profile in 6D was generated. Also give the details of the imaging setup in methods: size of the pinhole, imaging mode, etc. The colocalization is not very convincing.

Response: As recommended by the Reviewer, in the revised manuscript, we have indicated the region used for intensity profile generation by drawing a line in the representative image (**Figure 6G, I**). Further, we have updated the methodology of colocalization microscopy with details of the size of the pinhole and imaging mode.

Comment 7. Also, all the imaging and pull down do not prove conclusively direct interaction between MARCH8 and Orai3, they rather show that the proteins are in the same complex. Although it is unlikely best for the text to be moderated accordingly.

Response: We understand the concern raised by Reviewer 2 regarding direct or indirect interaction of MARCH8 and Orai3. Hence, in the revised manuscript, we have performed co-immunoprecipitation assays in which we deleted the potential MARCH8 interacting domains in Orai3 protein and checked for direct interaction of these proteins. Bioinformatic analysis and literature survey highlighted two possible MARCH8 interacting domains in Orai3. The first domain is present in the intracellular loop region, present between the 2nd and 3rd transmembrane domains at the LMVXXXL (AA113-120) motif and the second potential domain is present at the GXXXG (AA235-239) motif, present in the 3rd loop region of Orai3. We removed these domains from Orai3 protein individually and checked its effect on MARCH8 interaction. Our co-immunoprecipitation experiments showed that the Orai3 LMVXXXL deletion mutant failed to interact with MARCH8, whereas the Orai3 GXXXG deletion mutant retained interaction similar to wild-type Orai3 (**Figure 6F**). These results demonstrate that MARCH8 specifically interacts with the LMVXXXL domain within intracellular loop of Orai3. Hence, this experiment has provided conclusive evidence of direct interaction between Orai3 and MARCH8.

Comment 8. May be worth showing that overexpression of MARCH8 in the metastatic cell lines decreases their migration and metastasis as the argument is that these cells need high Orai3 but not too high. So, it would be predicted that overexpression of MARCH8 should lower Orai3 levels enough to prevent their metastasis.

Response: We would like to thank the Reviewer for this insightful suggestion. In our revised manuscript, we carried out trans-well migration assays with MARCH8 overexpression as well as MARCH8 knockdown in CFPAC-1 (metastatic) cells. Our data shows that stable lentiviral knockdown of MARCH8 increased the number of migrated CFPAC-1 cells compared to shNT CFPAC-1 cells while MARCH8 overexpression decreased the number of migrated CFPAC-1 cells compared to empty vector control cells (**Figure 9F, G**). Therefore, as pointed out by the Reviewer, MARCH8 overexpression lowers Orai3 levels in metastatic pancreatic cancer cells and hinders their metastatic potential.

Comment 9: Fig. 10. Show higher levels of Orai3 protein in the metastatic side.

Response: As suggested, we have updated the summary figure (**Figure 10**) showing higher Orai3 protein levels in the metastatic side.

Comment 10: Please show all full WBs in the supplementary data.

Response: As recommended by the Reviewer, we have provided all full western blots in the “Source Data” files of the manuscript.

Reviewer #3

The study by Raju et al. demonstrated that NFAT2 drives both Orai3 transcription and protein degradation. They find a clearly distinct mechanism between non-metastatic cancerous and metastatic cells. While in non-metastatic cells NFAT2 drives Orai3 transcription and increases Orai3 expression, in invasive and metastatic cells degradation of Orai3 is driven. They find a physical interaction of MARCH8 with Orai3 resulting in degradation. This degradation is not happening in non-metastatic cells as MARCH8 promoter is highly methylated. This study is highly interesting for a broad readership and provides a solid basis for the development of novel therapeutic strategies for cancer treatment. Before publication the authors should address a few minor comments.

Response: We would like to thank the Reviewer for believing that our study is highly interesting for a broad readership and that it provides a solid base for further translational work. We have addressed all the minor comments of the Reviewer in this revised manuscript.

Comment 1: The authors show that MARCH8 physically associates with Orai3 using Co-IP and Co-localization studies. For the co-localization studies the authors should still provide a quantitative analysis. Furthermore, can the authors detect FRET between March and Orai3? Can you please state the labels used in the co-localization experiments also in the figure legend.

Response: As suggested by the Reviewer, in the revised manuscript, we have provided quantitative analysis of Orai3 and MARCH8 co-localization. Further, we have stated the labels used in the co-localization experiment in the figure legend of the revised manuscript. Unfortunately, we could not perform FRET assay between Orai3 and MARCH8 due to limited resources. Instead, we have performed co-immunoprecipitation assays in which we deleted the potential MARCH8 interacting domain in Orai3 protein and checked for direct interaction of these proteins. Our co-immunoprecipitation experiments showed that the Orai3 LMVXXXL deletion mutant failed to interact with MARCH8, whereas the Orai3 GXXXG deletion mutant retained interaction similar to wild-type Orai3 (**Figure 6F**). These results demonstrate that MARCH8 specifically interacts with the LMVXXXL domain present in the intracellular loop of

Orai3. Taken together, we have substantially strengthened the data on Orai3 & MARCH8 physical interaction and additionally, we have identified MARCH8 interacting domain of Orai3.

Comment 2: In the abstract it is only getting clear at the end that pancreatic cancer cells are used. It would be great if the authors could introduce this fact already more at the beginning of the abstract.

Response: As recommended by the Reviewer, in the revised manuscript, we have introduced the use of pancreatic cancer cells at the beginning of the abstract.

Comment 3: In the scheme in Fig. 10, the authors highlight that Orai3 is ubiquitinated. Do they have any idea where the site of action of ubiquitination in Orai3 is located?

Response: Our robust bioinformatic analysis predicted 3 potential lysine residues which could be ubiquitinated in Orai3; K2 (at N-terminal), K274 and K279 (at C-terminal) (**Figure 6B**). In our revised manuscript, we independently mutated these lysine residues to arginine and performed ubiquitination assays to determine which lysine residues are essential for Orai3 ubiquitination. The ubiquitination assays revealed that the Orai3 K2A mutation significantly reduced ubiquitination compared to the wild-type (WT) Orai3, as well as the K274A and K279A mutants (**Figure 6C**). In summary, we demonstrate that Orai3 K2 residue is ubiquitinated.

Comment 4: In other cancer types recent reports suggest a co-expression of Orai1 and Orai3 and even the formation of heteromers. Does only Orai3 or also Orai1 play a role in pancreatic cancer cells? Could there be a difference in degradation when Orai3 forms homomers or heteromers with Orai1.

Response: We thank the reviewer for asking this interesting question. There is only one report on Orai1's role in pancreatic cancer. It was suggested that Orai1 can contribute to apoptotic resistance of pancreatic cancer cells (Kondratska et al. BBA-Molecular Cell Research, 2014). However, only one cell line i.e. PANC-1 was used in this study. While our earlier work and other studies have demonstrated that Orai3 drives pancreatic cancer metastasis (Arora et al. Cancers, 2021) and proliferation (Dubois et al. BBA-Molecular Cell Research, 2021) respectively. Therefore, emerging literature suggests that both Orai1 and Orai3 can contribute to different aspects of pancreatic cancer progression. But whether Orai1 and Orai3 form heteromers in pancreatic cancer cells remains unexplored. Further, we believe that the degradation machinery and the underlying molecular mechanisms would be analogous for both

Orai3 homomers and heteromers. Nonetheless, the rate of degradation may differ for Orai3 homomers and heteromers as literature suggests that usually proteins are more stable in large heteromeric protein complexes.

Dear Dr Motiani,

Thank you for submitting your revised manuscript (EMBOJ-2025-121095R) to The EMBO Journal, as well for your patience with our feedback. Your amended study was sent back to the referees for their scientific reassessment, and we have received reports from all of them, which I enclose below. As you will see, the reviewers state that the work has been substantially enhanced by the revisions and they are now broadly in favour of publication, pending minor revision.

Thus, we are pleased to inform you that your manuscript has been accepted in principle for publication in The EMBO Journal.

Please carefully consider the remaining minor points raised by referee #3 by adjusting the data presentation and discussion of the findings where appropriate.

Also, we now need you to take care of a number of issues related to formatting and data presentation as detailed below, which should be addressed at re-submission.

Please contact me at any time if you have additional questions related to below points.

Thank you for giving us the chance to consider your manuscript for The EMBO Journal. I look forward to your final revision.

Again, please contact me at any time if you need any help or have further questions.

Best regards,

Daniel Klimmeck

>> Please add up to five keywords to your study.

>> Author Contributions: Remove the author contributions information from the manuscript text. Note that CRediT has replaced the traditional author contributions section as of now because it offers a systematic machine-readable author contributions format that allows for more effective research assessment. and use the free text boxes beneath each contributing author's name to add specific details on the author's contribution.

More information is available in our guide to authors.
<https://www.embopress.org/page/journal/14602075/authorguide>

>> Adjust the title of the 'Competing Interests' section to 'Disclosure and Competing Interests Statement'.

>> Section order should be corrected as follows: title page with complete author information, abstract, keywords, introduction, results, discussion, methods, data availability section, acknowledgements, disclosure and competing interests statement, references, main figure legends, tables, expanded figure legends.

>> Figures: Please remove the figures from the main manuscript text file.

>> Figure callouts: Please ensure that the Appendix Tables S1 - 4 , S 6 -7 are called out in sequential order in the main text.

>> Reagents and Tools table: Please remove from the manuscript and upload as a separate file using the existing template in

the Guide For Authors, listing key reagents, experimental models, software and relevant equipment.

>> Please provide a completed source data checklist for the study as requested by the separate e-mail by my colleague Hannah Sonntag.

>> References: adjust the reference format to EMBO Journal format, 10 authors et al, and place References after the Disclosure and competing interests statement, before figure legends.

>> Data availability section: please change the statement to 'No data amenable to large-data repository deposition have been generated in this study.' .

>> Consider additional changes and comments from our production team as indicated below:

- Figure legends:

1. Please note that the exact p values are not provided in the legends of figures 1E, F, I, J; 2A, C, E, F, H, J, K, M, O; 3A, C, E, F, H, J, K, M, O; 4B, D, G, I, K, M, O, Q; 5B, D, F, H, J, L; 6H, 7C, D, E, G, H, J, K, M, N, P, Q, S, T, V; 8E, F, H, I; 9C, E, G, J; EV1 B-D; EV2 E, C, D, E, F, G, H, I, K, M, N, P, R, S, U, W; EV3 A, C, D, F, G, I, J, L, M, O, P, R; EV5 A-F;

2. Please indicate the statistical test used for data analysis in the legends of figures EV4 B, C

3. Please note that the box plots need to be defined in terms of minima, maxima, centre, bounds of box and whiskers, and percentile in the legends of figures 8A, B; EV4 B, C

4. Please note that information related to n is missing in the legends of figures 4B, D; 9C, E

Referee #1:

The authors have addressed all my comments. The manuscript is now convincingly demonstrating the interplay of MARCH8 and Orai3 in pancreatic cancer.

Referee #2:

I thank the authors for carefully and convincingly addressing all my comments. I have no additional input and recommend publication.

Referee #3:

The manuscript entitled, "NFAT2 drives both Orai3 transcription and protein degradation by harnessing the differences in epigenetic landscape of MARCH8 E3 ligase" has been extensively revised in alignment with my comments and those by the other reviewers. There are a number of very interesting observations within this study. Hence, the authors reveal that NFATc1 selectively and directly regulates Orai3 transcription, while Orai3 protein levels are differentially regulated in a stage-dependent manner. The authors further establish that MARCH8 is differentially methylated during PDAC progression, which provides potential insight into the dissociation between Orai3 RNA and protein levels during cancer progression. As outlined below, I remained unsatisfied with your explanations for the observation that both stimulation and inhibition of NFATc1 lead to increased Orai1 mRNA content in the same cells. For EMBOJ, I think that some type of reasonable and supported theory is needed for this observation. Beyond that, I have a number of additional concerns that I have no doubts can be relatively easily addressed upon revision.

Major Comments:

1. I find the data showing that both NFATc1 o/e and inhibition BOTH increase Orai3 mRNA expression to be convincing, but

difficult to interpret. The argument here is that NFATc1 drives the expression of a 3rd protein that drives Orai3 expression. If so, it should be possible to block protein production with cycloheximide and determine the effect of VIVIT on Orai3 mRNA expression. If NFATc1 is a positive regulator of Orai3 transcription, it will inhibit rather than stimulate Orai3 mRNA levels in PDAC cells. Without this, the finding that BOTH stimulation and inhibition has the same effect in the same cells is not adequately explained.

2. I noted that the authors did an extensive analysis of methylation of the MARCH8 promoter to explain the dichotomy in regulation of MARCH8 during cancer progression. Methylation differences are relatively common during cancer progression, making this a very credible finding. However, why did you not do a similar analysis on the Orai3 promoter?

Minor Comments:

1. The authors' comments regarding the uniqueness of NFAT-mediated regulation of both the target and the ubiquitin ligase are well taken. Nevertheless, it is true that short-lived proteins (immediate early genes, for example) are rapidly degraded upon expression. If there is no evidence that the latter is regulated by gene transcription, then I suppose that makes you the first to identify such a relationship. Still, I would encourage you to recognize that the coupling of transcription and degradation is not so uncommon. Indeed, generalized statements like "Normally, each regulatory process for a single target protein is orchestrated by distinct molecular players with no intermingling effects on other mechanisms" should be supported in some way, as I don't really agree with these claims.

2. Throughout the manuscript, there is non-colloquial English. This can be addressed by a copy editor, however, in some cases, these writing problems extend to interpretation. This sentence from page 3 is unacceptable: At the transcriptional level, gene expression regulates the amount of mRNA produced, while alternative splicing can lead to diverse protein variants. Problems with descriptions of basic molecular biology persist throughout the paragraph.

3. The following is an incorrect definition: calcium-activated calcium release (CRAC) channels

4. On page 7, there is a claim that NFAT binding sites are conserved through in multiple mammalian species. While that may be, only primates were tested; that is not terribly stringent. That said, perhaps the authors should limit the claim to what was actually tested.

5. The bioinformatic analysis in figure EV4 lacks any statistical significance. The description of these data in the article does not convey this properly. In any case, without statistical significance, these data cannot be used to draw any conclusions.

Rev_Com_number: RC-2025-02850

New_manu_number: EMBOJ-2025-121095R

Corr_author: Motiani

Title: NFATc1 drives both Orai3 transcription and proteolysis by harnessing differences in MARCH8 epigenome

Reviewer #1:

The authors have addressed all my comments. The manuscript is now convincingly demonstrating the interplay of MARCH8 and Orai3 in pancreatic cancer.

Response: We sincerely thank the Reviewer for concluding that our manuscript convincingly demonstrates the interplay of MARCH8 and Orai3 in pancreatic cancer.

Reviewer #2

I thank the authors for carefully and convincingly addressing all my comments. I have no additional input and recommend publication.

Response: We thank the Reviewer for appreciating our study and for finding our work highly convincing. Further, we thank him/her for recommending publication of our manuscript.

Reviewer #3

The manuscript entitled, "NFAT2 drives both Orai3 transcription and protein degradation by harnessing the differences in epigenetic landscape of MARCH8 E3 ligase" has been extensively revised in alignment with my comments and those by the other reviewers. There are a number of very interesting observations within this study. Hence, the authors reveal that NFATc1 selectively and directly regulates Orai3 transcription, while Orai3 protein levels are differentially regulated in a stage-dependent manner. The authors further establish that MARCH8 is differentially methylated during PDAC progression, which provides potential insight into the dissociation between Orai3 RNA and protein levels during cancer progression. As outlined below, I remained unsatisfied with your explanations for the observation that both stimulation and inhibition of NFATc1 lead to increased Orai1 mRNA content in the same cells. For EMBOJ, I think that some type of reasonable and supported theory is needed for this observation. Beyond that, I have a number of additional concerns that I have no doubts can be relatively easily addressed upon revision.

Response: We would like to thank the Reviewer for appreciating our revised manuscript and believing that our study is very interesting. We have addressed all the minor comments of the Reviewer in this revised manuscript.

Comment 1: I find the data showing that both NFATc1 o/e and inhibition BOTH increase Orai3 mRNA expression to be convincing, but difficult to interpret. The argument here is that NFATc1 drives the expression of a 3rd protein that drives Orai3 expression. If so, it should be possible to block protein production with cycloheximide and determine the effect of VIVIT on Orai3 mRNA expression. If NFATc1 is a positive regulator of Orai3 transcription, it will inhibit rather than stimulate Orai3 mRNA levels in PDAC cells. Without this, the finding that BOTH stimulation and inhibition has the same effect in the same cells is not adequately explained.

Response: We thank the Reviewer for stating that he/she is convinced with our data demonstrating both NFATc1 o/e and inhibition increase Orai3 mRNA expression. We apologize for not being very clear in our explanation during the first revision.

Actually, we performed NFATc1 inhibition studies using VIVIT and siRNA-mediated NFATc1 knockdown. Both of these assays show an increase in Orai3 mRNA levels in all non-metastatic,

invasive and metastatic pancreatic cancer cell lines. To understand if the increase in Orai3 mRNA levels is via transcriptional regulation, we performed luciferase assays which showed that VIVIT mediated NFAT inhibition leads to increase in luciferase activity suggesting the binding of other transcription factors on the Orai3 promoter. To corroborate this hypothesis, in our revised manuscript, we performed luciferase assays with wild type Orai3 promoter and truncated Orai3 promoter with no NFATc1 binding sites. In these assays, NFAT inhibition via VIVIT transfection led to an increase in luciferase activity in both wild type and truncated Orai3 promoter (**Figure EV2A**). Therefore, we strongly believe that in conditions of NFATc1 silencing or inhibition, other endogenous transcription factors are involved in regulating Orai3 transcription. We have not identified all the transcription factors that can modulate Orai3 upon NFAT inhibition as it is beyond the scope of this study.

Comment 2: I noted that the authors did an extensive analysis of methylation of the MARCH8 promoter to explain the dichotomy in regulation of MARCH8 during cancer progression. Methylation differences are relatively common during cancer progression, making this a very credible finding. However, why did you not do a similar analysis on the Orai3 promoter?

Response: We would like to thank the Reviewer for this insightful comment. Indeed, we are working on epigenetic changes in the Orai3 promoter during cancer progression. However, that is an independent study with a number of interesting observations, which would require substantial amount of time and resources to validate. Therefore, it is beyond the scope of this manuscript to include Orai3 epigenetics data. Further, we have discussed it in the “Discussion” section of the revised manuscript.

Minor Comment 1: The authors' comments regarding the uniqueness of NFAT-mediated regulation of both the target and the ubiquitin ligase are well taken. Nevertheless, it is true that short-lived proteins (immediate early genes, for example) are rapidly degraded upon expression. If there is no evidence that the latter is regulated by gene transcription, then I suppose that makes you the first to identify such a relationship. Still, I would encourage you to recognize that the coupling of transcription and degradation is not so uncommon. Indeed, generalized statements like "Normally, each regulatory process for a single target protein is orchestrated by distinct molecular players with no intermingling effects on other mechanisms" should be supported in some way, as I don't really agree with these claims.

Response: As suggested by the Reviewer, in the revised manuscript, we have modified the text in the “Abstract” as well as “Introduction” section.

Minor Comment 2: Throughout the manuscript, there is non-colloquial English. This can be addressed by a copy editor, however, in some cases, these writing problems extend to interpretation. This sentence from page 3 is unacceptable: At the transcriptional level, gene expression regulates the amount of mRNA produced, while alternative splicing can lead to diverse protein variants. Problems with descriptions of basic molecular biology persist throughout the paragraph.

Response: We have thoroughly proof read the manuscript and have modified the text wherever necessary. Further, we have modified the suggested paragraph in the “Introduction” section of the revised manuscript.

Minor Comment 3: The following is an incorrect definition: calcium-activated calcium release (CRAC) channels.

Response: We apologize for the typo. In the revised manuscript, we have corrected the definition of CRAC.

Minor Comment 4: On page 7, there is a claim that NFAT binding sites are conserved through in multiple mammalian species. While that may be, only primates were tested; that is not terribly stringent. That said, perhaps the authors should limit the claim to what was actually tested.

Response: As recommended by the Reviewer, we have limited the claim to the primates only in the revised manuscript.

Minor Comment 5: The bioinformatic analysis in figure EV4 lacks any statistical significance. The description of these data in the article does not convey this properly. In any case, without statistical significance, these data cannot be used to draw any conclusions.

Response: As suggested by the Reviewer and the production team, we have included the details of statistical analysis in the legend of Figure EV4.

Dear Dr Motiani,

Thank you for submitting the revised version of your manuscript. I have now evaluated your amended manuscript and concluded that the remaining minor concerns have been sufficiently addressed.

I am thus pleased to inform you that your manuscript has been accepted for publication in the EMBO Journal.

On a different note, I would like to alert you that EMBO Press offers a format for a video-synopsis of work published with us, which essentially is a short, author-generated film explaining the core findings in hand drawings, and, as we believe, can be very useful to increase visibility of the work. Please see the following link for representative examples and their integration into the article web page:

<https://www.embopress.org/doi/full/10.15252/emj.2019103932>

Best regards,

Daniel Klimmeck

Daniel Klimmeck, PhD
Senior Editor
The EMBO Journal
EMBO
Postfach 1022-40
Meyerhofstrasse 1
D-69117 Heidelberg
contact@embojournal.org
